RESEARCH COMMUNICATION

# Neural precursors of decisions that matter—an ERP study of deliberate and arbitrary choice

Uri Maoz[1,2,3,4,5,6]*, Gideon Yaffe[7], Christof Koch[8], Liad Mudrik[9,10]

[1]Department of Psychology at Crean College of Health and Behavioral Sciences, Chapman University, Orange, United States; [2]Institute for Interdisciplinary Brain and Behavioral Sciences, Chapman University, Orange, United States; [3]Anderson School of Management, University of California, Los Angeles, Los Angeles, United States; [4]Department of Psychology, University of California, Los Angeles, Los Angeles, United States; [5]Division of Humanities and Social Sciences, California Institute of Technology, Pasadena, United States; [6]Division of Biology and Bioengineering, California Institute of Technology, Pasadena, United States; [7]Yale Law School, Yale University, New Haven, United States; [8]Allen Institute for Brain Science, Seattle, United States; [9]Sagol School of Neuroscience, Tel Aviv University, Tel Aviv, Israel; [10]School of Psychological Sciences, Tel Aviv University, Tel Aviv, Israel

*For correspondence:
urimaoz@ucla.edu

**Abstract** The readiness potential (RP)—a key ERP correlate of upcoming action—is known to precede subjects' reports of their decision to move. Some view this as evidence against a causal role for consciousness in human decision-making and thus against free-will. But previous work focused on arbitrary decisions—purposeless, unreasoned, and without consequences. It remains unknown to what degree the RP generalizes to deliberate, more ecological decisions. We directly compared deliberate and arbitrary decision-making during a $1000-donation task to non-profit organizations. While we found the expected RPs for arbitrary decisions, they were strikingly absent for deliberate ones. Our results and drift-diffusion model are congruent with the RP representing accumulation of noisy, random fluctuations that drive arbitrary—but not deliberate—decisions. They further point to different neural mechanisms underlying deliberate and arbitrary decisions, challenging the generalizability of studies that argue for no causal role for consciousness in decision-making to real-life decisions.
**Editorial note:** This article has been through an editorial process in which the authors decide how to respond to the issues raised during peer review. The Reviewing Editor's assessment is that all the issues have been addressed (see decision letter).
DOI: https://doi.org/10.7554/eLife.39787.001

## Introduction

Humans typically experience freely selecting between alternative courses of action, say, when ordering a particular item off a restaurant menu. Yet a series of human studies using electroencephalography (EEG) (*Haggard and Eimer, 1999*; *Libet et al., 1983*; *Salvaris and Haggard, 2014*), fMRI (*Bode and Haynes, 2009*; *Bode et al., 2011*; *Soon et al., 2008*; *Soon et al., 2013*), intracranial (*Perez et al., 2015*), and single-cell recordings (*Fried et al., 2011*) challenged the veridicality of this common experience. These studies found neural correlates of decision processes hundreds of milliseconds and even seconds prior to the moment that subjects reported having consciously decided.

The seminal research that launched this series of studies was conducted by Benjamin Libet and colleagues (*Libet et al., 1983*). There, the authors showed that the readiness potential (RP)—a ramp-up in EEG negativity before movement onset, thought to originate from the presupplementary motor area (pre-SMA)—began before subjects reported a conscious decision to act. Libet and colleagues took the RP to be a marker for an unconscious decision to act (*Libet et al., 1983*; *Soon et al., 2008*) that, once it begins, ballistically leads to action (*Shibasaki and Hallett, 2006*). Under that interpretation, the fact that RP onset precedes the report of the onset of the conscious decision to act was taken as evidence that decisions about actions are made unconsciously. And thus the subjective human experience of freely and consciously deciding to act is but an illusion (*Harris, 2012*; *Libet et al., 1983*; *Wegner, 2002*). This finding has been at the center of the free-will debate in neuroscience for almost four decades, captivating scholars from many disciplines in and outside of academia (*Frith et al., 2000*; *Frith and Haggard, 2018*; *Haggard, 2008*; *Jeannerod, 2006*; *Lau et al., 2004*; *Mele, 2006*; *Wegner, 2002*).

Critically, in the above studies, subjects were told to arbitrarily move their right hand or flex their right wrist; or they were instructed to arbitrarily move either the right or left hand (*Haggard, 2008*; *Hallett, 2016*; *Roskies, 2010*). Thus, their decisions when and which hand to move were always unreasoned, purposeless, and bereft of any real consequence. This stands in sharp contrast to many real-life decisions that are deliberate—that is, reasoned, purposeful, and bearing consequences (*Ullmann-Margalit and Morgenbesser, 1977*): which clothes to wear, what route to take to work, as well as more formative decisions about life partners, career choices, and so on.

Deliberate decisions have been widely studied in the field of neuroeconomics (*Kable and Glimcher, 2009*; *Sanfey et al., 2006*) and in perceptual tasks (*Gold and Shadlen, 2007*). Yet, interestingly, little has been done in that field to assess the relation between decision-related activity, subjects' conscious experience of deciding, and the neural activity instantaneously contributing to this experience. Though some studies compared, for example, internally driven and externally cued decisions (*Thut et al., 2000*; *Wisniewski et al., 2016*), or stimulus-based and intention-based actions (*Waszak et al., 2005*), these were typically arbitrary decisions and actions with no real implications. Therefore, the results of these studies provide no direct evidence about potential differences between arbitrary and deliberate decisions.

Such direct comparisons are critical for the free will debate, because it is deliberate, rather than arbitrary, decisions that are at the center of philosophical arguments about free will and moral responsibility (*Breitmeyer, 1985*; *Maoz and Yaffe, 2016*; *Roskies, 2010*). Deliberate decisions typically involve more conscious and lengthy deliberation and might thus be more tightly bound to conscious processes than arbitrary ones. Consequently, if the RP is a marker for unconscious decisions, while deliberate decisions are driven more by conscious than by unconscious processes, then the RP might be substantially diminished, or even absent, for deliberate decisions.

Another reason that the RP might be completely absent during deliberate decisions has to do with a recent computational model (*Schurger et al., 2012*). This model claims that the RP—which has been deemed a preparatory signal with a causal link to the upcoming movement—actually reflects an artifact that results from a combination of (i) biased sampling stemming from the methodology of calculating this component and (ii) autocorrelation (or smoothness) in the EEG signal. The RP is calculated by aligning EEG activity (typically in electrode $C_z$) to movement onset, then segmenting a certain time duration around each movement onset (i.e., epoching), and finally averaging across all movements. Hence, we only look for an RP before movement onset, which results in biased sampling (as 'movement absent' is not probed). Put differently, we search for and generally find a ramp up in EEG negativity in $C_z$ before movement onset. But we do not search for movement onset every time there is a ramp up in EEG negativity on $C_z$. What is more, as EEG is autocorrelated, ramps up or down are to be expected (unlike, say, for white-noise activity).

Schurger and colleagues demonstrated that RPs can be modeled using a mechanistic, stochastic, autocorrelated, drift-diffusion process that integrates noise to a bound (or threshold; see Model section in Materials and methods for details). In the model, it is only the threshold crossing that reflects decision completion and directly leads to action. And thus the beginning of (what is in hindsight and on average) the ramp up toward the threshold is certainly not the completion of the decision that ballistically leads to the threshold crossing and hence to movement onset (*Schurger et al., 2012*). This interpretation of the RP thus takes the sting out of the Libet argument against free will, as the latter was based on interpreting the RP as reflecting an unconscious decision to act. Importantly for

our purposes, within the framework of the model, this artificial accumulation of stochastic fluctuations toward a threshold is expected to occur for arbitrary decisions, but not for deliberate ones. Unlike arbitrary decisions, deliberate decisions are generally not driven by random fluctuations. Rather, it is the values of the decision alternatives that mainly drive the decision and ultimately lead to action. Therefore, if the RP indeed reflects the artificial accumulation of stochastic fluctuations, as the model suggests, a key prediction of the model is that no RP will be found for deliberate decisions (see more below).

Thus, demonstrating the absence of an RP in deliberate decisions challenges the interpretation of the RP as a general index of internal, unconscious decision-making; if this interpretation were correct, such a marker should have been found for all decision types. What is more, and importantly, it questions the generalizability of any studies focused on arbitrary decisions to everyday, ecological, deliberate decisions. In particular, it challenges RP-based claims relating to moral responsibility (*Haggard, 2008*; *Libet, 1985*; *Roskies, 2010*), as moral responsibility can be ascribed only to deliberate decisions.

Here, we tested this hypothesis and directly compared the neural precursors of deliberate and arbitrary decisions—and in particular the RP—on the same subjects, in an EEG experiment. Our experiment utilized a donation-preference paradigm, in which a pair of non-profit organizations (NPOs) were presented in each trial. In deliberate-decision trials, subjects chose to which NPO they would like to donate $1000. In arbitrary-decision trials, both NPOs received an equal donation of $500, irrespective of subjects' key presses (*Figure 1*). In both conditions, subjects were instructed to report their decisions as soon as they made them, and their hands were placed on the response keys, to make sure they could do so as quickly as possible. Notably, while the visual inputs and motor outputs were identical between deliberate and arbitrary decisions, the decisions' meaning for

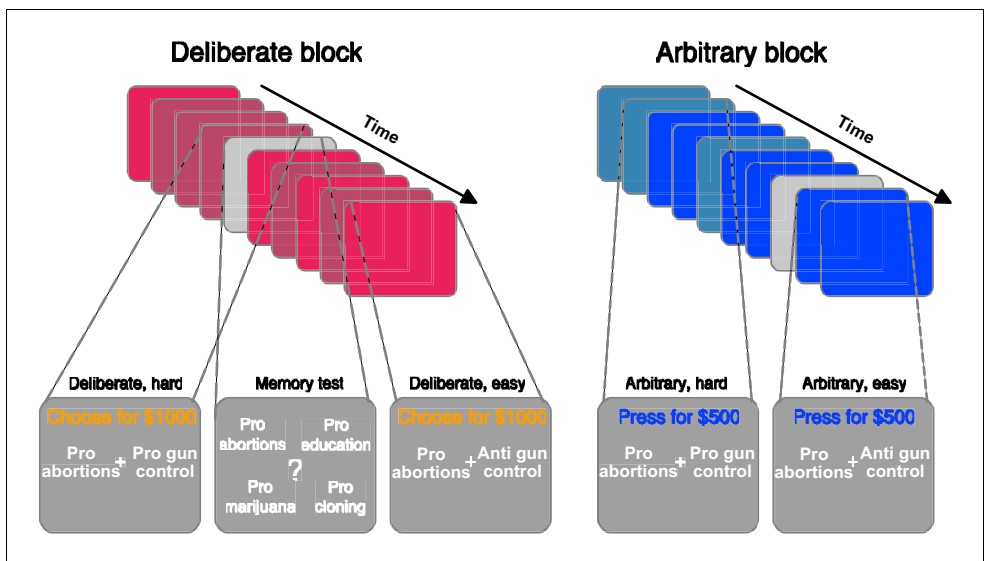

**Figure 1.** Experimental paradigm. The experiment included deliberate (red, left panel) and arbitrary (blue, right panel) blocks, each containing nine trials. In each trial, two causes—reflecting NPO names—were presented, and subjects were asked to either choose to which NPO they would like to donate (deliberate), or to simply press either right or left, as both NPOs would receive an equal donation (arbitrary). They were specifically instructed to respond as soon as they reached a decision, in both conditions. Within each block, some of the trials were easy (lighter colors) decisions, where the subject's preferences for the two NPOs substantially differed (based on a previous rating session), and some were hard decisions (darker colors), where the preferences were more similar; easy and hard trials were randomly intermixed within each block. To make sure subjects were paying attention to the NPO names, even in arbitrary trials, and to better equate the cognitive load between deliberate and arbitrary trials, memory tests (in light gray) were randomly introduced. There, subjects were asked to determine which of four NPO names appeared in the immediately previous trial. For a full list of NPOs and causes see *Supplementary file 1*.

DOI: https://doi.org/10.7554/eLife.39787.002

the subjects was radically different: in deliberate blocks, the decisions were meaningful and consequential—reminiscent of important, real-life decisions—while in arbitrary blocks, the decisions were meaningless and bereft of consequences—mimicking previous studies of volition.

## Results

### Behavioral results

Subjects' reaction times (RTs) were analyzed using a 2-way ANOVA along decision type (arbitrary/deliberate) and difficulty (easy/hard). This was carried out on log-transformed data (raw RTs violated the normality assumption; W = 0.94, p=0.001). As expected, subjects were substantially slower for deliberate (M = 2.33, SD = 0.51) than for arbitrary (M = 0.99, SD = 0.32) decisions (*Figure 2*, left; F (1,17)=114.87, p<0.0001 for the main effect of decision type). A main effect of decision difficulty was also found (F(1,17)=21.54, p<0.0005), with difficult decisions (M = 1.77, SD = 0.40) being slower than easy ones (M = 1.56, SD = 0.28). Importantly, subjects were significantly slower for hard (M = 2.52, SD = 0.62) vs. easy (M = 2.13, SD = 0.44) decisions in the deliberate case (t(17)=4.78, p=0.0002), yet not for the arbitrary case (M = 1.00, SD = 0.34; M = 0.98, SD = 0.32, for hard and easy arbitrary decisions, respectively; t(17)=1.01, p=0.33; F(1,17)=20.85, p<0.0005 for the interaction between decision type and decision difficulty). This validates our experimental manipulation and further demonstrates that, in deliberate decisions, subjects were making meaningful decisions, affected by the difference in the values of the two NPOs, while for arbitrary decisions they were not. What is more, the roughly equal RTs between easy and hard arbitrary decisions provide evidence inconsistent with concerns that subjects were deliberating during arbitrary decisions.

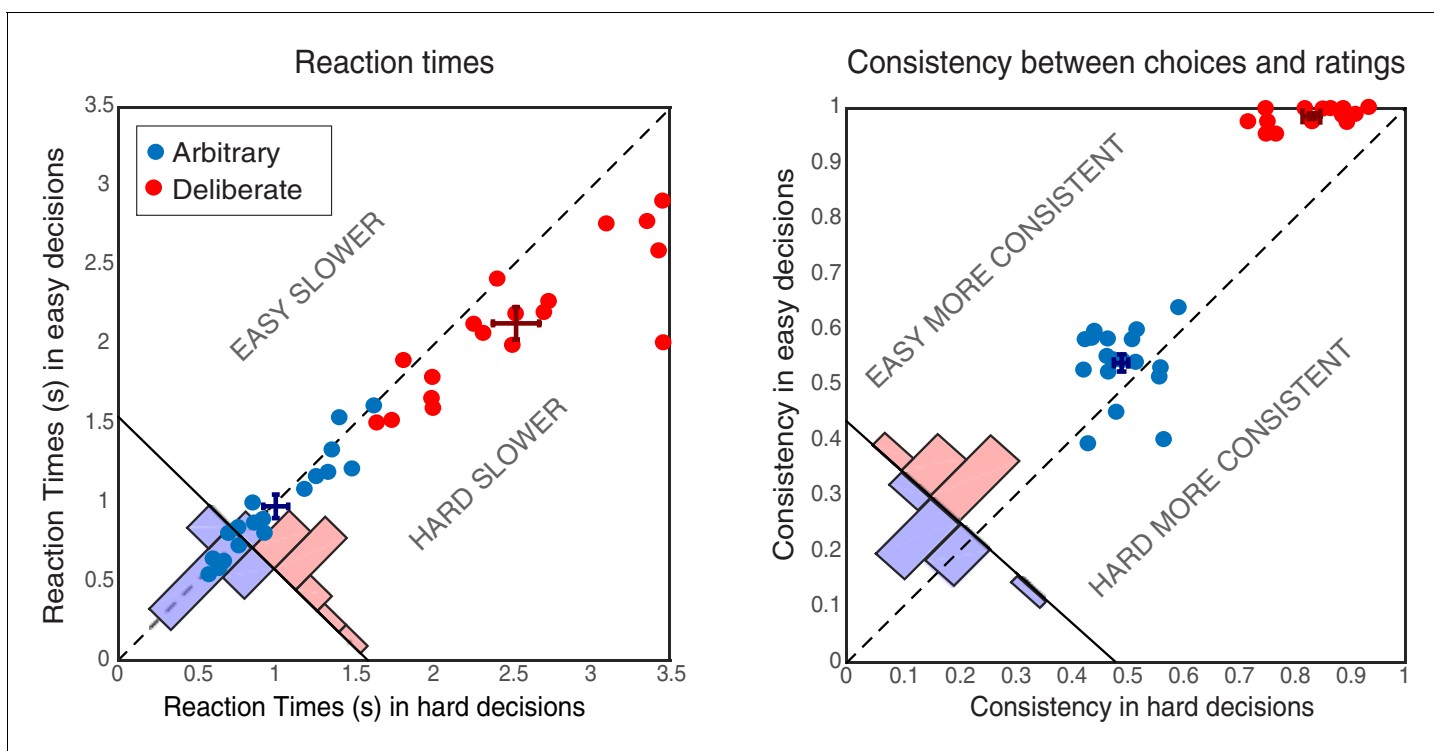

**Figure 2.** Behavioral results. Reaction Times (RTs; left) and Consistency Grades (CG; right) in arbitrary (blue) and deliberate (red) decisions. Each dot represents the average RT/CG for easy and hard decisions for an individual subject (hard decisions: x-coordinate; easy decisions: y-coordinate). Group means and SEs are represented by dark red and dark blue crosses. The red and blue histograms at the bottom-left corner of each plot sum the number of red and blue dots with respect to the solid diagonal line. The dashed diagonal line represents equal RT/CG for easy and hard decisions; data points below that diagonal indicate longer RTs or higher CGs for hard decisions. In both measures, arbitrary decisions are more centered around the diagonal than deliberate decisions, showing no or substantially reduced differences between easy and hard decisions.
DOI: https://doi.org/10.7554/eLife.39787.003

The consistency between subjects' choices throughout the main experiment and the NPO ratings they gave prior to the main experimental session was also analyzed using a 2-way ANOVA (see Materials and methods). As expected, subjects were highly consistent with their own, previous ratings when making deliberate decisions (M = 0.91, SD = 0.04), but not when making arbitrary ones (M = 0.52, SD = 0.04; *Figure 2*, right; F(1,17)=946.55, p<0.0001, BF = 2.32*$10^{29}$) for the main effect of decision type. A main effect of decision difficulty was also found (F(1,17)=57.39, p<0.0001, though BF = 1.57), with hard decisions evoking less consistent scores (M = 0.66, SD = 0.05) than easy ones (M = 0.76, SD = 0.03). Again, decision type and decision difficulty interacted (F(1,17) =25.96, p<0.0001, BF = 477.47): subjects were much more consistent with their choices in easy (M = 0.99, SD = 0.02) vs. hard (M = 0.83, SD = 0,64) deliberate decisions (t(17)=11.15, p<0.0001, BF = 3.68*$10^6$), than they were in easy (M = 0.54, SD = 0.07) vs. hard (M = 0.49, SD = 0.05) arbitrary decisions (t(17)=2.50, p=0.023, BF = 2.69). Nevertheless, though subjects were around chance (i.e., 0.5) in their consistency in arbitrary decisions (ranging between 0.39 and 0.64), it seems that some subjects were slightly influenced by their preferences in easy-arbitrary decisions trials, resulting in the significant difference between hard-arbitrary and easy-arbitrary decisions above, though the Bayes factor was inconclusive. Finally, no differences were found between subjects' tendency to press the right vs. left key in the different conditions (both main effects and interaction: F < 1).

## EEG results: Readiness Potential (RP)

The RP is generally held to index unconscious readiness for upcoming movement (*Haggard, 2008*; *Kornhuber and Deecke, 1990*; *Libet et al., 1983*; *Shibasaki and Hallett, 2006*); although more recently, alternative interpretations of the RP have been suggested (*Miller et al., 2011*; *Schmidt et al., 2016*; *Schurger et al., 2012*; *Trevena and Miller, 2010*; *Verleger et al., 2016*). It has nevertheless been the standard component studied in EEG versions of the Libet paradigm (*Haggard, 2008*; *Haggard and Eimer, 1999*; *Hallett, 2007*; *Libet, 1985*; *Libet et al., 1983*; *Libet et al., 1982*; *Miller et al., 2011*; *Schurger et al., 2012*; *Shibasaki and Hallett, 2006*; *Trevena and Miller, 2010*). As is common, we measured the RP over electrode $C_z$ in the different conditions by averaging the activity across trials in the 2 s prior to subjects' movement.

Focusing on the last 500 ms before movement onset for our statistical tests, we found a clear RP in arbitrary decisions, yet RP amplitude was not significantly different from 0 in deliberate decisions (*Figure 3A*; F(1,17)=11.86, p=0.003, BF = 309.21 for the main effect of decision type; in t-tests against 0 for this averaged activity in the different conditions, corrected for multiple comparisons, an effect was only found for arbitrary decisions (hard: t(17)=5.09, p=0.0001, BF = 307.38; easy: t(17) =5.75, p<0.0001, BF = 1015.84) and not for deliberate ones). The Bayes factor—while trending in the right direction—indicated inconclusive evidence (hard: t(17)=1.24, p>0.5, BF = 0.47; easy: t(17) =1.84, p=0.34, BF = 0.97). Our original baseline was stimulus locked (see Materials and methods). And we hypothesized that the inconclusive Bayes factor for deliberate trials had to do with a constant, slow, negative drift that our model predicted for deliberate trials (see below) rather than reflecting a typical RP. As the RTs for deliberate trials were longer than for arbitrary ones, this trend might have become more pronounced for those trials. To test this, we switched the baseline period to −1000 ms to −500 ms relative to *movement* onset (i.e., a baseline that immediately preceded our time of interest window). Under this analysis, we found moderate evidence that deliberate decisions (pooled across decision difficulty) are not different from 0 (BF = 0.332), supporting the claim that the RP during the last 500 ms before response onset was completely absent (BF for similarly pooled arbitrary decisions was 5.07·$10^4$).

In an effort to further test for continuous time regions where the RP is different from 0 for deliberate and arbitrary trials, we ran a cluster-based nonparametric permutation analysis (*Maris and Oostenveld, 2007*) for all four conditions against 0. Using the default parameters (see Materials and methods), we found a prolonged cluster (~1.2 s) of activation that reliably differed from 0 in both arbitrary conditions (designated by horizontal blue-shaded lines above the x axis in *Figure 3A*). The same analysis revealed no clusters of activity differing from zero in either of the deliberate conditions.

In a similar manner, regressing voltage against time for the last 1000 ms before response onset, the downward trend was significant for arbitrary decisions (*Figure 3B*; p<0.0001, BF >$10^{25}$ for both easy and hard conditions) but not for deliberate decisions, with the Bayes factor indicating conclusive evidence for no effect (hard: p>0.5, BF = 0.09; easy: p=0.35, BF = 0.31; all Bonferroni corrected

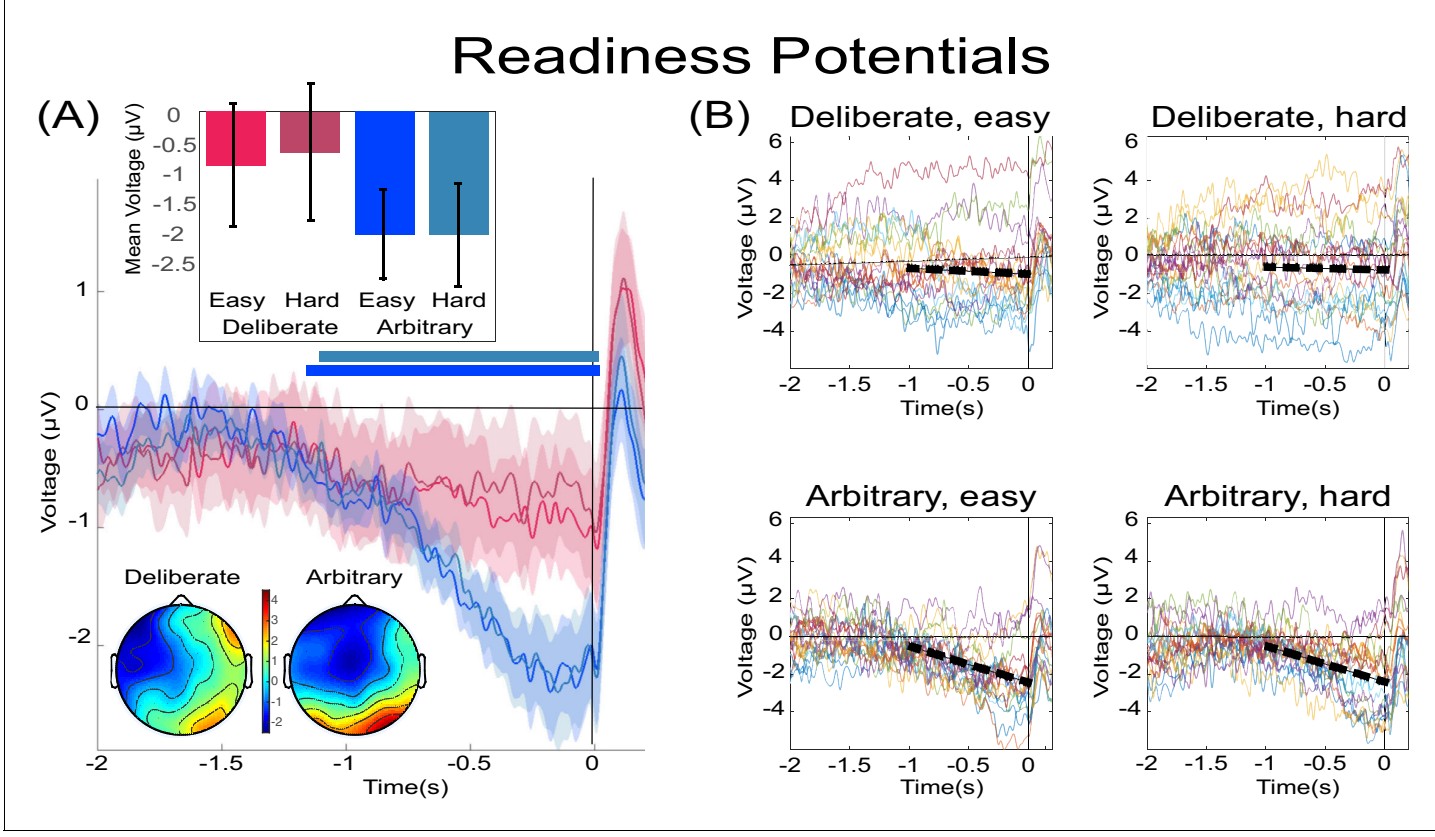

**Figure 3.** The readiness potentials (RPs) for deliberate and arbitrary decisions. (**A**) Mean and SE of the readiness potential (RP; across subjects) in deliberate (red shades) and arbitrary (blue shades) easy and hard decisions in electrode C$_z$, as well as scalp distributions. Zero refers to time of right/left movement, or response, made by the subject. Notably, the RP significantly differs from zero and displays a typical scalp distribution for arbitrary decisions only. Similarly, temporal clusters where activity was significantly different from 0 were found for arbitrary decisions only (horizontal blue lines above the x axis). Scalp distributions depict the average activity between −0.5 and 0 s, across subjects. The inset bar plots show the mean amplitude of the RP, with 95% confidence intervals, over the same time window. Response-locked potentials with an expanded timecourse, and stimulus-locked potentials are given in *Figure 6B and A*, respectively. The same (response-locked) potentials as here, but with a *movement-locked baseline* of −1 to −0.5 s (same as in our Bayesian analysis), are given in *Figure 6C*. (**B**) Individual subjects' C$_z$ activity in the four conditions (n = 18). The linear-regression line for voltage against time over the last 1000 ms before response onset is designated by a dashed, black line. The lines have slopes significantly different from 0 for arbitrary decisions only. Note that the waveforms converge to an RP only in arbitrary decisions.
DOI: https://doi.org/10.7554/eLife.39787.004

for multiple comparisons). Notably, this pattern of results was also manifested for single-subject analysis (*Figure 4*; 14 of the 18 subjects had significant downward slopes for arbitrary decisions—that is, p<0.05, Bonferroni corrected for multiple comparisons—when regressing voltage against time for every trial over the last 1000 ms before response onset; but only 5 of the 18 subjects had significant downward slopes for the same regression analysis for deliberate decisions; see Materials and methods. In addition, the average slopes for deliberate and arbitrary decisions were −0.28 ± 0.25 and −1.9 ± 0.32 (mean ± SE), respectively, a significant difference: t(17)=4.55, p<0.001, BF = 380.02). The regression analysis complements the averaged amplitude analysis above, and further demonstrates that the choice of baseline cannot explain our results. This is because the slopes of linear regressions are, by construction, independent of baseline.

## Control analyses

We further tested whether differences in reaction time between the conditions, eye movements, filtering, and subjects' consistency scores might explain our effect. We also tested whether the RPs might reflect some stimulus-locked potentials or be due to baseline considerations.

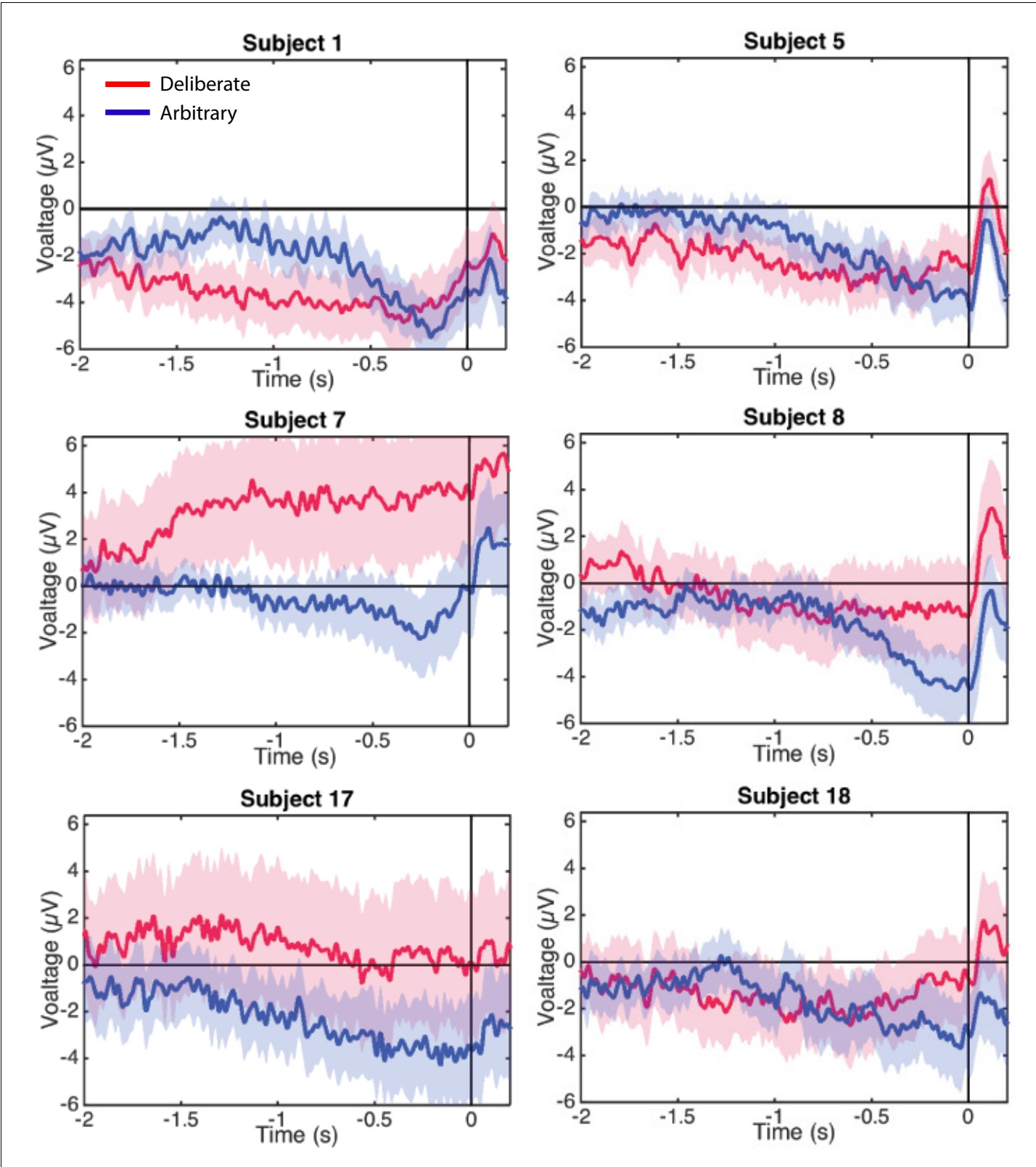

**Figure 4.** Individual-subjects RPs. Six examples of for individual subjects' RPs for deliberate decisions (in red) and arbitrary ones (in blue) pooled across decision difficulty.

DOI: https://doi.org/10.7554/eLife.39787.005

## Differences in reaction times (RT) between conditions, including stimulus-locked potentials and baselines, do not drive the effect

RTs in deliberate decisions were typically more than twice as long as RTs in arbitrary decisions. We therefore wanted to rule out the possibility that the absence of RP in deliberate decisions stemmed from the difference in RTs between the conditions. We carried out six analyses for this purpose. First, we ran a median split analysis—dividing the subjects into two groups based on their RTs: lower (faster) and higher (slower) than the median, for deliberate and arbitrary trials, respectively. We then ran the same analysis using only the faster subjects in the deliberate condition (M = 1.91 s, SD = 0.25) and the slower subjects in the arbitrary condition (M = 1.25 s, SD = 0.23). If RT length affects RP amplitudes, we would expect the RP amplitudes to be more similar between these two groups. However, though there were only half the data points, a similar pattern of results to those over the whole dataset was observed (*Figure 5A*; compare to *Figure 3A*). Deliberate and arbitrary decisions were still reliably different (F(1,17)=5.22, p=0.03), with significant RPs found in arbitrary (easy: t(8)=4.57, p=0.0018; hard: t(8)=4.09, p=0.0035), but not deliberate (easy: t(8)=1.92, p=0.09; hard: t(8)=0.63, p=0.54) decisions. In addition, the RPs for arbitrary decisions were not significantly different between the subjects with above-median RTs and the entire population for the easy or hard conditions (easy: t(25)=0.14, p>0.5; hard: t(25)=0.56, p>0.5). Similarly, the RPs for deliberate decisions were not significantly different between the subjects with below-median RTs and the entire population for the easy or hard conditions (easy: t(25)=-0.34, p>0.5; hard: t(25)=0.17, p>0.5). This suggest that RTs do not reliably affect $C_z$ activation for deliberate or arbitrary decisions in our results.

Second, we regressed the difference between RPs in deliberate and arbitrary decisions (averaged over the last 500 ms before response onset) against the difference between the RTs in these two conditions for each subject (*Figure 5B*). Again, if RT length affects RP amplitudes, we would expect differences between RTs in deliberate and arbitrary conditions to correlate with differences between RPs in the two conditions. But no correlation was found between the two measures (r = 0.28, t(16) =0.86, p=0.4). We further tried regressing the RP differences on RT differences. The regression did not produce any reliable relation between RT and RP differences (regression line: y = 0.54 [CI −0.8, 1.89] x - 0.95 [CI −2.75, 0.85]; the $R^2$ was very low, at 0.05 (as expected from the r value above), and, as the confidence intervals suggest, the slope was not significantly different from 0, F(1,16) =0.74, p=0.4).

While the results of the above analyses suggested that our effects do not stem from differences between the RTs in deliberate and arbitrary decisions, the average RTs for fast deliberate subjects were still 660 ms slower than for slow arbitrary subjects. In addition, we had only half of the subjects in each condition due to the median split, raising the concern that some of our null results might have been underpowered. We also wanted to look at the effect of cross-trial variations within subjects and not just cross-subjects ones. We therefore ran a third, within-subjects analysis. We combined the two decision difficulties (easy and hard) for each decision type (arbitrary and deliberate) for greater statistical power. And then we took the faster (below-median RT) deliberate trials and slower (above-median RT) arbitrary trials for each subject separately. So, this time we had 17 subjects (again, one was removed) and better powered results. Here, fast deliberate arbitrary trials (M = 1.63 s, SD = 0.25) were just 230 ms slower than slow arbitrary decisions (M = 1.40 s, SD = 0.45), on average. This cut the difference between fast deliberate and slow arbitrary by about 2/3 from the between-subjects analysis. We then computed the RPs for just these fast deliberate and slow arbitrary trials within each subject (*Figure 5C*). Visually, the pattern there is the same as the main analysis (*Figure 3A*). What is more, deliberate and arbitrary decisions remained reliably different (t(16)=3.36, p=0.004). Arbitrary trials were again different from 0 (t(16)=-4.40, p=0.0005), while deliberate trials were not (t(16)=-1.54, p=0.14).

We further regressed the within-subject differences between RPs in fast deliberate and slow arbitrary decisions (defined as above) against the differences between the corresponding RTs for each subject to ascertain that such a correlation would not exist for trials that are closer together. We again found no reliable relation between the two differences (*Figure 5D*; regression line: y = 1.27 [CI −0.2, 2.73] x - 0.95 [CI 0.14, 1.76]; $R^2$ = 0.18).

Yet another concern that could relate to the RT differences among the conditions is that the RP in arbitrary blocks might actually be some potential evoked by the stimuli (i.e., the presentations of the

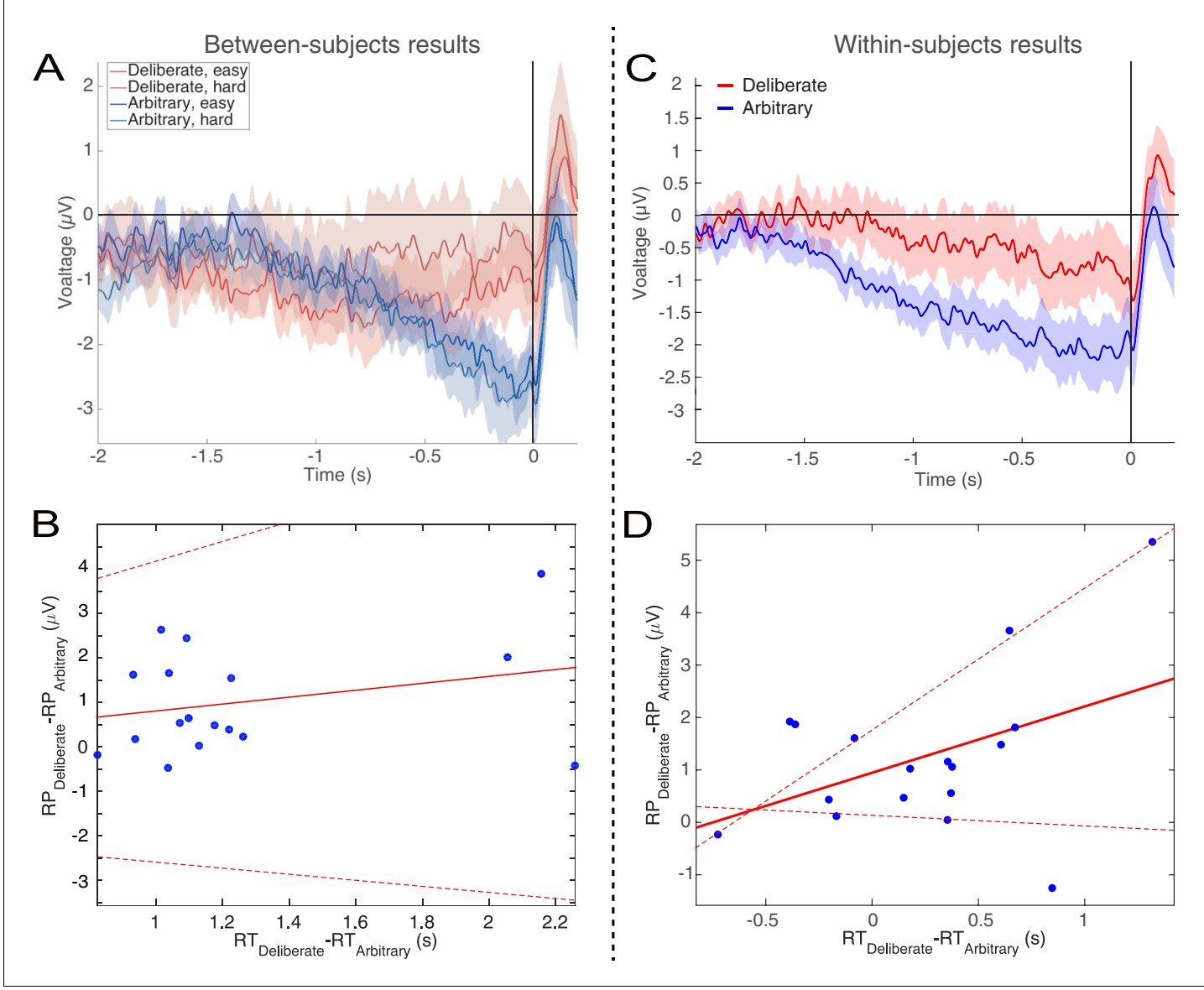

**Figure 5.** Relations between RTs and RPs between subjects (A and B) and within subjects (C and D). (**A**) The subjects with above-median RTs for arbitrary decisions (in blue) and below-median RTs for deliberate decisions (in red), show the same activity pattern that was found in the main analysis (compare *Figure 3A*). (**B**) A regression of the difference between the RPs versus the difference between the RTs for deliberate and arbitrary decisions for each subject. The equation of the regression line (solid red) is y = 0.54 [CI −0.8, 1.89] x - 0.95 [CI −2.75, 0.85] (confidence intervals: dashed red lines). The $R^2$ is 0.05. One subject, #7, had an RT difference between deliberate and arbitrary decisions that was more than six interquartile ranges (IQRs) away from the median difference across all subjects. That same subject's RT difference was also more than 5 IQRs higher than the 75th percentile across all subjects. That subject was therefore designated an outlier and removed only from this regression analysis. (**C**) For each subject separately, we computed the RP using only the faster (below-median RT) deliberate trials and slower (above-median RT) arbitrary trials. The pattern is again the same as the one found for the main analysis. (**D**) We computed the same regression between the RP differences and the RT differences as in B, but this time the median split was within subjects. The equation of the regression line is y = 1.27 [CI −0.2, 2.73] x - 0.95 [CI 0.14, 1.76]. The $R^2$ is 0.18.
DOI: https://doi.org/10.7554/eLife.39787.006

two causes), specifically in arbitrary blocks, where the RTs are shorter (and thus stimuli-evoked effects could still affect the decision). In particular, a stimulus-evoked potential might just happen to bear some similarity to the RP when plotted locked to response onset. To test this explanation, we ran a fifth analysis, plotting the potentials in all conditions, locked to the onset of the stimulus (*Figure 6A*). We also plotted the response-locked potentials across an expanded timecourse for comparison (*Figure 6B*). If the RP-like shape we see in *Figures 3A* and *6B* is due to a stimulus-

locked potential, we would expect to see the following before the four mean response onset times (indicated by vertical lines at 0.98 and 1.00, 2.13, and 2.52 s for arbitrary easy, arbitrary hard, deliberate easy, and deliberate hard, respectively) in the stimulus-locked plot (*Figure 6A*): Consistent potentials, which precede the mean response times, that would further be of a similar shape and magnitude to the RPs found in the decision-locked analysis in the arbitrary condition (though potentially more smeared for stimulus locking). We thus calculated a stimulus-locked version of our ERPs, using the same baseline (*Figure 6A*). As the comparison between *Figure 6A and B* clearly shows, no such consistent potentials were found before the four response times, nor were these potentials similar to the RP in either shape or magnitude (their magnitudes are at the most around 1μV, while the RP magnitudes we found are around 2.5 μV; *Figures 3A* and *6B*). This analysis thus suggests that it is unlikely that a stimulus-locked potential drives the RP we found.

Notably, the stimulus-locked alignment did imply that the arbitrary easy condition evoked a stronger activity in roughly the last 0.5 s before stimulus onset. However, this prestimulus activity cannot explain the response-locked RP, as it was found *only in arbitrary easy trials* and not in arbitrary hard trials. At the same time, the response-locked RP did not differ between these conditions. What is more, easy and hard trials were randomly interspersed within deliberate and arbitrary blocks, and the subject discovered the trial difficulty only at stimulus onset. Thus, there could not have been differential preparatory activity that varies with decision difficulty. This divergence in one condition only is accordingly not likely to reflect any preparatory RP activity.

One more concern is that the differences in RTs may affect the results in the following manner: Because the main baseline period we used thus far was 1 to 0.5 s before stimulus onset, the duration from the baseline to the decision varied widely between the conditions. To make sure this difference in temporal distance between the baseline period and the response to which the ERPs were locked did not drive our results, we recalculated the potentials for all conditions with a *response-locked* baseline of −1 to −0.5 s (*Figure 6C*; the same baseline we used for the Bayesian analysis above). The rationale behind this choice of baseline was to have the time that elapsed from baseline to response onset be the same across all conditions. As is evident in *Figure 6C*, the results for this new baseline were very similar to those for the stimulus-locked baseline we used before. Focusing again on the −0.5 to 0 s range before response onset for our statistical tests, we found a clear RP in arbitrary decisions, yet RP amplitude was not significantly different from 0 in deliberate decisions (*Figure 6C*; ANOVA F(1,17)=12.09, p=0.003 for the main effect of decision type; in t-tests against 0, corrected for multiple comparisons, an effect was only found for arbitrary decisions (hard: t(17) =4.13, p=0.0007; easy: t(17)=4.72, p=0.0002) and not for deliberate ones (hard: t(17)=0.38, p>0.5; easy: t(17)=1.13, p=0.27). This supports the notion that the choice of baseline does not strongly affect our main results. Taken together, the results of the six analyses above provide strong evidence against the claim that the differences in RPs stem from or are affected by the differences in RTs between the conditions.

## Eye movements do not affect the results

Though ICA was used to remove blink artifacts and saccades (see Materials and methods), we wanted to make sure our results do not stem from differential eye movement patterns between the conditions. We therefore computed a saccade-count metric (SC; see Materials and methods) for each trial for all subjects. Focusing again on the last 500 ms before response onset, we computed mean (± s.e.m.) SC values of 1.65 ± 0.07 and 1.67 ± 0.06 saccades for easy and hard deliberate decisions, respectively, and 1.69 ± 0.07 and 1.73 ± 0.07 saccades for easy and hard arbitrary decisions, respectively. We found no reliable differences between the number of saccades during deliberate and arbitrary trials (F(1,17)=2.56, p=0.13 for main effect of decision type).

We further investigated potential effects of saccades by running a median-split analysis—dividing the trials for each subject into two groups based on their SC score: lower and higher than the median, for deliberate and arbitrary trials, respectively. We then ran the same analysis using only the trials with more saccades in the deliberate condition (SC was 2.02 ± 0.07 and 2.04 ± 0.07 for easy and hard, respectively) and those with less saccades for the arbitrary condition (SC was 1.33 ± 0.07 and 1.31 ± 0.08 for easy and hard, respectively). If the number of saccades affects RP amplitudes, we would expect that the differences in RPs between arbitrary and deliberate trials will diminish, or even reverse (as now we had more saccades in the deliberate condition). However, though there

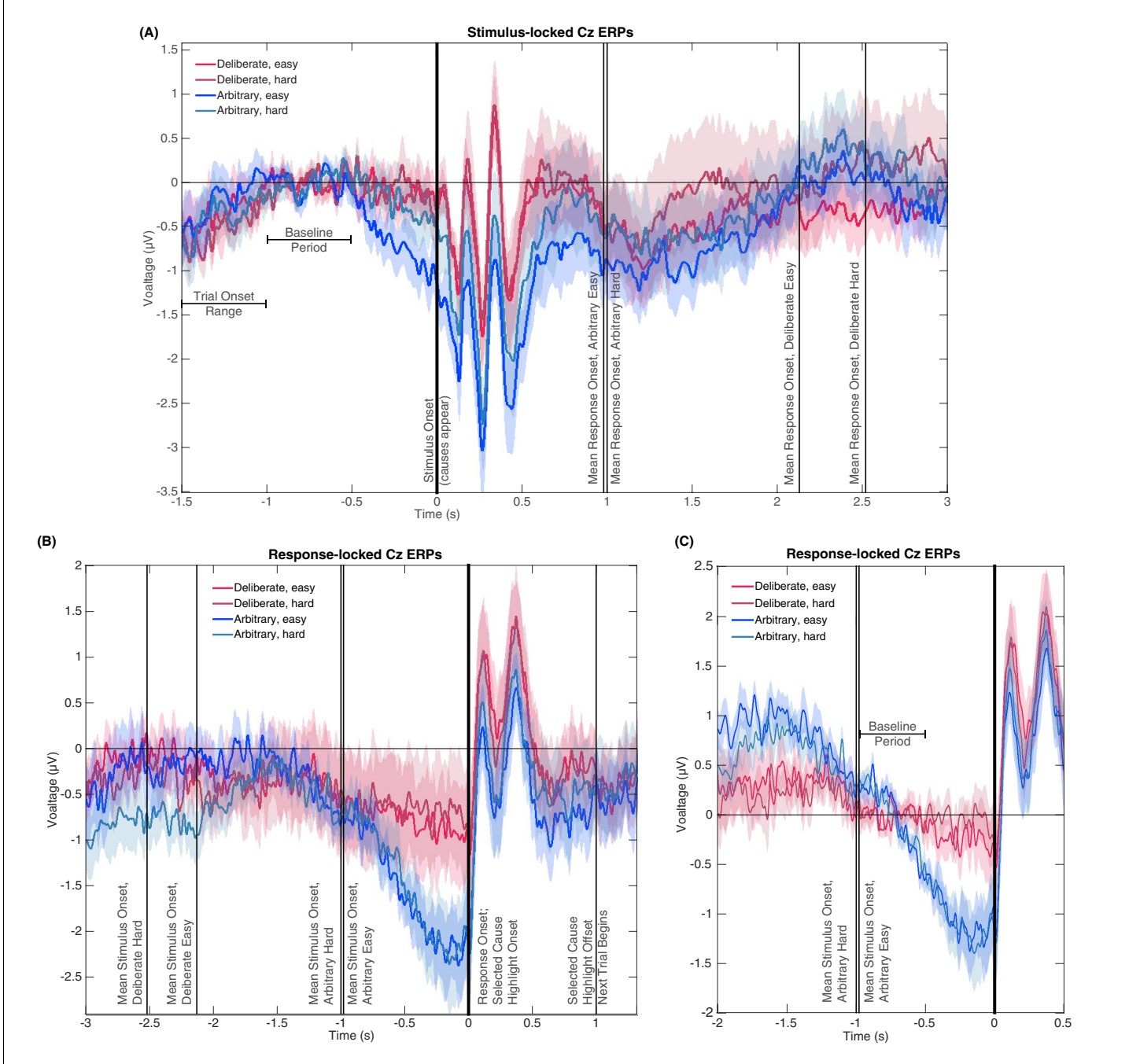

**Figure 6.** Stimulus- and response-locked Cz-electrode ERPs with different baselines and overlaid events. (**A**) Stimulus-locked waveforms including the trial onset range, baseline period, and mean reaction times for all four experimental conditions. (**B**) Response-locked waveforms with mean stimulus onsets for all four conditions as well as the offset of the highlighting of the selected cause and the start of the next trial. (**C**) Same potentials and timeline as *Figure 3A*, but with a *response-locked* baseline of −1 to −0.5 s—the same baseline used for our Bayesian analysis.
DOI: https://doi.org/10.7554/eLife.39787.007

were only half the data points for each subject in each condition, a similar pattern of results to those over the whole dataset was observed: Deliberate and arbitrary decisions were still reliably different within the median-split RPs ($F(1,17)=16.70$, $p<0.001$), with significant RPs found in arbitrary (easy: $t(17)=4.79$, $p=0.002$; hard: $t(17)=5.77$, $p<0.001$), but not deliberate (easy: $t(17)=0.90$, $p=0.38$; hard: $t(17)=0.30$, $p>0.5$) decisions. In addition, we compared the RP data across all the trials with the

median-split RP data above. No significant differences were found for arbitrary decisions (easy: t(17)=1.02, p=0.32; hard: t(17)=0.75, p=0.46) or for deliberate decisions (easy: t(17)=1.63, p=0.12; hard: t(17)=1.47, p=0.16). Taken together, the analyses above provide strong evidence against the involvement of eye movements in our results.

### Testing alternative explanations

We took a closer look at subjects' behavior in the easy arbitrary condition, where some subjects had a consistency score that was further above 0.5 (chance) than others. It seems like those subjects had a greater difficulty ignoring their preferences, despite the instructions to do so. We therefore wanted to test to what extent the RP of those subjects was similar to the RPs of the other subjects. Focusing on the eight subjects that had a consistency score above 0.55 (M = 0.59, SD = 0.03) and comparing their RPs to those of the 10 other subjects (consistency M = 0.50, SD = 0.06) in easy arbitrary trials, we found no reliable differences (t(16)=0.94, p=0.36). This is not surprising, as the mean consistency score of these subjects—though higher than chance—was still far below their consistency score for easy deliberate decisions (M = 0.99, SD = 0.02).

### High-pass filter cutoff frequency does not affect the results

Finally, another alternative explanation for the absence of an RP in deliberate decisions might rely on our selection of high-pass filter cutoff frequency, which was 0.1 Hz. Though this frequency was used in some studies of the RP (e.g., *Lew et al., 2012*; *MacKinnon et al., 2013*), others opted for lower cutoff frequencies (e.g., *Haggard and Eimer, 1999*). Arguably, a higher cutoff frequency for the high-pass filter might reduce the chances to find the RP, which is a low-frequency component. And this might have affected the RP for deliberate decisions more than the RP for arbitrary ones, given the slower RTs there. To examine this possible confound, we reanalyzed the data using a 0.01 high-pass filter. This reduced the number of usable trials for each subject, as it allowed lower-frequency trends to remain in the data. Given that our focus was on arbitrary vs. deliberate decisions (with decision difficulty serving mostly to validate the manipulation), we collapsed the trials across decision difficulty, and only tested RP amplitudes in arbitrary vs. deliberate decisions against each other and against zero. In line with our original results, a difference was found between RP amplitude in the two conditions (t(13)=2.29, p=0.039), with RP in the arbitrary condition differing from zero (t(13)=-5.71, p<0.0001), as opposed to the deliberate condition, where it did not (t(13)=-0.76, p=0.462). This provides evidence against the claim that our results are due to our choice of high-pass filter.

## EEG results: Lateralized Readiness Potential (LRP)

The LRP, which reflects activation processes within the motor cortex for action preparation after action selection (*Eimer, 1998*; *Masaki et al., 2004*), was measured by subtracting the difference potentials (C3-C4) in right-hand response trials from this difference in left-hand response trials and averaging the activity over the same time window (*Eimer, 1998*; *Haggard and Eimer, 1999*). In this purely motor component, no difference was found between the two decision types and conclusive evidence against an effect of decision type was further found (*Figure 7*; all Fs < 0.35; BF = 0.299). Our analysis of EOG channels suggests that some of that LRP might be driven by eye movements (we repeated the LRP computation on the EOG channels instead of C3 and C4). However, the shape of the eye-movement-induced LRP is very different from the LRP we calculated from C3 and C4. Also, the differences that we found between conditions in the EOG LRP are not reflected in the C3/C4 LRP. So, while our LRP might be boosted by eye movements, it is not strictly driven by these eye movements.

## Modeling

The main finding of this study—the absent (or at least strongly diminished) RP in deliberate decisions, suggesting different neural underpinnings of arbitrary and deliberate decisions—is in line with a recent study using a drift-diffusion model (DDM) to investigate the RP (*Schurger et al., 2012*). There, the RP was modeled as an accumulation of white noise (that results in autocorrelated noise) up to a hard threshold. When activity crosses that threshold, it designates decision completion leading to movement. The model focuses on the activity leading up to the threshold crossing, when that

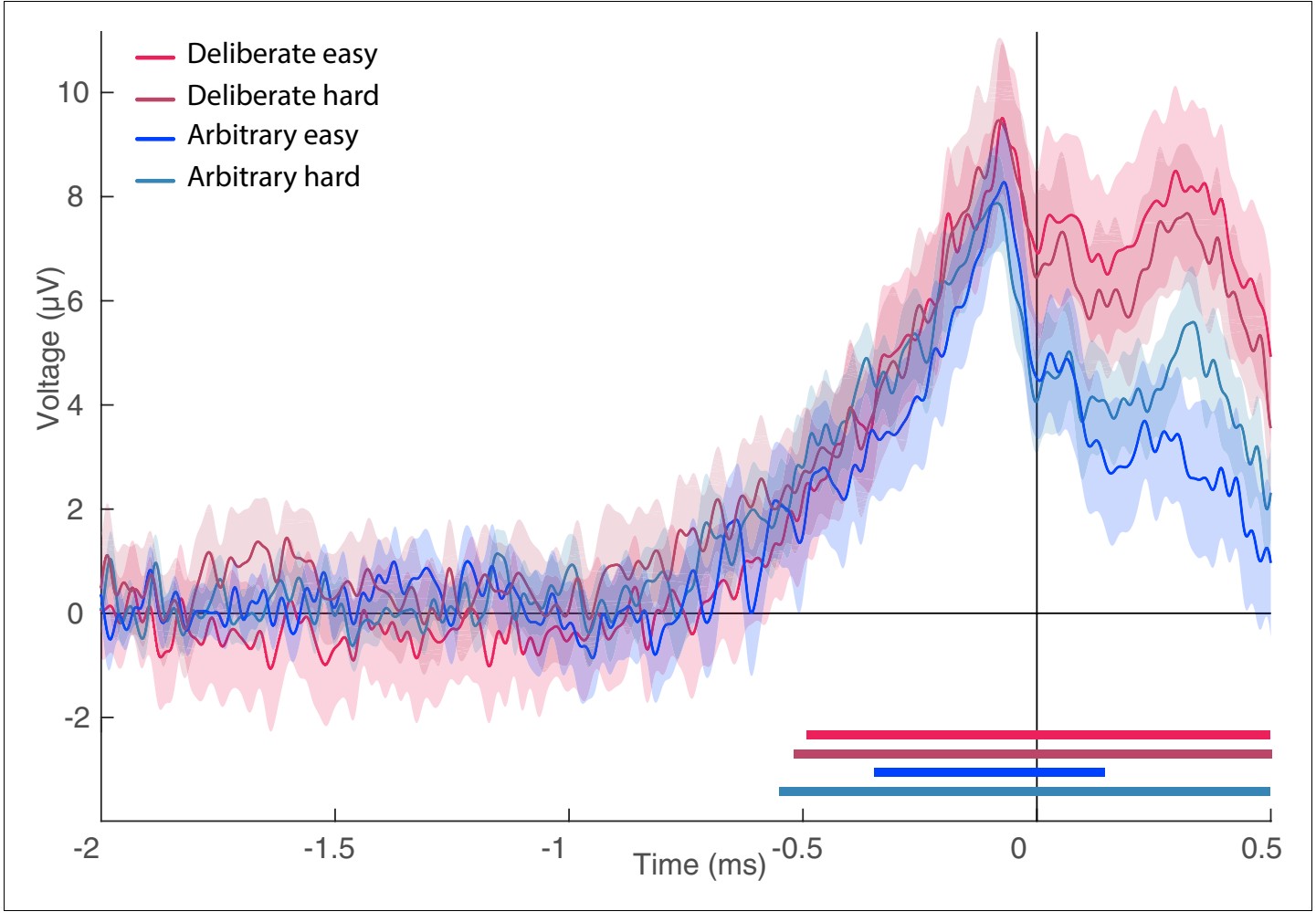

**Figure 7.** Lateralized readiness potential (LRP). The lateralized LRP for deliberate and arbitrary, easy and hard decisions. No difference was found between the conditions (ANOVA all Fs < 0.35). Temporal clusters where the activity for each condition was independently found to be significantly different from 0 are designated by horizontal thick lines at the bottom of the figure (with their colors matching the legend).
DOI: https://doi.org/10.7554/eLife.39787.008

activity is time-locked to the onset of the threshold crossing (corresponding to movement-locked epochs in EEG). Averaging across many threshold crossings, this autocorrelation activity resembles an RP (*Schurger et al., 2012*). Hence, according to this model, the exact time of the threshold crossing leading to response onset is largely determined by spontaneous, subthreshold, stochastic fluctuations of the neural activity. This interpretation of the RP challenges its traditional understanding as stemming from specific, unconscious preparation for, or ballistic-like initiation of, movement (*Shibasaki and Hallett, 2006*). Instead, Schurger and colleagues claimed, the RP is not a cognitive component of motor preparation; it is an artifact of accumulating autocorrelated noise to a hard threshold and then looking at signals only around threshold crossings.

We wanted to investigate whether our results could be accommodated within the general framework of the Schurger model, though with the deliberate and arbitrary decisions mediated by two different mechanisms. The first mechanism is involved in value assessment and drives deliberate decisions. It may be subserved by brain regions like the Ventromedial Prefrontal Cortex; VMPFC, (*Ramnani and Owen, 2004*; *Wallis, 2007*). But, for the sake of the model, we will remain agnostic about the exact location associated with deliberate decisions and refer to this region as *Region X*. A second mechanism, possibly at the (pre-)SMA, was held to generate arbitrary decisions driven by random, noise fluctuations.

Accordingly, we expanded the model developed by *Schurger et al. (2012)* in two manners. First, we defined two DDM processes—one devoted to value-assessment (in Region X) and the other to noise-generation (in SMA; see *Figure 8A* and Materials and methods). Both of them were run during both decision types, yet the former determined the result of deliberate trials, and the latter determined the results of arbitrary trials. Second, Schurger and colleagues modeled only *when* subjects would move and not *what* (which hand) subjects would move. We wanted to account for the fact that, in our experiment, subjects not only decided when to move, but also what to move (either to indicate which NPO they prefer in the deliberate condition, or to generate a meaningless right/left movement in the arbitrary condition). We modeled this by defining two types of movement. One was moving the hand corresponding to the location of the NPO that was rated higher in the first, rating part of the experiment (the *congruent* option; see Materials and methods). The other was moving the hand corresponding to the location of the lower-rated NPO (the *incongruent* option). We used the race-to-threshold framework to model the decision process between this pair of leaky, stochastic accumulators, or DDMs. One DDM simulated the process that leads to selecting the congruent option, and the other simulated the process that leads to selecting the incongruent option (see again *Figure 8A*). (We preferred the race-to-threshold model over a classic DDM with two opposing thresholds because we think it is biologically more plausible [*de Lafuente et al., 2015*] and because it is easier to see how a ramp-up-like RP might be generated from such a model without requiring a vertical flip of the activity accumulating toward one of the thresholds in each model run.) Hence, in each model run, the two DDMs in Region X and the two in the SMA ran in parallel; the first one to cross the threshold (only in Region X for deliberate decisions and only in the SMA for arbitrary ones) determined decision completion and outcome. Thus, if the DDM corresponding to the congruent (incongruent) option reached the threshold first, the trial ended with selecting the congruent (incongruent) option. For deliberate decisions, the congruent cause had a higher value than the incongruent cause and, accordingly, the DDM associated with the congruent option had a higher drift rate than that of the DDM associated with the incongruent option. For arbitrary decisions, the values of the decision alternatives mattered little and this was reflected in the small differences among the drift rates and in other model parameters (*Table 1*).

Therefore, within this framework, $C_z$-electrode activity (above the SMA) should mainly reflect the SMA component (Note that we suggest that noise generation might be a key function of the SMA and other brain regions underneath the $C_z$ electrode, *specifically during this task*. When subjects make arbitrary decisions, these might be based on some symmetry-breaking mechanism, which is driven by random fluctuations that are here simulated as noise. Thus, we neither claim nor think that noise generation is the main purpose or function of these brain regions in general.) And so, finding that the model-predicted EEG activity resembling the actual EEG pattern we found would imply that our findings are compatible with an account by which the RP represents an artifactual accumulation of stochastic, autocorrelated noise, rather than indexing a genuine marker of an unconscious decision ballistically leading to action.

For ease of explanation, and because decision difficulty had no consistent effect on the EEG data, we focus the discussion below on easy decisions (though the same holds for hard decisions). For arbitrary decisions, the SMA (or Noise) Component of the model is the one determining the decisions, and it is also the one which we pick up in electrode $C_z$. Hence, the resulting activity would be much like the original *Schurger et al. (2012)* model and we would expect to see RP-like activity, which we do see (*Figure 9B*). But the critical prediction of our model for our purposes relates to what happens during deliberate decisions in the SMA ($C_z$ electrode). According to our model, the race-to-threshold pair of DDMs that would determine deliberate decisions and trigger the ensuing action is the value-assessment one in Region X. Hence, when the first DDM of the Region X pair would reach the threshold, the decision would be completed, and movement would ensue. At the same time and in contrast, the SMA pair would not integrate toward a decision (*Figure 8B*). We modeled this by not including any decision threshold in the SMA in deliberate decisions (i.e., the threshold was set to infinity, letting the DDM accumulate forever). (The corresponding magnitudes of the drift-rate and other parameters are detailed in the Materials and methods and *Table 1*.) So, when Region X activity reaches the threshold, the SMA (supposedly recorded using electrode $C_z$) will have happened to accumulate to some random level (*Figure 8B*). This entails that, when we align such SMA activity to decision (or movement) onset, we will find just a simple, weak linear trend in the SMA. Importantly, the RP is measured in electrode $C_z$ above the SMA. Hence, we search for it in

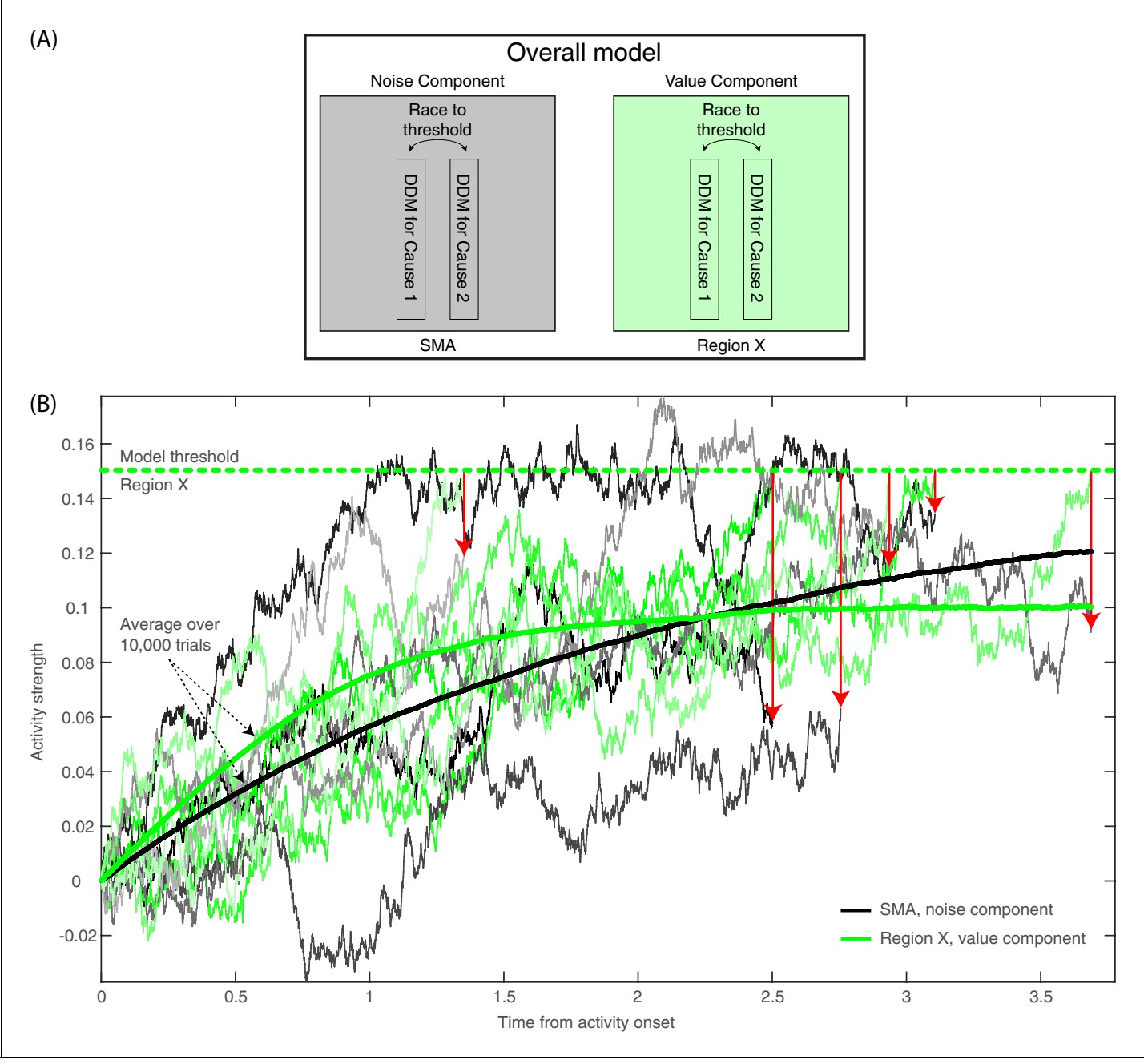

**Figure 8.** Model description and model runs in the SMA and in Region X. (**A**) A block diagram of the model, with its noise (SMA) and value (Region X) components, each instantiated as a race to threshold between a pair of DDMs (or causes—one congruent with the ratings in the first part of the experiment, the other incongruent). (**B**) A few runs of the model in the deliberate condition, in Region X (green colors), depicting the DDM for the congruent option. As is apparent, the DDM stops when the value-based component reaches threshold. Red arrows point from the Region X DDM trace at threshold to the corresponding time in the trace of the SMA (black and gray colors). The SMA traces integrate without a threshold (as the decision outcome is solely determined by the value component in Region X). The thick green and black lines depict average Region X and SMA activity, respectively, over 10,000 model runs locked to *stimulus onset*. Hence, we do not expect to find an RP in either brain region. (For decision-locked activity see *Figure 9B*).

DOI: https://doi.org/10.7554/eLife.39787.009

**Table 1.** Values of the model's parameters across decision types and decision difficulties.
Values of the drift-rate parameter, $I$, for the congruent and incongruent options; for the leak rate, $k$; and for the noise scaling factor, $c$. We fixed the threshold at the value of 0.3. The values in the table are for the component of the model where the decisions were made. Hence, they are for Region X in deliberate decisions and for the SMA in arbitrary ones. Note that, for deliberate decisions, drift-rate values in the SMA were 1.45 times smaller than the values in this table for each entry.

| Decision type | Decision difficulty | $I_{congruent}$ | $I_{incongruent}$ | $K$ | $C$ |
|---|---|---|---|---|---|
| Deliberate decisions (Region X values) | Easy | 0.23 | 0.06 | 0.52 | 0.08 |
| | Hard | 0.18 | 0.09 | 0.53 | 0.11 |
| Arbitrary decisions (SMA values) | Easy | 0.24 | 0.21 | 0.53 | 0.22 |
| | Hard | 0.22 | 0.20 | 0.54 | 0.23 |

DOI: https://doi.org/10.7554/eLife.39787.010

the SMA (or Noise) Component of our model (and not in Region X). The expected trend in the SMA is the one depicted in red in *Figure 9B* for the deliberate easy and hard conditions (here model activity was flipped vertically—from increasing above the $x$ axis to decreasing below it—as in *Schurger et al., 2012*). In arbitrary decisions, on the other hand, the SMA pair, from which we record, is also the one that determines the outcome. Hence, motion ensues whenever one of the DDMs crosses the threshold. Thus, when its activity is inspected with respect to movement onset, it forms the RP-like shape of *Figure 9B* (in blue), in line with the model by *Schurger et al. (2012)*. Note that the downward trend for deliberate hard trials is slightly smaller than for deliberate easy (*Figure 9B*). While the noise in the empirical EEG signals prohibits reliable statistical differences, the trend in the empirical data is interestingly in the same direction (see the last 500 ms before movement onset in *Figure 3A*).

Akin to the Schurger model, we simultaneously fit our DDMs to the complete distribution of our empirical reaction-times (RTs; *Figure 9A*) and to the empirical consistency scores (the proportions of congruent decisions; see Materials and methods). The models' fit to the empirical RTs and consistencies were good (RT and consistency errors for deliberate easy and hard were 0.054 and 0.004 and 0.166 and 0.013 respectively; for arbitrary easy and hard they were 0.053 and 0.002, and 0.055 and 0.003, respectively; *Figure 9A*) The averages of these empirical RT distributions were 2.13, 2.52, 0.98 and 1.00 s and the empirical consistency scores were 0.99, 0.83, 0.54 and 0.49 for deliberate easy, deliberate hard, arbitrary easy, and arbitrary hard, respectively.

Once we found the models with the best fit (see Materials and methods for details), we used those to predict the resulting ERP patterns in the SMA—that is, those we would expect to record in $C_z$. The ERP that the model predicted was the mean between the congruent and incongruent activities, as both would be picked up by $C_z$. The result was an RP-like activity for arbitrary decisions, but only a very slight slope for deliberate decisions (*Figure 9B*; both activities were flipped vertically, as in Schurger's model). This was in line with our empirical results (compare *Figure 3A*).

Note that the Schurger model aims to account for neural activity leading up to the decision to move, but no further (*Schurger et al., 2012*). Similarly, we expect our DDM to fit $C_z$ neural data only up to around −0.1 to −0.2 s (100 to 200 ms before empirical response onset). What is more, we model Region X activity here using a DDM for simplicity. But we would get a similar result—SMA RP-like activity for arbitrary decision and only a trend for deliberate ones—for other models of decision making, as long as the completion of deliberate decisions would remain statistically independent from threshold crossing in the DDMs of the SMA. Further, we make no claims that ours is the only, or even optimal, model that explains our results. Rather, by specifically extending the Schurger model, our goal was to show how that interpretation of the RP could also be applied to our more-complex paradigm. (We refer the reader to work by Schurger and colleagues [*Schurger, 2018*; *Schurger et al., 2012*] for more detailed discussions about the model, its comparison to other models, and the relation to conscious-decision completion).

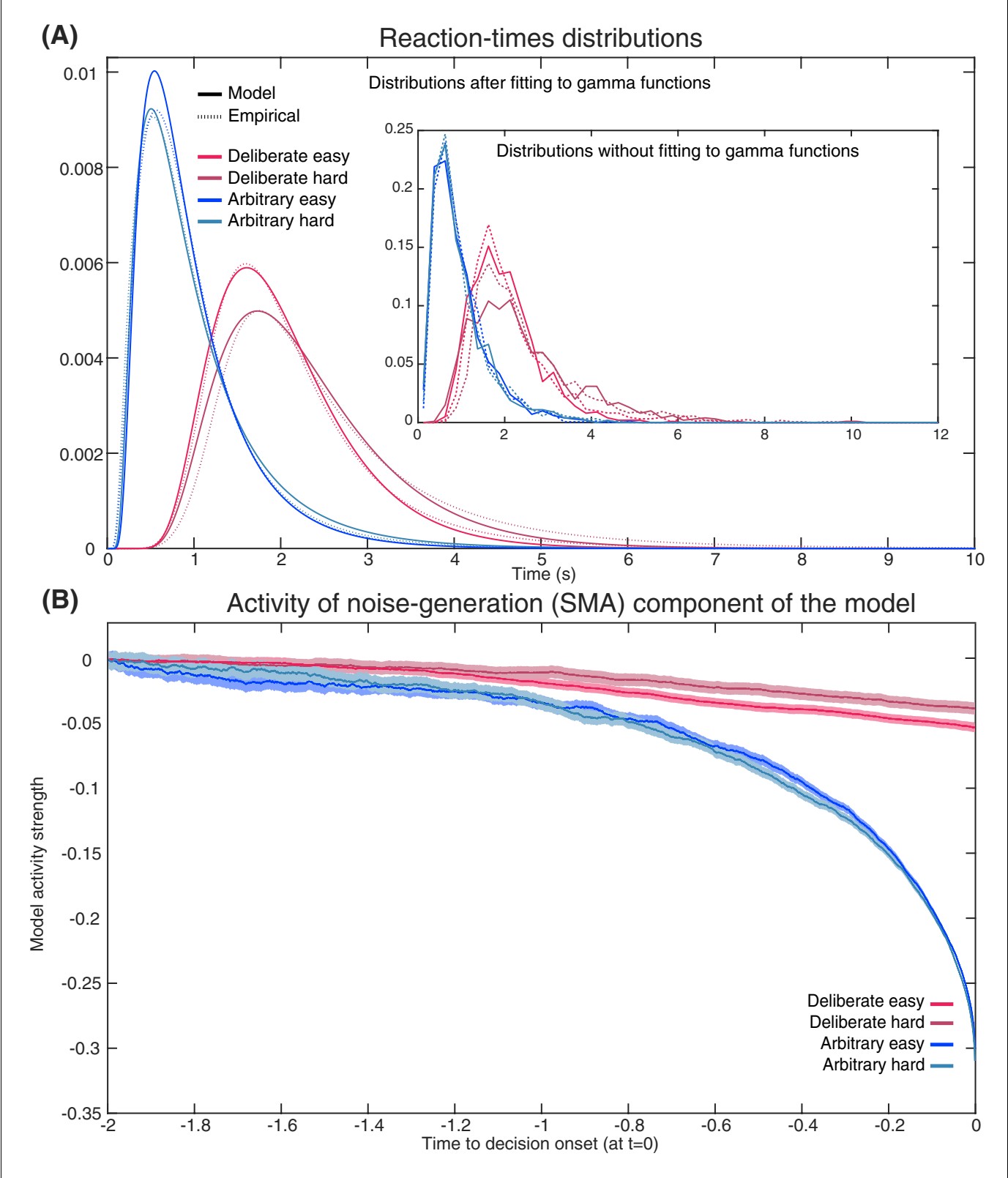

**Figure 9.** Empirical and model RTs and model prediction for C$_z$ activity. (**A**) The model (solid) and empirical (dashed) distributions of subjects' RTs. We present both the data as fitted with gamma functions to the cumulative distributions (see Materials and methods) across the four decision types in the main figure, and the original, non-fitted data in the inset. (**B**) The model's prediction for the ERP activity in its Noise Component (*Figure 8A*) in the SMA (electrode C$_z$), locked to decision completion (at t = 0 s), across all four decision types.

*Figure 9 continued on next page*

*Figure 9 continued*

DOI: https://doi.org/10.7554/eLife.39787.011

## Discussion

Since the publication of Libet's seminal work—which claimed that neural precursors of action, in the form of the RP, precede subjects' reports of having consciously decided to act (*Libet et al., 1983*)— a vigorous discussion has been raging among neuroscientists, philosophers, and other scholars about the meaning of the RP for the debate on free will (recent collections include *Mele, 2015*; *Pockett et al., 2009*; *Sinnott-Armstrong and Nadel, 2011*). Some claim that the RP is simply a marker for an unconscious decision to act and thus its onset at least reflects an intention to move and ballistically leads to movement (*Libet et al., 1983*; *Soon et al., 2008*). Under this interpretation, the onset of the RP before the reported completion of the conscious decision to move effectively removes conscious will from the causal chain leading to action (*Haggard, 2005*; *Haggard, 2008*; *Libet, 1985*; *Wegner, 2002*). Others do not agree (*Breitmeyer, 1985*; *Mele, 2009*; *Nahmias et al., 2014*; *Roskies, 2010*). But, regardless, the RP lies at the heart of much of this debate (*Kornhuber and Deecke, 1990*; *Libet et al., 1983*).

Notably, the RP and similar findings showing neural activations preceding the conscious decision to act have typically been based on arbitrary decisions (*Haggard and Eimer, 1999*; *Lau et al., 2004*; *Libet, 1985*; *Libet et al., 1983*; *Sirigu et al., 2004*; *Soon et al., 2008*; *Soon et al., 2013*). This, among other reasons, rested on the notion that for an action to be completely free, it should not be determined in any way by external factors (*Libet, 1985*)—which is the case for arbitrary, but not deliberate, decisions (for the latter, each decision alternative is associated with a value, and the values the of alternatives typically guide one's decision). But this notion of freedom faces several obstacles. First, most discussions of free will focus on deliberate decisions, asking when and whether these are free (*Frankfurt, 1971*; *Hobbes, 1994*; *Wolf, 1990*). This might be because everyday decisions to which we associate freedom of will—like choosing a more expensive but more environmentally friendly car, helping a friend instead of studying more for a test, donating to charity, and so on—are generally deliberate, in the sense of being reasoned, purposeful, and bearing consequences (although see *Deutschländer et al., 2017*). In particular, the free will debate is often considered in the context of moral responsibility (e.g., was the decision to harm another person free or not) (*Fischer, 1999*; *Haggard, 2008*; *Maoz and Yaffe, 2016*; *Roskies, 2012*; *Sinnott-Armstrong, 2014*; *Strawson, 1994*), and free will is even sometimes defined as the capacity that allows one to be morally responsible (*Mele, 2006*; *Mele, 2009*). In contrast, it seems meaningless to assign blame or praise to arbitrary decisions. Thus, though the scientific operationalization of free will has typically focused on arbitrary decisions, the common interpretations of these studies—in neuroscience and across the free will debate—have often alluded to deliberate decisions. This is based on the implicit assumption that the RP studies capture the same, or a sufficiently similar, process as that which occurs in deliberate decisions. And so, inferences from RP results on arbitrary decisions can be made to deliberate decisions.

However, here we show that this assumption may not be justified, as the neural precursors of arbitrary decisions, at least in the form of the RP, do not generalize to meaningful, deliberate decisions (*Breitmeyer, 1985*; *Roskies, 2010*). For *arbitrary decisions*, we replicated the waveform found in previous studies, where the subjects moved endogenously, at a time of their choice with no external cues (*Shibasaki and Hallett, 2006*). The RP was also similarly recorded in the $C_z$ electrode and with a typical scalp topography. However, the RP was altogether absent—or at least substantially diminished—for *deliberate decisions*; it showed neither the expected slope over time nor the expected scalp topography.

Null-hypothesis significance testing (NHST) suggested that the null hypothesis—that is, that there is no RP—can be rejected for arbitrary decisions but cannot be rejected for deliberate ones. A cluster-based nonparametric permutation analysis—to locate temporal windows where EEG activity is reliably different from 0—found prolonged activity of this type during the 1.2 s before movement onset for easy and hard arbitrary decisions, but no such activity was found for either type of deliberate decisions. A Bayesian analysis found clear evidence for an RP in arbitrary decisions and an inconclusive trend toward no RP in deliberate decisions. Changing the baseline to make it equally distant

from arbitrary and deliberate decisions did provide evidence for the absence of an RP in deliberate decisions, while still finding clear evidence for an RP in arbitrary decisions. Further, baseline-invariant trend analysis showed that there is no trend during the RP time window for deliberate decisions (here Bayesian analysis suggested moderate to strong evidence against a trend) while there existed a reliable trend for arbitrary decisions (and extremely strong evidence for an effect in the Bayesian framework). Thus, taken together, there is overwhelming evidence for an RP in arbitrary decisions (in all six different analyses that we conducted—NHST and Bayesian). But, in contrast, we found no evidence for the existence of an RP in deliberate decisions (in all six analyses) and, at the same time, there was evidence against RP existence in such decisions (in five of the six analyses, with the single, remaining analysis providing only inconclusive evidence for an absence of an RP). Therefore, when the above analyses are taken together, we think that the most plausible interpretation of our results is that the RP is absent in deliberate decisions.

Nevertheless, even if one takes our results to imply that the RP is only strongly diminished in deliberate compared to arbitrary decisions, this provides evidence against drawing strong conclusions regarding the free-will debate from the Libet and follow-up results. The assumption in the Libet-type studies is that the RP simply reflects motor preparation (*Haggard, 2019*; *Haggard and Eimer, 1999*; *Libet, 1985*; *Libet et al., 1983*; *Shibasaki and Hallett, 2006*) and in that it lives up to its name. However, in our paradigm, both the sensory inputs and the motor outputs were the same between arbitrary and deliberate trials. Thus, motor preparation is expected in both conditions and the RP should have been found in both. Accordingly, any consistent difference in the RP between the decision types suggests—at the very least—that it is a more complex signal than Libet and colleagues had assumed. For one, it shows that it is influenced by cognitive state and that it cannot be regarded as a genuine index of a voluntary decision, be it arbitrary or deliberate. Further, our model predicted an RP in arbitrary decisions but only a slow trend in movement-locked ERP during deliberate decisions that is in the same direction as the RP, *but is not an RP*. Hence, a signal that resembles a strongly diminished RP but is in fact just slow trend in the same direction as the RP is congruent with our model.

Interestingly, while the RP was present in arbitrary decisions but absent in deliberate ones, the LRP—a long-standing, more-motor ERP component, which began much later than the RP—was indistinguishable between the different decision types. This provides evidence that, at the motor level, the neural representation of the deliberate and arbitrary decisions that our subjects made may have been indistinguishable, as was our intention when designing the task.

Our findings and the model thus suggest that two different neural mechanisms may be involved in arbitrary and deliberate decisions. Earlier literature demonstrated that deliberate, reasoned decision-making—which was mostly studied in the field of neuroeconomics (*Kable and Glimcher, 2009*) or using perceptual decisions (*Gold and Shadlen, 2007*)—elicited activity in the prefrontal cortex (PFC; mainly the dorsolateral (DLPFC) part (*Sanfey et al., 2003*; *Wallis and Miller, 2003*) and ventromedial (VMPFC) part/orbitofrontal cortex (OFC) (*Ramnani and Owen, 2004*; *Wallis, 2007*) and the anterior cingulate cortex (ACC) (*Bush et al., 2000*; *Carter et al., 1998*). Arbitrary, meaningless decisions, in contrast, were mainly probed using variants of the Libet paradigm, showing activations in the Supplementary Motor Area (SMA), alongside other frontal areas like the medial frontal cortex (*Brass and Haggard, 2008*; *Krieghoff et al., 2011*) or the frontopolar cortex, as well as the posterior cingulate cortex (*Fried et al., 2011*; *Soon et al., 2008*) (though see *Hughes et al., 2011*, which suggests that a common mechanism may underlie both decision types). Possibly then, arbitrary and deliberate decisions may differ not only with respect to the RP, but be subserved by different underlying neural circuits, which makes generalization from one class of decisions to the other more difficult. Deliberate decisions are associated with more lateralized and central neural activity while arbitrary ones are associated with more medial and frontal ones. This appears to align with the different brain regions associated with the two decision types above, as also evidenced by the differences we found between the scalp distributions of arbitrary and deliberate decisions (*Figure 3A*). Further studies are needed to explore this potential divergence in the neural regions between the two decision types.

Therefore, at the very least, our results support the claim that the previous findings regarding the RP should be confined to arbitrary decisions and do not generalize to deliberate ones. What is more, if the ubiquitous RP does not generalize, it cannot simply be assumed that other markers will. Hence, such differences clearly challenge the generalizability of previous studies focusing on

arbitrary decisions to deliberate ones, regardless of whether they were based on the RP or not. In other words, our results put the onus on attempts to generalize markers of upcoming action from arbitrary to deliberate decisions; it is on them now to demonstrate that those markers do indeed generalize. And, given the extent of the claims made and conclusions derived based on the RP in the neuroscience of free will (see again *Mele, 2015*; *Pockett et al., 2009*; *Sinnott-Armstrong and Nadel, 2011*), our findings call for a re-examination of some of the basic tenets of the field.

It should be noted that our study does not provide positive evidence that consciousness is more involved in deliberate decisions than in arbitrary ones; such a strong claim requires further evidence, perhaps from future research. But our results highlight the need for such research. Under some (strong) assumptions, the onset of the RP before the onset of reported intentions to move may point to there being no role for consciousness in arbitrary decisions. But, even if such conclusions can be reached, they cannot be safely extended to deliberate decisions.

To be clear, and following the above, we do not claim that the RP captures all unconscious processes that precede conscious awareness. However, some have suggested that the RP represents unconscious motor-preparatory activity before any kind of decision (e.g., *Libet, 1985*). But our results provide evidence against that claim, as we do not find an RP before deliberate decisions, which also entail motor preparation. Furthermore, in deliberate decisions in particular, it is likely that there are neural precursors of upcoming actions—possibly involving the above neural circuits as well as circuits that represents values—which are unrelated to the RP (the lack of such precursors is not merely implausible; it implies dualism: *Mudrik and Maoz, 2015*; *Wood, 1985*).

Note also that we did not attempt to clock subjects' conscious decision to move. Rather, we instructed them to hold their hands above the relevant keyboard keys and press their selected key as soon as they made up their mind. This was to keep the decisions in this task more ecological and because we think that the key method of measuring decision completion (using some type of clock to measure Libet's W-time) is highly problematic (see Materials and methods). But, even more importantly, clock monitoring was demonstrated to have an effect on RP size (*Miller et al., 2011*), so it could potentially confound our results (*Maoz et al., 2015*).

Some might also claim that unconscious decision-making could explain our results, suggesting that in arbitrary decisions subjects engage in unconscious deliberation or in actively inhibiting their urge to follow their preference as well as in free choice, while in deliberate decisions only deliberation is required. But this interpretation is unlikely because the longer RTs in deliberate decisions suggest, if anything, that more complex mental processes (conscious or unconscious) took place before deliberate and not arbitrary decisions. In addition, these interpretations should impede our chances of finding the RP in arbitrary trials (as the design diverges from the original Libet task), yet the RP was present, rendering them less plausible.

Aside from highlighting the neural differences between arbitrary and deliberate decisions, this study also challenges a common interpretation of the function of the RP. If the RP is not present before deliberate action, it does not seem to be a necessary link in the general causal chain leading to action. *Schurger et al. (2012)* suggested that the RP reflects the accumulation of autocorrelated, stochastic fluctuations in neural activity that lead to action, following a threshold crossing, when humans arbitrarily decide to move. According to that model, the shape of the RP results from the manner in which it is computed from autocorrelated EEG: averaged over trials that are locked to response onset (that directly follows the threshold crossing). Our results and our model are in line with that interpretation and expand it to decisions that include both when and which hand to move. They suggest that the RP represents the accumulation of noisy, random fluctuations that drive arbitrary decisions, whereas deliberate decisions are mainly driven by the values associated with the decision alternatives (*Maoz et al., 2013*).

Our drift-diffusion model was based on the assumption that every decision can be driven by a component reflecting the values of the decision alternatives (i.e., subjects' support for the two NPOs we presented) or by another component representing noise—random fluctuations in neural activity. The value component plays little to no role in arbitrary decisions, so action selection and timing depend on when the accumulation of noise crosses the decision threshold for the congruent and incongruent decision alternatives. In deliberate decisions, in contrast, the value component drives the decisions, while the noise component plays little to no role. Thus, in arbitrary decisions, action onset closely tracks threshold crossings of the noise (or SMA) component. But, in deliberate decisions, the noise component reaches a random level and is then stopped; so, the value component

drives the decision. Hence, as we record from the SMA (the noise component)—locking the ERP to response onset and averaging over trials to obtain the RP leads to a slight slope for deliberate decisions but to the expected RP shape in arbitrary decisions. This provides evidence that the RP does not reflect subconscious movement preparation. Rather, it is induced by threshold crossing of stochastic fluctuations in arbitrary decisions, which do not drive deliberate decisions; accordingly, the RP is not found in deliberate decisions. Our model therefore challenges RP-based claims against free will in both arbitrary and deliberate decisions. Further studies of the causal role of consciousness in deliberate versus arbitrary decisions are required to test this claim.

Nevertheless, two possible, alternative explanations of our results can be raised. First, one could claim that—in the deliberate condition only—the NPO names act as a cue, thereby creating a stimulus-response mapping and in that turning what we term internal, deliberate decisions into no more than simple responses to external stimuli. Under this account, if the preferred NPO is on the right, it is immediately interpreted as 'press right'. It would therefore follow that subjects are actually not making decisions in deliberate trials, which in turn is reflected by the absence of the RP in those trials. However, the reaction time and consistency results that we obtained for deliberate trials provide evidence against this interpretation. We found longer reaction times for hard-deliberate decisions than for easy-deliberate ones (2.52 versus 2.13 s, on average, respectively; *Figure 2* left) and higher consistencies with the initial ratings for easy-deliberate decisions than for hard-deliberate decisions (0.99 versus 0.83, on average, respectively; *Figure 2* right). If the NPO names acted as mere cues, we would have expected no differences between reaction times or consistencies for easy- and hard-deliberate decisions. In addition, there were 50 different causes in the first part of the experiment. So, it is highly unlikely that subjects could memorize all 1225 pairwise preferences among these causes and simply transform any decision between a pair of causes into a stimulus instructing to press left or right.

Another alternative interpretation of our results is that subjects engage in (unconscious) deliberation also during arbitrary decisions (*Tusche et al., 2010*), as they are trying to find a way to break the symmetry between the two possible actions. If so, the RP in the arbitrary decisions might actually reflect the extra effort in those types of decisions, which is not found in deliberate decisions. However, this interpretation entails a longer reaction time for arbitrary than for deliberate decisions, because of the heavier cognitive load, which is the opposite of what we found (*Figure 2A*).

In conclusion, our study suggests that RPs do not precede deliberate decisions (or at the very least are strongly diminished before such decisions). In addition, it suggests that RPs represent an artificial accumulation of random fluctuations rather than serving as a genuine marker of an unconscious decision to initiate voluntary movement. Hence, our results challenge RP-based claims of Libet and follow-up literature against free will in arbitrary decisions and much more so the generalization of these claims to deliberate decisions. The neural differences we found between arbitrary and deliberate decisions as well as our model further put the onus on any study trying to draw conclusions about the free-will debate from arbitrary decisions to demonstrate that these conclusions generalize to deliberate ones. This motivates future investigations into other precursors of action besides the RP using EEG, fMRI, or other techniques. It also highlights that it would be of particular interest to find the neural activity that precedes deliberate decisions as well as neural activity, which is not motor activity, that is common to both deliberate and arbitrary decisions.

## Materials and methods

### Subjects

Twenty healthy subjects participated in the study. They were California Institute of Technology (Caltech) students as well as members of the Pasadena community. All subjects had reported normal or corrected-to-normal sight and no psychiatric or neurological history. They volunteered to participate in the study for payment ($20 per hour). Subjects were prescreened to include only participants who were socially involved and active in the community (based on the strength of their support of social causes, past volunteer work, past donations to social causes, and tendency to vote). The data from 18 subjects was analyzed; two subjects were excluded from our analysis (see *Sample size and exclusion criteria* below). The experiment was approved by Caltech's Institutional Review Board (14–0432;

Neural markers of deliberate and random decisions), and informed consent was obtained from all participants after the experimental procedures were explained to them.

## Sample size and exclusion criteria

We ran a power analysis based on the findings of *Haggard and Eimer (1999)*. Their RP in a free left/right-choice task had a mean of 5.293 µV and standard deviation of 2.267 µV. Data from a pilot study we ran before this experiment suggested that we might obtain smaller RP values in our task (they referenced to the tip of the nose and we to the average of all channels, which typically results in a smaller RP). Therefore, we conservatively estimated the magnitude of our RP as half of that of Haggard and Eimer, 2.647 µV, while keeping the standard deviation the same at 2.267 µV. Our power analysis therefore suggested that we would need at least 16 subjects to reliably find a difference between an RP and a null RP (0 µV) at a p-value of 0.05 and power of 0.99. This number agreed with our pilot study, where we found that a sample size of at least 16 subjects resulted in a clear, averaged RP. Following the above reasoning, we decided beforehand to collect 20 subjects for this study, taking into account that some could be excluded as they would not meet the following predefined inclusion criteria: at least 30 trials per experimental condition remaining after artifact rejection; and averaged RTs (across conditions) that deviated by less than three standard deviations from the group mean.

Subjects were informed about the overall number of subjects that would participate in the experiment when the NPO lottery was explained to them (see below). So, we had to finalize the overall number of subjects who would participate in the study—but not necessarily the overall number of subjects whose data would be part of the analysis—before the experiment began. After completing data collection, we ran only the EEG preprocessing and behavioral-data analysis to test each subject against the exclusion criteria. This was done before we looked at the data with respect to our hypothesis or research question. Two subjects did not meet the inclusion criteria: the data of one subject (#18) suffered from poor signal quality, resulting in less than 30 trials remaining after artifact rejection; another subject (#12) had RTs longer than three standard deviations from the mean. All analyses were thus run on the 18 remaining subjects.

## Stimuli and apparatus

Subjects sat in a dimly lit room. The stimuli were presented on a 21' Viewsonic G225f (20' viewable) CRT monitor with a 60 Hz refresh rate and a 1024 × 768 resolution using Psychtoolbox version three and Mathworks Matlab 2014b (*Brainard, 1997*; *Pelli, 1997*). They appeared with a gray background (RGB values: [128, 128,128]). The screen was located 60 cm away from subjects' eyes. Stimuli included names of 50 real, non-profit organizations (NPOs). Twenty organizations were consensual (e.g., the Cancer Research Institute, or the Hunger project), and thirty were more controversial: we chose 15 causes that were widely debated (e.g., pro/anti guns, pro/anti abortions), and selected one NPO that supported each of the two sides of the debate. This was done to achieve variability in subjects' willingness to donate to the different NPOs. In the main part of the experiment, succinct descriptions of the causes (e.g., pro-marijuana legalization, pro-child protection; for a full list of NPOs and causes see *Supplementary file 1*) were presented in black Comic Sans MS.

## Study design

The objective of this study was to compare ERPs elicited by arbitrary and deliberate decision-making, and in particular the RP. We further manipulated decision difficulty to validate our manipulation of decisions type: we introduced hard and easy decisions which corresponded to small and large differences between subjects' preferences for the pairs of presented NPOs, respectively. We reasoned that if the manipulation of decision type (arbitrary vs. deliberate) was effective, there would be behavioral differences between easy and hard decisions for deliberate choices but not for arbitrary choices (because differences in preferences should not influence subjects' arbitrary decisions). Our 2 × 2 design was therefore decision type (arbitrary vs. deliberate) by decision difficulty (easy vs. hard). Each condition included 90 trials, separated into 10 blocks of 9 trials each, resulting in a total of 360 trials and 40 blocks. Blocks of different decision types were randomly intermixed. Decision difficulty was randomly counterbalanced across trials within each block.

## Experimental procedure

In the first part of the experiment, subjects were presented with each of the 50 NPOs and the causes with which the NPOs were associated separately (see *Supplementary file 1*). They were instructed to rate how much they would like to support that NPO with a $1000 donation on a scale of 1 ('I would not like to support this NPO at all) to 7 ('I would very much like to support this NPO'). No time pressure was put on the subjects, and they were given access to the website of each NPO to give them the opportunity to learn more about the NPO and the cause it supports.

After the subjects finished rating all NPOs, the main experiment began. In each block of the experiment, subjects made either deliberate or arbitrary decisions. Two succinct cause descriptions, representing two actual NPOs, were presented in each trial (*Figure 1*). In deliberate blocks, subjects were instructed to choose the NPO to which they would like to donate $1000 by pressing the <Q> or <P> key on the keyboard, using their left and right index finger, for the NPO on the left or right, respectively, as soon as they decided. Subjects were informed that at the end of each block one of the NPOs they chose would be randomly selected to advance to a lottery. Then, at the end of the experiment, the lottery will take place and the winning NPO will receive a $20 donation. In addition, that NPO will advance to the final, inter-subject lottery, where one subject's NPO will be picked randomly for a $1000 donation. It was stressed that the donations were real and that no deception was used in the experiment. To persuade the subjects that the donations were real, we presented a signed commitment to donate the money, and promised to send them the donation receipts after the experiment. Thus, subjects knew that in deliberate trials, every choice they made was not hypothetical, and could potentially lead to an actual $1020 donation to their chosen NPO.

Arbitrary trials were identical to deliberate trials except for the following crucial differences. Subjects were told that, at the end of each block, the pair of NPOs in one randomly selected trial would advance to the lottery together. And, if that pair wins the lottery, both NPOs would receive $10 (each). Further, the NPO pair that would win the inter-subject lottery would receive a $500 donation each. Hence it was stressed to the subjects that there was no reason for them to prefer one NPO over the other in arbitrary blocks, as both NPOs would receive the same donation regardless of their button press. Subjects were told to therefore simply press either <Q> or <P> as soon as they decided to do so.

Thus, while subjects' decisions in the deliberate blocks were meaningful and consequential, their decisions in the arbitrary blocks had no impact on the final donations that were made. In these trials, subjects were further urged not to let their preferred NPO dictate their response. Importantly, despite the difference in decision type between deliberate and arbitrary blocks, the instructions for carrying out the decisions were identical: Subjects were instructed to report their decisions (with a key press) as soon as they made them in both conditions. They were further asked to place their left and right index fingers on the response keys, so they could respond as quickly as possible. Note that we did not ask subjects to report their 'W-time' (time of consciously reaching a decision), because this measure was shown to rely on neural processes occurring after response onset (*Lau et al., 2007*) and to potentially be backward inferred from movement time (*Banks and Isham, 2009*). Even more importantly, clock monitoring was demonstrated to have an effect on RP size (*Miller et al., 2011*), so it could potentially confound our results (*Maoz et al., 2015*).

Decision difficulty (Easy/Hard) was manipulated throughout the experiment, randomly intermixed within each block. Decision difficulty was determined based on the rating difference between the two presented NPOs. NPO pairs with one or at least four rating-point difference were designated hard or easy, respectively. Based on each subject's ratings, we created a list of NPO pairs, half of each were easy choices and the other half hard choices.

Each block started with an instruction written either in dark orange (Deliberate: 'In this block choose the cause to which you want to donate $1000') or in blue (Arbitrary: 'In this block both causes may each get a $500 donation regardless of the choice') on a gray background that was used throughout the experiment. Short-hand instructions appeared at the top of the screen throughout the block in the same colors as that block's initial instructions; Deliberate: 'Choose for $1000' or Arbitrary: 'Press for $500 each' (*Figure 1*).

Each trial started with the gray screen that was blank except for a centered, black fixation cross. The fixation screen was on for a duration drawn from a uniform distribution between 1 and 1.5 s. Then, the two causes appeared on the left and right side of the fixation cross (left/right assignments

were randomly counterbalanced) and remained on the screen until the subjects reported their decisions with a key press—<Q> or <P> on the keyboard for the cause on the left or right, respectively. The cause corresponding to the pressed button then turned white for 1 s, and a new trial started immediately. If subjects did not respond within 20 s, they received an error message and were informed that, if this trial would be selected for the lottery, no NPO would receive a donation. However, this did not happen for any subject on any trial.

To assess the consistency of subjects' decisions during the main experiment with their ratings in the first part of the experiment, subjects' choices were coded in the following way: each binary choice in the main experiment was given a consistency grade of 1, if subjects chose the NPO that was rated higher in the rating session, and 0 if not. Then an averaged consistency grade for each subject was calculated as the mean consistency grade over all the choices. Thus, a consistency grade of 1 indicates perfect consistency with one's ratings across all trials, 0 is perfect inconsistency, and 0.5 is chance performance.

We wanted to make sure subjects were carefully reading and remembering the causes also during the arbitrary trials. This was part of an effort to better equate—as much as possible, given the inherent difference between the conditions—memory load, attention, and other cognitive aspects between deliberate and arbitrary decisions—except those aspects directly associated with the decision type, which was the focus of our investigation. We therefore randomly interspersed 36 memory catch-trials throughout the experiment (thus more than one catch trial could occur per block). On such trials, four succinct descriptions of causes were presented, and subjects had to select the one that appeared in the previous trial. A correct or incorrect response added or subtracted 50 cents from their total, respectively. (Subjects were informed that, if they reached a negative balance, no money will be deducted off their payment for participation in the experiment.) Thus, subjects could earn $18 more for the experiment, if they answered all memory test questions correctly. Subjects typically did well on these memory questions, on average erring in 2.5 out of 36 memory catch trials (7% error) and gaining additional $16.75 (SD = 3.19). Subjects' error rates in the memory task did not differ significantly between the experimental conditions (2-way ANOVA; decision type: $F_{(1,17)}$ =2.51, p=0.13; decision difficulty: $F_{(1,17)}$=2.62, p=0.12; interaction: $F_{(1,17)}$=0.84, p=0.37).

## ERP recording methods

The EEG was recorded using an Active two system (BioSemi, the Netherlands) from 64 electrodes distributed based on the extended 10–20 system and connected to a cap, and seven external electrodes. Four of the external electrodes recorded the EOG: two located at the outer canthi of the right and left eyes and two above and below the center of the right eye. Two external electrodes were located on the mastoids, and one electrode was placed on the tip of the nose. All electrodes were referenced during recording to a common-mode signal (CMS) electrode between POz and PO3. The EEG was continuously sampled at 512 Hz and stored for offline analysis.

## ERP analysis

ERP analysis was conducted using the 'Brain Vision Analyzer' software (Brain Products, Germany) and in-house Mathworks Matlab scripts. Data from all channels were referenced offline to the average of all channels (excluding external electrodes), which is known to result in a reduced-amplitude RP (because the RP is such a spatially diffuse signal). The data were then digitally high-pass filtered at 0.1 Hz using a Finite Impulse Response (FIR) filter to remove slow drifts. A notch filter at 59–61 Hz was applied to the data to remove 60 Hz electrical noise. The signal was then cleaned of blink and saccade artifacts using Independent Component Analysis (ICA) (*Junghöfer et al., 2000*). Signal artifacts were detected as amplitudes exceeding ± 100 µV, differences beyond 100 µV within a 200 ms interval, or activity below 0.5 µV for over 100 ms (the last condition was never found). Sections of EEG data that included such artifacts in any channel were removed (150 ms before and after the artifact). We further excluded single trials in which subjects pressed a button that was not one of the two designated response buttons (<Q> or <P> ) as well as trials where subjects' RTs were less than 200 ms, more than 10 s, or more than three standard deviations away from that subject's mean in that condition (mean number of excluded trials = 7.17, SD = 2.46, which are 1.99% of the trials). Overall, the average number of included trials in each experimental cell was 70.38 trials with a range of 36–86 out of 90 trials per condition. Channels that consistently had artifacts were replaced using

interpolation (0–6 channels, 1.95 channels per subject, on average). No significant differences were found in the number of excluded trials across conditions (2-way ANOVA; decision type: $F_{(1,17)}$ =3.31, p=0.09; decision difficulty: $F_{(1,17)}$=1.83, p=0.19; interaction: $F_{(1,17)}$=0.42, p=0.53).

The EEG was segmented by locking the waveforms to subjects' movement onset, starting 2 s prior to the movement and ending 0.2 s afterwards, with the segments averaged separately for each decision type (Deliberate/Arbitrary x Easy/Hard) and decision content (right/left hand). The baseline period was defined as the time window between −1000 ms and −500 ms prior to *stimulus* onset, that is, the onset of the causes screen, rather than prior to movement onset. In addition to the main baseline, we tested another baseline—from −1000 ms to −500 ms relative to *movement* onset—to investigate whether the baseline period influenced our main results (see Results). Furthermore, we segmented the EEG based on *stimulus* onset, using the same baseline, for stimulus-locked analysis (again, see Results).

To assess potential effects of eye movements during the experiment, we defined the radial eye signal as the average over all 4 EOG channels, when band-pass filtered to between 30 and 100 Hz. We then defined a saccade as any signal that was more than 2.5 standardized IQRs away from the median of the radial signal for more than 2 ms. Two consecutive saccades had to be at least 50 ms apart. The saccade count (SC) was the number of saccades during the last 500 ms before response onset (*Keren et al., 2010*; see also *Croft and Barry, 2000*; *Elbert et al., 1985*; *Shan et al., 1995*).

## Statistical analysis

EEG differences greater than expected by chance were assessed using two-way ANOVAs with decision type (deliberate, arbitrary) and decision difficulty (easy, hard), using IBM SPSS statistics, version 24. For both RP and LRP signals, the mean amplitude from 500 ms before to button-press onset were used for the ANOVAs. Greenhouse–Geisser correction was never required as sphericity was never violated (*Picton et al., 2000*).

Trend analysis on all subjects' data was carried out by regressing the voltage for every subject against time for the last 1000 ms before response onset using first-order polynomial linear regression (see Results). We first added a 25 Hz low-pass filter for anti-aliasing and then used every 10[th] time sample for the regression (i.e., the 1[st], 11[th], 21[st], 31[st] samples, and so on) to conform with the individual-subject analysis (see below). For the individual-subject analysis, the voltage on all trials was regressed against time in the same manner (i.e., for the last 1000 ms before response onset and using first-order polynomial linear regression). As individual-trial data is much noisier than the mean over all trials in each subject, we opted for standard robust-regression using iteratively reweighted least squares (implemented using the `robustfit()` function in Mathworks Matlab). The iterative robust-regression procedure is time consuming. So, we used every 10[th] time sample instead of every sample to make the procedure's run time manageable. Also, as EEG signals have a 1/f power spectrum, taking every 10[th] sample further better conforms with the assumption of i.i.d. noise in linear regression.

Furthermore, we conducted Bayesian analyses of our main results. This allowed us to assess the strength of the evidence for or against the existence of an effect, and specifically test whether null results stem from genuine absence of an effect or from insufficient or underpowered data. Specifically, the Bayes factor allowed us to compare the probability of observing the data given $H_0$ (i.e., no RP in deliberate decisions) against the probability of observing the data given $H_1$ (i.e., RP exists in deliberate decisions). We followed the convention that BF <0.33 implies substantial evidence for lack of an effect (that is, the data is at least three times more likely to be observed given $H_0$ than given $H_1$), 0.33 < BF < 3 suggests insensitivity of the data, and BF >3 denotes substantial evidence for the presence of an effect ($H_1$) (*Jeffreys, 1998*). Bayesian analysis was carried out using JASP (ver. 0.8; default settings).

In addition to the above, we used the cluster-based nonparametric method developed by Maris and Oostenveld to find continuous temporal windows where EEG activity was reliably different from 0 (*Maris and Oostenveld, 2007*). We used an in-house implementation of the method in Mathworks Matlab with a threshold of 2 on the *t* statistic and with a significance level of p=0.05.

## Model and simulations

All simulations were performed using Mathworks Matlab 2018b. The model was devised off the one proposed by *Schurger et al. (2012)*. Like them, we built a drift-diffusion model (DDM) (*Ratcliff, 1978*; *Usher and McClelland, 2001*), which included a leaky stochastic accumulator (with a threshold on its output) and a time-locking/epoching procedure. The original model amounted to iterative numerical integration of the differential equation

$$\delta x_i = (I - kx_i)\Delta t + c\xi_i\sqrt{\Delta t} \tag{1}$$

where *I* is the drift rate, *k* is the leak (exponential decay in *x*), $\xi$ is Gaussian noise, and *c* is a noise-scaling factor. $\Delta t$ is the discrete time step used in the simulation (we used $\Delta t = 0.001$). The model integrates $x_i$ over its iterations until it crosses a threshold, which represents a decision having been made.

In such drift-diffusion models, for a given *k* and *c*, the values of *I* and the threshold together determine how quickly a decision will be reached, on average. If we further fix the threshold, a higher drift rate, *I*, represents a faster decision, on average. The drift rate alone can thus be viewed as a constant 'urgency to respond' (using the original Schurger term) that is inherent in the demand characteristics of the task, evidenced by the fact that no subject took more than 20 s to make a decision on any trial. The leak term, *k*, ensures that the model would not be too linear; that is, it prevented the drift rate from setting up a linear trajectory for the accumulator toward the threshold. Also, *k* has a negative sign and is multiplied by $x_i$. So, $kx_i$ acts against the drift induced by *I* and gets stronger as $x_i$ grows. Hence, due to the leak term, doubling the height of the threshold could make the accumulator rarely reach the threshold instead of reaching it in roughly twice the amount of time (up to the noise term).

When comparing the model's activity for the SMA and Region X, we needed to know how to set the drift rates for the DDM in the two regions for deliberate decisions. We made the assumption that the ratio between the drift rate in Region X and in the SMA during deliberate decisions would be the same as the ratio between the average empirical activity in the SMA and in the rest of the brain during arbitrary decisions. Our EEG data suggested that this ratio (calculated as activity in $C_z$ divided by the mean activity in the rest of the electrodes) is 1.45. Hence, we set the drift rates in the SMA to be 1.45 times smaller than those of Region X for deliberate decisions (see *Table 1* for all parameter values).

Our model differed from Schurger's in two main ways. First, it accounted for both arbitrary and deliberate decisions and was thus built to fit our paradigm and empirical results. We devised a model that was composed of two distinct components (*Figure 8A*), each a race to threshold between 2 DDMs based on *Equation 1* (see below), but with different parameter values for each DDM (*Table 1*). The first component accumulated activity that drove arbitrary decisions (i.e., stochastic fluctuations [*Schurger et al., 2012*]). Such model activation reflects neural activity that might be recorded over the SMA by the $C_z$ electrode. We term this component of the model the *Noise* or *SMA* component. The second component of the model reflects brain activity that drives deliberate decisions, based on the values that subjects associated with the decision alternatives. We term this second component the *Value* or *Region X* component. Our model relied on its noise component to reflect arbitrary decisions and on its value component to reflect deliberate decisions.

A second difference between our model and Schurger and colleagues' is that theirs modeled only the decision *when* to move (during arbitrary decisions), as those were the only decisions that their subjects faced. But our subjects decided both *when* and *which hand* to move. So, we had to extend the Schurger model in that respect as well. We did this using a race-to-threshold mechanism between the decision alternatives. In our empirical paradigm, the difference in rating of the two causes was either 1 (for hard decisions) or 4–6 (for easy decisions; see 'Experimental Procedure' in Materials and methods), so there was always an alternative that was ranked higher than the other. Choosing the higher- or lower-ranked alternative was termed a congruent or incongruent choice with respect to the initial ratings, respectively. Hence, we modeled each decision the subjects made as a race to threshold between the DDMs for the congruent and incongruent alternatives in the noise component (for arbitrary decisions) or value component (for deliberate ones).

We found the reaction time (RT) distribution and consistency score for each subject (as detailed above). The model's RT was defined as the overall time that it took from the onset of the simulation

until the first threshold crossing in the race-to-threshold pair (between the congruent and incongruent DDM; for $\Delta t$ = 0.001 s). To determine the RT error, we ran a similar procedure to *Schurger et al. (2012)*. We computed the empirical, cumulative RT distribution for each subject and fit a gamma function to it. We then averaged those gamma-fitted distributions across all subjects and designated that the empirical distribution. Then, we computed the RT for the model for each parameter set from 1000 model runs and fitted a gamma function to that too. Finally, we computed the ratio of the intersection of the two cumulative distributions and their union; and the RT error was defined as one minus that ratio. The consistency error was computed as the absolute difference between the empirical and model consistencies. Finally, the overall error was defined as mean of these two errors.

To find the parameter values that minimize this overall error, we ran an exhaustive grid search over all the parameters. To do this, we fixed the threshold at 0.3 and found the values of the congruent and incongruent drift rates as well as the leak and the noise scaling factors that, at the same time, best fit the empirical distribution of the RTs and the consistency scores (with equal weights) for (easy, hard) x (deliberate, arbitrary) decisions (*Table 1*) in the manner described below. We had four parameters to fit: the congruent drift rate ($I_{congruent}$), the incongruent drift rate ($I_{incongruent}$), the leak ($k$), and the noise-scaling factor ($c$). We first explored the 4D space of those four parameters to test how smooth it was and to find the range in which to run the exhaustive grid search. We found that $I_{congruent}$ was minimized between 0.05 and 0.4 and that $I_{incongruent}$ was minimized somewhere between the value of the $I_{congruent}$ and five times smaller than that. Then $k$ was found to lie between 0.2 and 0.55, and $c$ between 0.01 and 0.3. Based on our tests and specifically the smoothness of the search space, we found that we could divide each parameter range into five equal parts. So, we then tested the model in each of the 5 × 5×5 × 5 (=625) entries in the grid by running the model 1000 times per entry and computing its error by comparing its RT distribution and consistency rate to the empirical ones. Once we found the minimum point in that 4D grid, we zoomed in on the range between the grid value to the left of the entry where the minimum error was found and the one to the right of that minimum. (We chose the initial range such that the minimum error was never achieved on the smallest or largest point for all parameters.) We continued this process of finding the minimum and zooming in until each parameter reached a range that was smaller than 0.025 or the error was less than 0.025.

Once we found the parameters that minimized the mismatch between the empirical and model RTs and consistencies for each decision type, we ran the model with those parameters 1000 times again. For each run of the model, we identified the first threshold crossing point (for the congruent or incongruent DDM), which made that model the 'winning' DDM. We then extracted the activity in both the winning and losing DDM in the last 2 s (2000 steps) before the crossing. And we averaged across the activity of the two DDMs (because an electrode over the SMA or Region X would be expected to pick up signals from both DDMs). If the first crossing was earlier than sample no. 2000 by $n > 0$ samples, we padded the beginning of the epoch with $n$ null values (NaN or 'not-a-number' in Matlab). Finally, we averaged across all 1000 model runs to calculate the model's RP. Note that NaN values did not contribute to the average across simulated trials, so the simulated average RP became noisier at earlier time points in the epoch. Hence, our model was similarly limited to the Schurger model in its inability to account for activity earlier than the beginning of the trial (see Results).

Our model describes the mechanism we propose for subjects' decisions. We therefore now examine how the values in *Table 1* relate to the experimental design and actual results. As discussed above, we view $I_{congruent}$ and $I_{incongruent}$ as reflecting the values of the decision alternatives that were rated higher and lower in the first part of the experiment, respectively. We thus expect $I_{congruent}$ and $I_{incongruent}$ to be more different (similar) for easy (hard) deliberate decisions. And we indeed find an almost 4-fold differences between $I_{congruent}$ and $I_{incongruent}$ in deliberate easy versus only 2-fold difference for deliberate hard. For arbitrary decisions, be they easy or hard, we do not expect the values of the decision alternatives to matter much. And this is reflected in the similar values of $I_{congruent}$ and $I_{incongruent}$ there.

The leak values ($k$) were similar across all conditions. In contrast, the values of the noise-scaling factor ($c$) for deliberate decisions were only 35% to 45% of those in arbitrary decisions. As decision types were blocked, this appears to indicate that subjects adopted different decision parameters across blocks. One reason for the larger levels of noise in arbitrary trials might be the difference in

consistency scores. The noise-scaling factor provides the randomness to our model. Hence, the lower the noise factor, the more deterministic the result of the model's simulation—and thus the higher the chance that the consistent DDM (i.e., with $I_{congruent}$, which is larger than $I_{incongruent}$) would win the race against the inconsistent DDM. The consistency scores for deliberate easy, deliberate hard, arbitrary easy, and arbitrary hard, were 0.99, 0.83, 0.54 and 0.49, respectively. And these are nicely anti-correlated with the respective increase in the noise levels: 0.08, 0.11, 0.22, and 0.23 ($r = -0.99$, $p = 0.007$).

It should be noted that, when minimizing the error in the deliberate-hard condition, we found two disjoint regions within the parameter space ($I_{congruent}$, $I_{incongruent}$, $k$, $c$) where the error reached relatively similar local minima. The parameter values corresponding to the smaller among the two errors are listed in *Table 1* (deliberate hard row). The parameter values at the other local minimum are ($I_{congruent}$, $I_{incongruent}$, $k$, $c$) = (0.15, 0.07, 0.21, 0.09).

## Acknowledgements

The experiments reported in this paper were carried out at the Caltech Brain Imaging Center. We thank Ralph Adolphs for his invaluable guidance and support in designing and running the experiment as well as for very useful discussions of the results. We thank Ram Rivlin for various conceptual discussions about deliberate versus arbitrary decision-making and about the initial experimental paradigm design. We thank Caitlin Duncan for her help in patiently and meticulously gathering the EEG data. We thank Daw-An Wu for discussions about EEG data collection and preprocessing and for his help with actual data collection. We thank Daniel Grossman for his help in carefully preprocessing the data and suggesting potential interpretations of it. We thank Aaron Schurger and Ueli Rutishauser for various discussions about the model and its simulations. We thank Shlomit Yuval-Greenberg and Leon Deouell for important discussions about EEG processing and analysis. Last, we thank the anonymous reviewers for their invaluable comments, which greatly improved this manuscript.

## Additional information

### Funding

| Funder | Grant reference number | Author |
| --- | --- | --- |
| John Templeton Foundation | BQFW initiative to FSU | Uri Maoz<br>Gideon Yaffe<br>Christof Koch |
| Ralph Schlaeger Charitable Foundation | | Uri Maoz<br>Christof Koch |
| Bial Foundation | 388/14 | Uri Maoz<br>Liad Mudrik |
| German-Israeli Foundation for Scientific Research and Development | I-2426-421.13/2016 | Liad Mudrik |
| John Templeton Foundation | Consciousness and Free Will: A Joint Neuroscientific-Philosophical Investigation | Uri Maoz<br>Gideon Yaffe<br>Liad Mudrik |
| Fetzer Institute | Consciousness and Free Will: A Joint Neuroscientific-Philosophical Investigation | Uri Maoz<br>Gideon Yaffe<br>Liad Mudrik |

The funders had no role in study design, data collection and interpretation, or the decision to submit the work for publication.

### Author contributions

Uri Maoz, Liad Mudrik, Conceptualization, Data curation, Software, Formal analysis, Funding acquisition, Investigation, Visualization, Methodology, Writing—original draft, Writing—review and editing; Gideon Yaffe, Conceptualization, Funding acquisition, Methodology, Writing—review and editing; Christof Koch, Conceptualization, Supervision, Methodology, Writing—review and editing

## Author ORCIDs
Uri Maoz (iD) https://orcid.org/0000-0002-7899-1241
Liad Mudrik (iD) https://orcid.org/0000-0003-3564-6445

## Ethics
Human subjects: The experiment was approved by Caltech's Institutional Review Board (14-0432; Neural markers of deliberate and random decisions), and informed consent was obtained from all participants after the experimental procedures were explained to them.

## Decision letter and Author response
Decision letter https://doi.org/10.7554/eLife.39787.017
Author response https://doi.org/10.7554/eLife.39787.018

## Additional files

### Supplementary files
• Supplementary file 1. NPO names and causes acronyms.
DOI: https://doi.org/10.7554/eLife.39787.012
• Transparent reporting form DOI: https://doi.org/10.7554/eLife.39787.013

### Data availability
The EEG data recorded for this project as well as the behavioral data and a file explaining how to read the behavioral data have been deposited with the Open Science Framework at https://osf.io/s3fdv/.

The following dataset was generated:

| Author(s) | Year | Dataset title | Dataset URL | Database and Identifier |
|---|---|---|---|---|
| Uri Maoz, Gideon Yaffe, Christof Koch, Liad Mudrik | 2018 | Neural precursors of deliberate and arbitrary decisions in the study of voluntary action | https://osf.io/s3fdv/ | Open Science Framework, s3fdv |

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
