## [Decision Letter]

[Editorial note: This article has been through an editorial process in which the authors decide how to respond to the issues raised during peer review. The Reviewing Editor's assessment is that all the issues have been addressed.]

Thank you for submitting your article "Neural precursors of deliberate and arbitrary decisions in the study of voluntary action" for consideration by *eLife*. Your article has been reviewed by three peer reviewers, including Redmond G O'Connell as a guest Reviewing Editor and Reviewer #1, and the evaluation has been overseen by Joshua Gold as the Senior Editor. The following individuals involved in review of your submission have also agreed to reveal their identity: Jiaxiang Zhang (Reviewer #2) and Boris Burle (Reviewer #3).

The Reviewing Editor has highlighted the concerns that require revision and/or responses, and we have included the separate reviews below for your consideration. If you have any questions, please do not hesitate to contact us.

Summary:

This paper sets out to examine whether the readiness potential (RP) relates differently to decision reports when those decisions are deliberative as opposed to the arbitrary decision scenarios typically studied in the literature spawned by Libet et al. EEG data were acquired while participants performed a value-based decision making task involving apportioning donations to one of two causes. In the arbitrary condition the same amount would be apportioned to both causes irrespective of the participant's choice while in the deliberative condition participants determined which of the two causes would receive the money. The authors report a significant RP build-up prior to response in the arbitrary condition but argue that no significant RP activity is present in the deliberative condition. They conclude from this that the RP may act as a form of random number generator (or noise accumulator) specifically for the purposes of arbitrary decisions and, therefore, that the findings from Libet-type experiments may not generalize to the more deliberative decisions that pervade our daily lives.

Major concerns:

The three reviewers agree that this manuscript addresses an important topic and that the findings have potentially important implications for our understanding of the neural mechanisms governing willed action. However, each of the three reviewers has raised major concerns regarding aspects of the data analysis and interpretation which potentially call into question the central claims that the authors are making. The key points that we would recommend the authors address are:

1) The authors are making the strong claim that the readiness potential is absent in the deliberative decision condition but the reviewers (particularly reviewer 1 and 3) have significant concerns regarding the degree to which the data truly support this claim. In particular the initial Bayes Factors analysis indicate inconclusive evidence, the reviewers expressed concerns regarding re-baselining to a response-aligned interval in which the RP appears to be already active in the arbitrary condition and reviewer 3 has also highlighted an important concern regarding the high-pass filer cutoff.

2) All reviewers agree that the model fitting procedures are inadequately described and, as currently presented, it is not clear what value the model is really adding here.

We hope that these comments will prove helpful to the authors.

Separate reviews (please respond to each point):

Reviewer #1:

I thought this was an interesting paper whose results, if correct, could have a transformative effect on how we think about the RP-voluntary action literature. The manuscript is nicely written and the experiment seems to have been conducted with sound methodology on the whole. I do however have very substantial concerns about several aspects of the data analysis and, by extension, the authors' interpretation of the data which I feel would need to be addressed prior to publication.

Major Comments:

1) I think in the first instance it is important that the authors establish specific hypotheses regarding their data. These should be provided at the end of the Introduction. At present the authors state that:

"Demonstrating differences in RP between arbitrary and deliberate decisions would first challenge the generalizability of the RP (from arbitrary to deliberate decisions) as an index for internal decision-making. Second, it would more generally suggest that different neural mechanisms might be at play between deliberate and arbitrary decisions. This, in turn, would question the generalizability of studies focused on arbitrary decisions to everyday, ecological, deliberate decisions-regardless of whether these studies relied on the RP or not."

I find these statements problematic from the outset. First, given how different the arbitrary and deliberate conditions are, it is wholly expected that some trivial differences would be observed in the RP. In the arbitrary condition the participant knows in advance of stimulus onset that they can randomly select a response and so can prepare to act in advance. In contrast the participant cannot prepare a specific action until after stimulus-onset in the deliberative condition. This is borne out in the stimulus-locked traces where clear preparatory RP activity is observed pre-stimulus in the arbitrary condition. So simply stating that a 'difference' would undermine the generalisability of RP findings is not correct. The authors need to be much more specific about what difference that might be. In fact, their analyses are very much geared toward showing that the RP is wholly absent during deliberative decisions. If so, this should be clearly stated from the beginning.

2) Given the emphasis that is placed on demonstrating an absent RP in the deliberative condition, and my aforementioned concerns that trivial differences in RP are to be expected when comparing across these conditions, it is important to consider this aspect of the analysis very carefully. The authors do well to conduct Bayes Factor analyses to complement null-hypothesis significance testing. In the first instance however their analysis highlights inconclusive evidence for the RP's absence on deliberative trials. The authors then re-baseline their waveforms to -1000 to -500ms to exclude possible negativities that may not be RP related. The BF then reduces to 0.332 i.e. hovering just above the conventional cutoff of <0.33. I have a couple of concerns around this. First, the study is designed to ensure 80% probability of detecting an RP >=2.6uV. Thus it is potentially possible that the RP is indeed present but too small to detect with a relatively small sample size (it is worth noting here that visual inspection of the waveforms would tend to favour there being an RP in both conditions albeit with substantial differences in amplitude). I wonder is it a safer bet for the authors to focus on the fact that the RP is substantially diminished rather than absent in the deliberative condition. If so, what new insights might this offer into the nature of our decisions? For example, it would be very interesting if it turned out that the RP is particularly invoked for arbitrary actions as a sort of random-number generator (but see below comments).

3) Following on from the above, given the quite massive differences in RT across conditions, I am concerned that the response-locked baseline is not appropriate. It is quite plausible that the RP will build more slowly in the deliberative condition reflecting the much slower decision process and applying this baseline will simply subtract away those differences and shift the peak amplitude of the RP closer to 0. The authors do conduct some analyses to test for a relationship between RT and RP amplitude but these are based on a cross-subject median split and, consequently, an N of just 9 in each group. Moreover, even after splitting the groups this way there is still a circa 700ms difference in RT between conditions. The authors observe p-values >0.05 in these follow-ups but no Bayes Factor analyses are provided. Similarly, the regression of RT difference against RP amplitude is likely to be underpowered and no BFs are reported. I suspect that if the authors were to conduct a within-subject median-splits based on RT that an RT effect would be much more likely to emerge in a within-subjects statistical analysis. In my experience, across subject variations in signal amplitude are a much bigger problem than cross trial within-subject variations. In any case, wouldn't a non-significant relationship between the RP and RT be confusing given the previous literature? At minimum, shouldn't we be observing a relationship within the arbitrary condition?

4) The leaky DDM accounts for the shape of the ERP at CZ. I cannot find any description of precisely how the plot in Figure 8C was generated. Does it represent the cumulative noise component?

Minor Comments:

To what degree might the deliberative RP amplitude be impacted by overlapping decision signals. Recent work by my own group has highlighted a P3-like centro-parietal potential that traces perceptual decision formation and peaks at response (e.g. O'Connell et al., 2012, Nat Neurosci; Kelly and O'Connell, 2013, J Neurosci; Twomey et al., 2015, Eur J Neurosci). In Kelly and O'Connell, 2013, we actually observed an RP-like signal evolving around the same time as the CPP and we found that the two signals were impacting one another, something that we were able to resolve using current-source density (CSD) analysis. I wonder what the authors thoughts are on A) the fact that we seemed to observe an RP during deliberative decisions in this case and B) whether an overlapping centro-parietal positivity could partly account for the small size of the RP in their analyses – something that could be verified through application of CSD transforms.

The description of the Schurger study in subsection “Drift Diffusion Model (DDM)” is a little unclear, probably just down to the particular wording used. The meaning of 'non-linear threshold crossing' is not clear to me. 'The crossing of that threshold reflects the onset of the decision in the model, typically leading to action.' 'Onset of the decision' is an ambiguous phrase to use here as it is often used with reference to the time at which evidence accumulation commences. According to these models the threshold instead reflects the completion/termination of the decision. Also the authors state 'Schurger and colleagues claimed, time-locking to response onset ensures that these spontaneous fluctuations appear, when averaged over many trials, as a gradual increase in neural activity.' It is not clear to me whether the authors are aware that according to Schurger this arises because the RP reflects an accumulation of those spontaneous fluctuations over time as opposed to a moment-by-moment reflection of instantaneous fluctuations.

LRP analyses. Again these hinge on embracing the null hypothesis. Given the low statistical power and the signs of a larger peak amplitude (and possibly slower build-up) for deliberative decisions, at minimum bayes factor analyses should be provided.

Reviewer #2:

This is an interesting study. The authors challenged the role of RP in internally-generated decision, an EEG signature often linked to volition or action awareness since Libet's work. They contrast deliberate decision (choices did matter) with arbitrary decision (choices do not matter) in a "two-alternative NPO choice" task. Both types of decisions had the similar LRP component, but only arbitrary decision showed reliable RP (at Cz). The authors further simulated an altered version of DDM, showing the absence of RP when assuming a value component underlying the deliberate decision.

The experimental design and analysis are rigorous. The authors should be applauded for their effects on assessing several alternative explanations to the absence of RP in the deliberate decision.

I have a few comments on current version of the manuscript.

1) The modelling procedure is unclear. First, it is unclear why drift rate was the only parameter allowed to change between conditions. Since different task conditions were blocked (see the other comments below), one may argue that it is possible to have the threshold (or other parameters) to vary between blocks, as an adaptive decision strategy. Would model comparison be feasible to identify the optimal parameters?

Second, DDM (or many other accumulator models) assumed one-off, rapid decisions without deliberate thinking. In other words, the very first boundary crossing would render a decision, as in the current model. Although the subjects were instructed to respond asap, the current design cannot rule out the possibility of rethinking/change-of-mind in a single trial, which violates the model/simulation assumption. The prolonged RT in deliberate decision support such possibility, that subjects might not rush to a decision, even when a decision threshold was reached.

2) The arbitrary decision explicitly urged subjects "not to let their preferred NPO dictate their response". This is a strong requirement (asking NOT to follow their preference), compared with previous studies using the typical free-choice paradigm). I wonder if the authors could comment on whether this may inflate the difference between the two types of decisions in the current study. Second, is there any regularity in the arbitrary decision, such as alternating response in consecutive trials?

3) Different decisions were blocked in the current study. Could the difference in RP be due to a contextual effect? For example, was the early visual ERP comparable between conditions?

4) Subsection “Differences in reaction times (RT) between conditions, including stimulus-locked potentials and baselines, do not drive the effect”. For the median split analysis to work, my understanding is that no subject had above-median deliberate RT as well as below-median arbitrary RT (i.e., the two groups did not overlap). Could the authors confirm that this was the case? Because inferences on the null hypothesis are important to rule out the effect of RT and other confounding variables on RP, it would be useful to report Bayes factor as well.

Minor Comments:

Figures 8A and 8B could be superimposed over each other, allowing easy assessment of the goodness of fit of RT distributions.

Reviewer #3:

The present manuscript presents a single experiment aiming at evaluating whether the typically observed early BP onset on spontaneous (arbitrary, purposeless, unreasoned and without consequences…) movements generalizes to deliberate decision making. The authors hence contrasted two decision making situations: in a first condition, participants had to decide, on every trials, to which NPOs the wanted to give money; their choice was effective. In a second condition, while facing the same situation, they were told that their response was consequenceless, since the same amount of money would be given to each NPOs. While a standard BP was observed in the second condition, the authors argue that there is no such BP in the first condition. They hence argue that the results classically observed in arbitrary movement do not generalize to deliberate decision. The early onset of BP has long been considered as reflecting an activation of a response before the actual decision to move was taken, casting doubts on the very notion of free-will. The absence of such pre-conscious marker in deliberate decision is taken by the authors as an argument that, if there is no real free-will in arbitrary movement, this argument cannot be used for deliberate decision.

General comments:

Although I may generally agree with the authors' conclusions, I'm afraid they are not strongly supported by the facts as they stand. The limitations are methodological and conceptual.

While the data show a clear difference in the BP between the two conditions, the core of the authors reasoning is based on the presence of a BP in the arbitrary condition and the *total absence* of BP in the deliberate one. Indeed, if there were a BP, even of smaller amplitude, in this condition, the argument would immediately fall. There are, however, several points that may challenge this view.

1) Data were high-pass filtered at 0.1 Hz (which corresponds to a time-constant of 1.6 s). This value is way too high for a proper estimate of the BP. As a matter of comparison, the very first measures of BP were even performed under DC conditions, that is without any filtering… At the very minimum, a filter of.01 Hz, or ideally even lower, like.001Hz should be used. Indeed, the BP is a very slow component, such a high filter value has very likely largely deceased the amplitude of the BP. One may argue that filtering has impacted the two conditions in the same way, and hence that any potential distortion cannot explain the results. This is certainly true concerning the difference in amplitude (see however below). But, as indicated above, the rationale put forward by the authors only holds if there is no BP on the deliberate condition, not if the BP is simply reduced. If, with a more adapted filter, there would have been a BP, the whole argument is invalidated. Second, the response time are much longer in the deliberate than in the arbitrary case. So the time constant of the slow potential might be much longer in this case, and hence much more affected by the inappropriate filter (the control proposed in only partially addressing this issue since i) the comparison is between participants, and ii) at least for the easy deliberate condition, the results are ambiguous).

2) Figure 6 presents the stimulus-locked data, supposed to invalidate a stimulus-locked contamination. While I agree that the response-locked BP is not purely a stimulus induced effect, several features of the stimulus-locked averaged deserve comments. First, as can be seen in Figure 6A, a negative shift (CNV-like) is already present before stimulus presentation in the arbitrary condition, and it seems even modulated by (precued) task difficulty. So, even before participants enter any decision making process related to the relevant choice, one can see differences in the slow negative potentials between the conditions. At minimum, this indicate that the difference observed response-locked might not be specific to the decision period. Furthermore, following the large visual evoked potentials after the stimulus, one can see around 500 ms a first positive component, followed by a negative-going one (starting around 650 ms). This negative-going peaks close to the average RT in the arbitrary condition, and hence likely contribute to the BP. The same activity is present in the deliberate condition (and is actually very similar), but the response being given much later, this early negative-going activity does not contribute to the BP. By itself, this activity, which is not strictly speaking a BP, does create a difference between the two conditions. More generally, the general shape of the late evoked potentials (after 600 ms) are remarkably similar across conditions, which dramatically contrasts with the large difference observed response-locked. This may suggest that the difference observed is, at least in part, simply due the different averaging event is an otherwise very similar signals.

3) The authors focused on Cz to extract the BP. However, there is large literature indicating that the BP is actually made of several sources, some medial in the (pre)SMA, some more lateral in the (primary) motor cortices. Note that, for the authors' rationale to be valid, none of the sources should present a (early) BP. The topographies presented on Figure 3B indicate a clear negative activity above Cz and neighbors in the arbitrary condition, and no activity in the same region in the deliberate condition. But the topography suggests a clear negative activity located more lateral over the left hemisphere. Does this activity correspond to a BP? Since only Cz is shown, one cannot evaluate this possibility.

4) In real spontaneous responses, the BP starts much earlier than the LRP. But in the present data set, as can be inferred from Figure 3A, the conditions diverge around -800 ms (roughly estimated). This is basically the same latency for LRP onset. Note that the latency of the LRP (even with all the cautions that may come with this measure…) seems exactly the same for all 4 conditions. So, at the time of BP divergence, the LRP onset indicates not difference in response selection / decision making. Why is the (CZ-)BP more important for the conclusions the authors want to defend? LRP seem to say something very different…

5) A further issue concerns the modeling part. First, it is very unclear how the fit was performed. In subsection “Drift Diffusion Model (DDM)”, it is indicated that "We fit our DDM to our average empirical reaction-times, which were 2.13, 2.52, 0.98 and 1.00s for the different conditions ". This sentence suggests that only the grand mean were used, not the whole distribution of RTs. Besides the fact that this is in complete deviation with the overall logic of fitting DDM which aims at capturing RT distribution shape and errors at the same time, a fit performed on averages only is largely under-constrained. As a matter of fact, although there is no information about quality of fit, comparison of Figures 8A and B indicate that the empirical and predicted distribution largely differ, especially in terms of spread (i.e. variance). The obtained parameters are also weird and likely invalid: Table 1 indicates that, in the deliberate condition, the estimated drift is actually *higher* for the hard condition than for the easy one. This does not make any sense, and is disagreement with the empirical RT that are, as expected, longer for the hard than for the easy condition. Furthermore, one reads "[…] The model was *further* fit to the empirical consistency ratio […]" (emphasizes are mine…). "Further" suggests that the fit to the mean RT and the consistency ration were not performed simultaneously? Is that the case? Please clarify.

Besides these fitting problems, I have conceptual issues. Threshold crossing in the arbitrary condition is supposed to be triggered by noise only. In standard accumulation-to-a-bound models, noise is classically assumed to be gaussian noise with mean 0 and a standard deviation s. For "noise" to hit the threshold one needs to assume a very high s. Or to assume that the mean is not 0… But in this case, this is not noise, this is a drift. Actually, Table 1 indicates: the "drift" in the arbitrary condition is much higher than in the deliberate condition. But what is the meaning of this "drift" if there is no decision, and the response is triggered by random fluctuations of noise? In contrast, what does it mean that a deliberate decision was reached with an information accumulation equals to 0? How can a threshold be reached if there is no information accumulation? If it's simply noise, why is different from the arbitrary condition? Furthermore, if random noise can hit the bound early in the accumulation process, if noise is 0-centered, the probability to hit the threshold decreases with time. So late responses are unlikely to be triggered by noise. So, this whole fitting part is largely based on inconsistent assumptions.

My general feeling is that the authors provide further arguments against the idea that BP can be used as a marker of volition, but this is not new. In contrast, they provide few (if any) new arguments in the discussion of whether there is free-will or not.

Minor comments:

– Bayesian analysis. The core idea of Bayesian ANOVA is that the values are indicative, but no hard threshold should be used. Hence, considering that a BF of.332 (that is a probability ratio > 3) is an argument for no effect is wrong. Actually, in this uncertainty zone, one cannot conclude anything, and this value is NOT an argument for H0

– EEG results. Authors report a main effect of decision type. But is there an interaction with difficulty? If not, the following t test are not completely valid (although this does not change the conclusions…)

– Figure 5B: the data plotted in this figure puzzles me. The y-axis plots the difference (deliberate – arbitrary). 12 points out of 17 (?) are actually negative, indicating that deliberate was more negative than arbitrary. How can this be?

– Subsection “EEG Results: Lateralized Readiness Potential (LRP)”, the argument that LRP amplitude difference is due to the reference used is incorrect: LRP measure is reference-free. Indeed, the formula is (C3 -C4)r + (C4-C3)l. But actually all electrodes values should be written as (C3-ref) and (C4-ref). It becomes then obvious that the reference annihilates and does not intervenes in the computation

– Modeling. The authors sometimes refer to a DDM, sometimes to a LCA, sometimes to a race. Those are very different architectures (although they can be related, see Bogacz et al., 2006). Please clarify what is really used.

– Experimental procedure. It is never indicated how the response were given! From the description (response keys) and EEG analysis (computation of LRP), one can infer that the response were given with the left and right hand, right? But with which finger?

– Subsection “Experimental Procedure” and other: besides providing the t or F values, please provide the behavioral data. For example, please provide error rates (even if they do not statistically differ). Furthermore, considering that a difference with a p value equal to.09 as not significant, is a bit strong.

[Editors' note: further revisions were suggested prior to acceptance, as described below.]

Thank you for resubmitting your work entitled "Neural precursors of deliberate and arbitrary decisions in the study of voluntary action" for further consideration at *eLife*. Your revised article has been favorably evaluated by Joshua Gold (Senior Editor) and three reviewers, one of whom served as the guest Reviewing Editor.

All three reviewers agree that you have made substantial efforts to revise the manuscript in light of the initial round of comments and this clearly took time and effort on your part. However, I'm afraid we are all in agreement that some further revisions would be strongly recommended prior to publication. The remaining issues boil down to two substantive points:

First, we are still not convinced that the present results call for any alteration in our conception of free will. The key contribution of the paper seems to be to show that the RP may be specialised for arbitrary action selection and this seems worth reporting. However, beyond suggesting that the RP might not be a universal marker of the 'urge to act', the authors have not yet made a sufficiently clear case that their findings call for any change in how we think about the Libet findings or free will in general. Our recommendation at this point would be that the authors reframe the paper to focus more specifically on the RP and how the current findings bear on functional accounts of this signal.

Second, the reviewers also have significant outstanding concerns regarding the modelling. At minimum we would suggest clearly flagging the limitations of the adopted approach in the Discussion.

Reviewer #1:

I think the authors have, by and large, done a good job of addressing the methodological concerns raised by the reviewers. Importantly the authors assert that they are not claiming that the RP is actually absent but rather significantly diminished during deliberative decisions. I have a couple of outstanding concerns.

I am still confused about the overarching premise or rationale for this study. It seems to me that the findings of this study boil down to showing that the RP is smaller for deliberative vs arbitrary decisions but I am not yet convinced that this has any bearing on our understanding of free-will or even on the significance of Libet's original reports. Libet showed that a neural signature of action preparation preceded conscious awareness of the decision to arbitrarily act. Perhaps that same signal (RP) is absent during deliberative decisions (and this is worth reporting) but that tells us nothing regarding the role or timing of conscious processes in this context. Of course it goes without saying that arbitrary and deliberative decisions involve distinct cognitive elements and, consequently, will activate some distinct brain areas. We know that several other signals that reflect action preparation (e.g. LRP, β-band desynchronisation) and evidence accumulation (e.g. P300, LIP spiking activity) precede explicit decision reports by substantial amounts of time.

The authors seem to suggest that Schurger's model presents a challenge to Libet's interpretation: "A further reason to expect such differences stems from a recent computational model, which challenged the claim that the RP represents a genuine marker of unconscious decisions. Rather, the model suggested that the RP might reflect the artificial accumulation, up to a threshold, of stochastic fluctuations in neural activity"

I am still struggling to understand what difference it makes, in terms of our understanding of free will, if RP reflects 'action preparation' or 'noisy evidence accumulation' – wouldn't both processes would reflect the emergence of an 'urge to act'? Moreover, the authors seem to imply that Schurger's process must be conscious but I see no reason to make such an assumption. I think that particular sentence really captures my discomfort with the current framing of the findings.

Another way of putting it is that I am not clear on how exactly people were making generalisations to deliberative decisions based on the RP specifically. I get the impression the authors are trying to make the case that the RP has been thought of as the sole signature of pre-conscious decision making/action preparation in the brain but is that really a widely held view, again given the literature on decision making? I think that perhaps the authors should reframe the paper to focus more narrowly on the apparent domain specificity of the RP. On the same point I think the title of the paper may be too broad given the almost exclusive focus on the RP. I am very much open to being convinced/correct on all of the above points but I feel that currently the paper is rather confused and confusing as regards the relevance of the findings to free will.

My only other comment is that I do not agree with the authors insistence on only allowing one DDM parameter to vary across conditions on the grounds of parsimony. Although this is indeed a common approach in behavioural modelling, there are formal model comparison procedures available that would objectively identify the model that provides the best balance between parsimony and goodness of fit. Personally I would probably be ok with the authors acknowledging this in their Discussion as I do not view the modelling as a critical part of the story

Reviewer #2:

In this revision, the authors addressed some of the concerns in my last review. However, although the MS now has more detailed description of the model and simulation results (which was missing in the previous version), there are some additional issues.

1) The model used 11 parameters, 8 drift rates, a scaling parameter (1.45), threshold and decay to fit 8 data points (4 mean RT and 4 choice probability). The result is a largely unconstrained model that does not describe behaviour accurately. In the response letter, the authors showed model simulation overlayed with empirical data, and there is a large discrepancy in the fit of arbitrary condition. If the authors are determined to present current modelling results, as a proof-of-concept that the model can provide qualitative RP patterns, the limitation (that it does produce satisfactory quantitative fit to RT distributions as in other applications of DDM) needs to be acknowledged in Discussion.

2) Subsection “Model and Simulations”. "Using a parameter sweep.…". Please provide details on the fitting procedure. Was any optimization algorithm used here? If so, what was the cost function? Was the model fitting performed on averaged data or individual data?

3) In the response letter to my point 1, "we think that the longer RT in deliberate trials stemmed from longer deliberation time, reflected in a higher threshold." This is confusing as the same threshold (0.15) across conditions was used in the paper?

4) Figure 8A. There are two independent accumulators in the "noise component". From my understanding of the model description, in arbitrary decisions, SMA activities were reflected in the traces of the winning accumulator only. If that is the case, why is the activity of the losing accumulator in the noise component not taken into account, as sculp EEG activity would not be sensitive or selective to one accumulator vs. the other. The same issue holds for the deliberate decisions, if both accumulators in the noise component do not dictate decision (as threshold was set to infinity), which accumulator (or both?) was representative of SMA activity? Please clarify.

Reviewer #3:

In the first round of review, several questions were raised. The authors did a very good job in answering most of them. It is now clear that a RP is observed in the arbitrary condition, but not in the deliberate one. There are still, however, two points that remain unclear to me.

1) What are the theoretical consequences of these results?

As said above, the authors convincingly show that RP is absent in the deliberate condition. Although this is not entirely new (some previous studies have reported absence of the BP before voluntary movement depending on the context), this report certainly adds to our understanding of the RP, its origins and functional interpretation. However, as can be read in the Abstract, the goals of the authors is much more than that. But I am not sure about the real theoretical impact of the results. Let me try to explain my trouble.

Libet's original report was that the RP starts before the conscious decision to move. It was thus argued that our intention to move is rather a consequence, not a cause, of the preparatory brain activity. This was taken as an argument that "free-will" is an illusion.

Here, the authors report that there is not RP in the deliberate condition. Hence, the nature of the decision differs between the two conditions. So, what do exactly the authors conclude from that?

– the Libet's argument vanishes for deliberate condition, hence there is no evidence against free will in this context. But do they think it still holds for arbitrary ?

– since RP might simply accumulated random noise, it is not an indication of voluntary movement decision, and hence Libet's argument is wrong even for arbitrary movement?

– if deliberate decision are made on another region X, it might still be that activity in region X starts before conscious detection, but this remains to be explicitly studied.

– something else?

I must confess that I cannot really get the real conclusion the authors want to defend, and they should try to be more explicit on what these results do really imply, and what they do not, on this free-will debate.

2) Modeling.

Although many points have been clarified in this new version, some still remain a bit unclear.

To account for the choice situation, the authors modified Schurger et al. original model (who contained a single accumulator), and implemented two accumulators racing for the response. First, one may wonder why they did not choose the standard competitive version of the leaky accumulator (Usher and McClelland, 2001). Second, it is not completely clear how the averaged data on Figure 9C was actually computed. I guess that, for each "trial" the winning accumulator was chosen (left or right) and all the traces of the winning accumulators were averaged. However, if two accumulators are racing within "SMA", the real simulated activity of SMA should be the sum of the two accumulators, not only the winning one. I'm not sure how this would modify the results, but for coherence, this is the way "SMA" activity should be evaluated in the model. Third, I still don't understand how the fit was performed. It is said "[…] we fit our DDMs to our *average* empirical reaction-times […]" (emphasizes are mine). It the fit was indeed performed only on the averages, this is non standard, and highly under-constrained. Such model are normally fitted to the whole response times distributions. Furthermore, there is no quantitative assessment of the fit quality. Comparison of figure 9 panels A and B, suggests that the fit was not very good, especially in the arbitrary condition. Fourth, Figure 9 only shows the activity of "SMA". However, there is another actor, which is never shown: region "X"… In the deliberate condition, the decision in made base on the activity of this "region", but its dynamics is likely very different from "SMA" one. It would be of interest to see the accumulated activity of this region "X" in both the deliberate and arbitrary conditions. A last question concerns what "SMA" is doing. In the arbitrary condition, its accumulating random, spontaneous noise. But why is it not doing the same in the deliberate condition (in addition to accumulation in region "X")? Do the authors assume a form inhibition of region "X" on "SMA" to prevent it from accumulating? This part is bit too "magic" and an explicit, mechanistic, explanation would be useful, instead of just claiming that accumulation is done differently as a function of the context/choice (which is vague). Somehow relate to this last point, there seems to be a bit of (simulated) accumulated activity in the deliberate conditions in "SMA". Where does it come from?

Besides, I have some more specific points:

Introduction section: "[…] Thus, one could speculate that different findings might be obtained when inspecting the RP in arbitrary compared to deliberate decisions. […]" is still very unspecific.

Introduction section: […] Demonstrating no, or considerably diminished, RP in deliberate decisions would challenge the interpretation of the RP as a general index of internal decision-making.[…] Ok, but the fact that it is not a "general index" does not, de facto, solve Libet's problem: even if reduced, if RP start before conscious decision, the argument is still valid.

[…] More critically, it would question the generalizability of studies focused on arbitrary decisions to everyday, ecological, deliberate decisions […] This is indeed critical for the functional interpretation of the RP, but this sounds partly orthogonal to the free-will debate.

Subsection “EEG Results: Readiness Potential (RP)” paragraph two: why are the student tests corrected for multiple comparisons, since only 4 are performed? Does it mean that the authors performed t-tests for all time-points? In this case, a multiple comparison is, indeed, necessary. But only one t-test is reported! Please clarify.

In the same section: BF = .332 is not a serious evidence for no effect. The authors should not take 3 (or.33 depending on how we compute it) as a threshold above which an effect would become "significant"…

Subsection “EEG Results: Readiness Potential (RP)” paragraph four: while a BF of.09 is indeed an argument for no effect, a BF of.31 is not really.

Subsection “Differences in reaction times (RT) between conditions, including stimulus-locked potentials and baselines, do not drive the effect”: the authors discuss at length the potential impact of (or absence of) the chosen baseline. Besides all the rather indirect arguments based on different baselines (none is immune of criticisms), one analysis suffices to invalidate this argument: the slopes of the linear regressions are, by construction, independent of baseline (only the intercept is). So the fact the slopes more negative for arbitrary than for deliberate condition is a strong and not disputable fact, much stronger than all the baseline changes.

Figure 6: I'm personally convinced that the RP observed on the arbitrary condition is not (only) a contamination by Stimulus-locked activities. However, the arguments based on Figure 6 are pretty weak. Indeed, in the time windows from about 600 ms to 1200 ms, a negative ramp is observed for all 4 conditions. The response is given in this time windows for the arbitrary condition, but for the deliberate one. So, this stimulus-locked negative ramp likely contributes to the RP.

Minor Comments:

Subsection “Experimental Procedure”, fourth paragraph: "right and left index finger" -> "left and right index finger" to be more consistent with the rest of the text.

Subsection “Experimental Procedure”, final paragraph: "[…] We wanted to make sure subjects were carefully reading and remembering the causes also during the arbitrary trials to better equate memory load, attention, and other cognitive aspects between deliberate and arbitrary decisions […]" Although adding a task is a good idea, it may sound a bit naive to say that the task were equated. For example, in the arbitrary decision, there is a short term memory component that is not present in the deliberate.

Subsection “ERP analysis”: "offline to the average of all channels,": including mastoids and nose? I guess not… At least, this should not be the case! Please clarify.

Subsection “ERP analysis”: " which subjects pressed the wrong button " What is a wrong button? Inconsistent response? and in arbitrary condition?

Subsection “ERP analysis”: "Channels that consistently had artifacts were replaced using interpolation (4.2 channels per subject, on average ": Although this is the range acceptable by some standards, I personally find this value high. Furthermore, could we have the range of channels interpolated?

Subsection “Statistical Analysis”: The authors took 1 point over 10 to re-sample the signal. However, re-sampling requires appropriate anti-aliasing filtering to avoid signal distortion. Data were acquired at 512 Hz; Biosemi anti-aliasing filter, if I'm not mistaken, should be around 100Hz. Since no other low-pass filtering was applied, the data contains signal up to 100Hz. Hence, (re)sampling at 50Hz a signal whose max frequency is around 100Hz is extremely problematic. At minimum a 25Hz low pass filter should have been applied… It is very hard to anticipate what would be the impact of such aliasing (especially since the activity of interest is low frequency), but this should be corrected to avoid having incorrect practices published.

Subsection “Model and Simulations”: "used Δt = 0.001, similar to our EEG sampling rate": if sampling rate is 512Hz, dt should be 0.002

[Editors' note: another round of revisions were suggested prior to acceptance.]

Thank you for resubmitting your work entitled "Neural precursors of deliberate and arbitrary decisions in the study of voluntary action" for further consideration at *eLife*. Your revised article has been favorably evaluated by Joshua Gold (Senior Editor) and two reviewers, one of whom served as a guest Reviewing Editor.

The manuscript has been substantially improved. As part of this peer review trial, as reviewing editor I am required to indicate whether all of the reviewer comments have been fully addressed or if minor or major issues remain. There are some minor comments arising from this latest round of reviews that I thought I would give you the opportunity to address prior to finalising the 'Accept' decision so that I can then potentially indicate that all reviewer concerns were addressed. I have outlined these below. If you prefer to expedite the publication and not address these comments you can let me know.

1) Table 1. As expected, the difference of drift rates between congruent and incongruent options was larger in the deliberate than the arbitrary conditions. Could the authors comment on the large difference in the noise scaling factor c, which was 10-fold between the two types of decisions? The second result I found difficult to conceptualize was the decay rate k, which doubled in the easy-deliberate than in the hard-deliberate condition. Given that task difficulty was randomized across trials, doesn't this imply that the model (and the participants) adjusted the decay rate according to task difficulty prior to trial onset?

2) Figure 9A. It is more meaningful to plot the empirical and simulated RT distributions, rather than their fitted γ functions.

3) In several instances the authors use the term 'decision onset' when referring (I think) to the completion of the decision. This is potentially confusing because for many readers 'decision onset' may refer to the beginning of deliberation/evidence accumulation which means something entirely different. So I would suggest the authors check their terminology and use 'decision completion' or 'commitment' in such instances.

Minor comments:

1) Subsection “Behavioral Results”. DDN

2) Figure 9A. Why was the y-axis labelled as voltage, for RT distributions?

3) Subsection “Model and Simulations” third paragraph and Table 1. I am confused about the scaling parameter 1.45. Does Table 1 show the drift rates only in Region X, and are the drift rates in SMA 1.45 times less than those values? The text and table indicated that the scaling applied only to the deliberate condition, if so, what were the drift rates in SMA in the arbitrary condition? Or do the drift rates in arbitrary decisions in Table 1 refer to the values in SMA?

---

## [Author Response]

Major concerns:The three reviewers agree that this manuscript addresses an important topic and that the findings have potentially important implications for our understanding of the neural mechanisms governing willed action. However, each of the three reviewers has raised major concerns regarding aspects of the data analysis and interpretation which potentially call into question the central claims that the authors are making. The key points that we would recommend the authors address are:1) The authors are making the strong claim that the readiness potential is absent in the deliberative decision condition but the reviewers (particularly reviewer 1 and 3) have significant concerns regarding the degree to which the data truly support this claim. In particular the initial Bayes Factors analysis indicate inconclusive evidence, the reviewers expressed concerns regarding re-baselining to a response-aligned interval in which the RP appears to be already active in the arbitrary condition and reviewer 3 has also highlighted an important concern regarding the high-pass filer cutoff.

We thank the reviewers for pointing out these potential issues. For the readiness potential (RP), we now ran an additional analysis using the gold-standard Maris and Oostenveld cluster-based nonparametric method as well as a Bayesian analysis on the trends, both of which further supports our claim that the RP is absent in deliberate decisions. Overall, we now conducted 6 different kinds of analyses, using both NHST and Bayesian methods. None of the analyses we conducted supported the claim that there exists an RP in deliberate decisions. All but one of our 6 analyses supported the claim that the RP is absent in deliberate decisions. The remaining analysis suggests inconclusive evidence for the absence of an RP. Therefore, we think that—taken together—our results provide clear evidence for an absence of RP in deliberate decisions.

Nevertheless, even a reader who remains unconvinced that the RP is absent in deliberate decisions would agree that the RP is strongly diminished during deliberate decisions in comparison to arbitrary ones. And that is enough to support our main claims in this manuscript:

1) Deliberate and arbitrary decisions may involve different neural mechanisms, and that

2) Generalizing from arbitrary to deliberate decisions is problematic.

We now explain these claims better at the end of the Introduction and in the Discussion. More details about this is provided in response to reviewer comments below. We also reran our analyses with the lower high-pass filer that Reviewer 3 suggested; the results remain qualitatively the same. This again is discussed in more detail in the response to Reviewer 3.

2) All reviewers agree that the model fitting procedures are inadequately described and, as currently presented, it is not clear what value the model is really adding here.

We thank the reviewers for pointing out that our explanation of the model was not clear enough. In the current version of the manuscript, we rewrote the section describing the model and added another figure to describe and explain it better. We think that in the current version the model and its contribution to the results are much clearer.

We hope that these comments will prove helpful to the authors.Separate reviews (please respond to each point):

Reviewer #1:

I thought this was an interesting paper whose results, if correct, could have a transformative effect on how we think about the RP-voluntary action literature. The manuscript is nicely written and the experiment seems to have been conducted with sound methodology on the whole. I do however have very substantial concerns about several aspects of the data analysis and, by extension, the authors' interpretation of the data which I feel would need to be addressed prior to publication.Major Comments:1) I think in the first instance it is important that the authors establish specific hypotheses regarding their data. These should be provided at the end of the Introduction. At present the authors state that:"Demonstrating differences in RP between arbitrary and deliberate decisions would first challenge the generalizability of the RP (from arbitrary to deliberate decisions) as an index for internal decision-making. Second, it would more generally suggest that different neural mechanisms might be at play between deliberate and arbitrary decisions. This, in turn, would question the generalizability of studies focused on arbitrary decisions to everyday, ecological, deliberate decisions-regardless of whether these studies relied on the RP or not."I find these statements problematic from the outset. First, given how different the arbitrary and deliberate conditions are, it is wholly expected that some trivial differences would be observed in the RP. In the arbitrary condition the participant knows in advance of stimulus onset that they can randomly select a response and so can prepare to act in advance. In contrast the participant cannot prepare a specific action until after stimulus-onset in the deliberative condition. This is borne out in the stimulus-locked traces where clear preparatory RP activity is observed pre-stimulus in the arbitrary condition. So simply stating that a 'difference' would undermine the generalisability of RP findings is not correct. The authors need to be much more specific about what difference that might be. In fact, their analyses are very much geared toward showing that the RP is wholly absent during deliberative decisions. If so, this should be clearly stated from the beginning.

We thank the reviewer for this comment. Following it, we went into more detail about our hypothesis in the Introduction and now clarify and justify them better (Introduction section). Briefly, we hypothesize that the RP is present in arbitrary decisions and absent in deliberate ones. However, we note that it is enough for the RP to be strongly diminished during deliberate decisions in comparison to arbitrary ones to support our main claims in this manuscript: (1) deliberate and arbitrary decisions may involve different neural mechanisms, and (2) generalizing from arbitrary to deliberate decisions is problematic.

Regarding differences between arbitrary and deliberate decisions, we tried to equate the two as much as possible in terms of the sensory input, motor output, and even memory and cognitive load (the catch trials). Additional differences are inherent to the difference in decision types, and accordingly part of the research question (e.g., in deliberate decisions subjects are devoting more thought to the decision and cannot prepare the decision in advance). In reference to the specific concern made by the reviewer regarding the differences in preparatory activity between arbitrary and deliberate decisions and stimulus-locked activity (Figure 6A), we do not think the data supports the interpretation of a preparatory activation in arbitrary and not deliberate decisions. Note that the arbitrary hard activity is indistinguishable from the two deliberate conditions, and it is only the arbitrary *easy* trials that diverge from the other conditions. However, easy and hard trials were randomly interspersed in deliberate and arbitrary blocks, and the subject discovered the trial difficulty only at stimulus onset. Thus, there couldn’t have been differential preparatory activity that differs with decision difficulty. This divergence in one condition only is accordingly likely due to some fluke in the data rather than reflecting any preparatory RP activity. We now specifically address that in the manuscript.

2) Given the emphasis that is placed on demonstrating an absent RP in the deliberative condition, and my aforementioned concerns that trivial differences in RP are to be expected when comparing across these conditions, it is important to consider this aspect of the analysis very carefully. The authors do well to conduct Bayes Factor analyses to complement null-hypothesis significance testing. In the first instance however their analysis highlights inconclusive evidence for the RP's absence on deliberative trials. The authors then re-baseline their waveforms to -1000 to -500ms to exclude possible negativities that may not be RP related. The BF then reduces to 0.332 i.e. hovering just above the conventional cutoff of <0.33. I have a couple of concerns around this. First, the study is designed to ensure 80% probability of detecting an RP >=2.6uV. Thus it is potentially possible that the RP is indeed present but too small to detect with a relatively small sample size (it is worth noting here that visual inspection of the waveforms would tend to favour there being an RP in both conditions albeit with substantial differences in amplitude). I wonder is it a safer bet for the authors to focus on the fact that the RP is substantially diminished rather than absent in the deliberative condition. If so, what new insights might this offer into the nature of our decisions? For example, it would be very interesting if it turned out that the RP is particularly invoked for arbitrary actions as a sort of random-number generator (but see below comments).

We agree with the reviewer that our results clearly state that the RP in deliberate decisions is at least strongly diminished with respect to arbitrary decisions, and we now specifically address this interpretation in the manuscript, in various locations. And we agree that a reader that remains unconvinced by our 6 different analyses would still likely agree that the RP is strongly diminished in deliberate decisions. Note, however, that our power analysis suggested that we would have a 99% (and not 80%) probability of detecting an RP>=2.6 uV with 16 usable subjects (Materials and method, “Sample size and exclusion criteria”). We had 18 subjects, so our power was even higher.

Regarding the visual inspection that the reviewer notes might suggest a diminished RP, this is actually in line with our results according to our model. The model predicts that we would find a roughly linear decrease in activation in the deliberate condition (Figure 9C). This is now discussed in more detail in the modeling section of the Results.

3) Following on from the above, given the quite massive differences in RT across conditions, I am concerned that the response-locked baseline is not appropriate. It is quite plausible that the RP will build more slowly in the deliberative condition reflecting the much slower decision process and applying this baseline will simply subtract away those differences and shift the peak amplitude of the RP closer to 0. The authors do conduct some analyses to test for a relationship between RT and RP amplitude but these are based on a cross-subject median split and, consequently, an N of just 9 in each group. Moreover, even after splitting the groups this way there is still a circa 700ms difference in RT between conditions. The authors observe p-values >0.05 in these follow-ups but no Bayes Factor analyses are provided. Similarly, the regression of RT difference against RP amplitude is likely to be underpowered and no BFs are reported. I suspect that if the authors were to conduct a within-subject median-splits based on RT that an RT effect would be much more likely to emerge in a within-subjects statistical analysis. In my experience, across subject variations in signal amplitude are a much bigger problem than cross trial within-subject variations. In any case, wouldn't a non-significant relationship between the RP and RT be confusing given the previous literature? At minimum, shouldn't we be observing a relationship within the arbitrary condition?

This comment includes several criticisms. We address them one by one.

1) We thank the reviewer for this comment but note that the response-locked baseline analysis is just 1 of 6 analyses we carry out on our data to investigate the RP (and, as we explain above, though it is indeed inconclusive, it is in the expected direction). One of these six analyses, which was added in this revision, was a Bayesian analysis of the downward trend in the time window of the RP (the last 1 s before movement onset). This new analysis provided extremely strong evidence for a downward slope in the arbitrary case and moderate to strong evidence against the existence of a slope in the deliberate case. Thus, taken together, our results provide clear evidence for an absence of RP in deliberate decisions. We detail this in the manuscript (Results and Discussion).

2) Following the reviewer’s suggestion, we conducted a within-subjects analysis, taking faster deliberate trials and slower arbitrary trials for each subject. As is now reported in the text (Results section), the mean difference between fast deliberate and slow arbitrary across subjects for the within-subject analysis was only about a 1/3 of what it was across subjects—230ms. Nevertheless, when comparing arbitrary and deliberate activity, we again find an RP in arbitrary and not in deliberate decisions and now plot this in a new panel of Figure 5 (Figure 5C). We also find no relation between the differences in RT and those in RP for the fast deliberate and slow arbitrary ones within subjects. This analysis therefore provides more evidence for there being no relation between RT and RP in our data.

3) To the best of our understanding, the key question here is whether the RT differences between arbitrary and deliberate decisions could explain the absence of an RP in deliberate decisions. We know of no literature suggesting a positive relationship between RT and RP, to which the reviewer appears to insinuate. There is, for example. literature suggesting that Parkinson’s patients tend to have longer RTs and normal or smaller RPs (e.g., Simpson and Kuraibet, Neurology, Neurosurgery and Psychiatry, 1987; Dick et al., Brain, 1989).

4) The leaky DDM accounts for the shape of the ERP at CZ. I cannot find any description of precisely how the plot in Figure 8C was generated. Does it represent the cumulative noise component?

Thank you for pointing out that the explanation of the model was not clear enough. We now go into much more details about the DDM and how Figure 9 (previously Figure 8) was generated in the Results and in the Materials and methods. We also added a new figure, Figure 8, that explains more about how the model works. To answer the reviewer’s question, that figure panel represents the noise and value components activation. Again, this is explained in much more detail in the current version of the manuscript.

Minor Comments:To what degree might the deliberative RP amplitude be impacted by overlapping decision signals. Recent work by my own group has highlighted a P3-like centro-parietal potential that traces perceptual decision formation and peaks at response (e.g. O'Connell et al., 2012, Nat Neurosci; Kelly and O'Connell, 2013, J Neurosci; Twomey et al., 2015, Eur J Neurosci). In Kelly and O'Connell, 2013, we actually observed an RP-like signal evolving around the same time as the CPP and we found that the two signals were impacting one another, something that we were able to resolve using current-source density (CSD) analysis. I wonder what the authors thoughts are on A) the fact that we seemed to observe an RP during deliberative decisions in this case and B) whether an overlapping centro-parietal positivity could partly account for the small size of the RP in their analyses – something that could be verified through application of CSD transforms.

This is an interesting suggestion, which we tested. We conducted an analysis on the CSD transformed data, to see if the lack of RP in the deliberate condition might be explained by the cooccurrence of a CPP component. To our best judgement, the data does not align with such an account. In Author response image 1 we present a figure of the CPP effect obtained in Kelly and O’Connell, 2013, and the topographies of the effects we found in the deliberate condition, both with and without a CSD transformation. These do not reveal any form of an RP, nor a component which resembles the CPP. Thus, though we think this might indeed have been a possible explanation of our results, it was not borne out by the data.

The description of the Schurger study in subsection “Drift Diffusion Model (DDM)” is a little unclear, probably just down to the particular wording used. The meaning of 'non-linear threshold crossing' is not clear to me. 'The crossing of that threshold reflects the onset of the decision in the model, typically leading to action.' 'Onset of the decision' is an ambiguous phrase to use here as it is often used with reference to the time at which evidence accumulation commences. According to these models the threshold instead reflects the completion/termination of the decision. Also the authors state 'Schurger and colleagues claimed, time-locking to response onset ensures that these spontaneous fluctuations appear, when averaged over many trials, as a gradual increase in neural activity.' It is not clear to me whether the authors are aware that according to Schurger this arises because the RP reflects an ACCUMULATION of those spontaneous fluctuations over time as opposed to a moment-by-moment reflection of instantaneous fluctuations.

We thank the reviewer for pointing out this lack of clarity. We are, of course, aware that the Schurger model accumulates spontaneous fluctuations. And rephrased the explanation of the Schurger model to be clearer. As noted elsewhere, we completely rewrote the section of the manuscript describing the model.

LRP analyses. Again these hinge on embracing the null hypothesis. Given the low statistical power and the signs of a larger peak amplitude (and possibly slower build-up) for deliberative decisions, at minimum bayes factor analyses should be provided.

We followed the reviewer’s suggestion and ran a Bayes factor analysis for the LRP. We found BF=0.299, which—according to the standard interpretation—provides moderate evidence against the effect of the decision type on the LRP.

Reviewer #2:

This is an interesting study. The authors challenged the role of RP in internally-generated decision, an EEG signature often linked to volition or action awareness since Libet's work. They contrast deliberate decision (choices did matter) with arbitrary decision (choices do not matter) in a "two-alternative NPO choice" task. Both types of decisions had the similar LRP component, but only arbitrary decision showed reliable RP (at Cz). The authors further simulated an altered version of DDM, showing the absence of RP when assuming a value component underlying the deliberate decision.The experimental design and analysis are rigorous. The authors should be applauded for their effects on assessing several alternative explanations to the absence of RP in the deliberate decision.I have a few comments on current version of the manuscript.1) The modelling procedure is unclear. First, it is unclear why drift rate was the only parameter allowed to change between conditions. Since different task conditions were blocked (see the other comments below), one may argue that it is possible to have the threshold (or other parameters) to vary between blocks, as an adaptive decision strategy. Would model comparison be feasible to identify the optimal parameters?Second, DDM (or many other accumulator models) assumed one-off, rapid decisions without deliberate thinking. In other words, the very first boundary crossing would render a decision, as in the current model. Although the subjects were instructed to respond asap, the current design cannot rule out the possibility of rethinking/change-of-mind in a single trial, which violates the model/simulation assumption. The prolonged RT in deliberate decision support such possibility, that subjects might not rush to a decision, even when a decision threshold was reached.

We thank the reviewer for pointing out that our explanation of the model was not clear enough. In the current version of the manuscript, we rewrote the section describing the model and added another figure to describe and explain it better. We think that in the current version the model and its contribution to the results are much clearer.

As for the reviewer’s specific questions:

1) We think that it is most parsimonious that as few parameters as possible change in the model between the different conditions. Having just one parameter change makes it is easier to understand what exactly changes between easy/hard arbitrary/deliberate decisions. In contrast, having multiple parameters change would add unneeded degrees of freedom to the model that must then be accounted for. Note that we do not claim that this is the optimal or even the only model that could explain our data. However, our model is relatively simple, and it makes some interesting, testable predictions that are borne out by the empirical data.

2) DDMs, race-to-threshold, and similar models are increasingly used to model value-based decision-making in neuroeconomics and elsewhere (e.g., Krajbich et al., Am Econ Rev, 2014). In our model, the value of each decision alternative is reflected by the drift in each of the components of the race to threshold. In this setting, one could—for example—consider a case where one component comes close to the threshold and then decreases away while the other component ends up reaching the threshold first. This might be termed a change of mind in this model. Though, just like a person cannot change their mind after they pressed the button in our experimental setup and task, so does the decision get finalized in the model once a component reaches a threshold. Importantly, changes of mind were not part of our task and we did not design our model to deal with them. Hence, the above is speculative. Last, we think that the longer RT in deliberate trials stemmed from longer deliberation time, reflected in a higher threshold. We see no reason to think that subjects went consistently counter to the instructions and continuously changed their mind, especially as no subject reported having had a problem with changing their mind in the post-experimental debriefing.

2) The arbitrary decision explicitly urged subjects "not to let their preferred NPO dictate their response". This is a strong requirement (asking NOT to follow their preference), compared with previous studies using the typical free-choice paradigm). I wonder if the authors could comment on whether this may inflate the difference between the two types of decisions in the current study. Second, is there any regularity in the arbitrary decision, such as alternating response in consecutive trials?

The reviewer is correct that the instructions for the arbitrary decisions might not be trivial.

However, no subjects reported difficulty in carrying out the instructions in post-experiment debriefing. Further, the behavioral results suggest that subjects were generally able to follow those instructions well. In addition, the EEG results replicate those of Libet and other studies, providing more evidence that our subjects were able to generate arbitrary-like behavior. What is more, under the reviewer’s account it should have been more difficult for us to find differences between arbitrary and deliberate decisions (the same goes for the regularity that the reviewer suggests). This is because, had the subjects exercised their preferences, their decisions would have been more deliberate (e.g., activating networks related to values). And, also, this would have been less like the Libet studies, so we would have been less likely to find an RP, for example. Thus, the fact that we find such clear differences goes against this account.

Regarding the regularities the reviewer mentions, we found no glaring regularities in subjects’ decisions during arbitrary blocks. For example, no subject always chose left or always right, constantly alternated left and right, and so on. However, it is known that humans cannot generate truly random series (e.g., Nickerson, Psychol Rev, 2002; Rapoport and Budescu J Experi Psychol: General, 1992; Budescu and Rapoport, J Behav Decision Making, 1994). So, we do not expect our subjects to pass any strict randomness tests. Such regularity would probably also be found in the Libet experiment and cannot explain our results. Again, if anything this would have made arbitrary decisions more deliberate, making us less likely to find differences between decision types.

3) Different decisions were blocked in the current study. Could the difference in RP be due to a contextual effect? For example, was the early visual ERP comparable between conditions?

Indeed, we decided to group the trials by decision type, because in pilot experiments we realized that subjects are having a hard time switching between decision types on a trial by trial basis. However, we do not think that this could explain the effects. If anything, we think that this should have increased the chances of obtaining an effect in deliberate decisions as well, since subjects do not have to pay attention to selecting the appropriate decision strategy (that is, deliberate/arbitrary), and can focus simply on making the decision.

As for the question about the early visual RP, this information can be found in Figure 6A. There, it does seem that the easy arbitrary waveform diverges from the other 3 conditions. However, as we explain in the Results section, this cannot explain the results we find. And this does not appear to reflect contextual effects either because easy and hard decisions are randomly interleaved for each block. So, subjects do not know whether an upcoming decision will be arbitrary easy or hard before stimulus onset, and so an effect found for easy trials only prior to the trial cannot be explained and cannot account for the RP.

4) Subsection “Differences in reaction times (RT) between conditions, including stimulus-locked potentials and baselines, do not drive the effect”. For the median split analysis to work, my understanding is that no subject had above-median deliberate RT as well as below-median arbitrary RT (i.e., the two groups did not overlap). Could the authors confirm that this was the case? Because inferences on the null hypothesis are important to rule out the effect of RT and other confounding variables on RP, it would be useful to report Bayes factor as well.

We thank the reviewer for this comment. First, we now carry out a within-subject median split as well as the between-subject median split. The within-subject analysis provides the same results. Second, following the reviewer’s comment, we checked for possible overlap and found it for only 3 of the 18 subjects. Hence, we removed those subjects and repeated the analysis. As is evident from Author response image 2, the results stay the same.

**Author response image 2. respfig2:** 

Minor Comments:Figures 8A and 8B could be superimposed over each other, allowing easy assessment of the goodness of fit of RT distributions.

In Author response image 3 is what superimposing the figures looks like. We think that with this format it is harder to understand what is going on and therefore opted for separate panels.

**Author response image 3. respfig3:** 

Reviewer #3:

The present manuscript presents a single experiment aiming at evaluating whether the typically observed early BP onset on spontaneous (arbitrary, purposeless, unreasoned and without consequences…) movements generalizes to deliberate decision making. The authors hence contrasted two decision making situations: in a first condition, participants had to decide, on every trials, to which NPOs the wanted to give money; their choice was effective. In a second condition, while facing the same situation, they were told that their response was consequenceless, since the same amount of money would be given to each NPOs. While a standard BP was observed in the second condition, the authors argue that there is no such BP in the first condition. They hence argue that the results classically observed in arbitrary movement do not generalize to deliberate decision. The early onset of BP has long been considered as reflecting an activation of a response before the actual decision to move was taken, casting doubts on the very notion of free-will. The absence of such pre-conscious marker in deliberate decision is taken by the authors as an argument that, if there is no real free-will in arbitrary movement, this argument cannot be used for deliberate decision.General comments:Although I may generally agree with the authors' conclusions, I'm afraid they are not strongly supported by the facts as they stand. The limitations are methodological and conceptual.While the data show a clear difference in the BP between the two conditions, the core of the authors reasoning is based on the presence of a BP in the arbitrary condition and the total absence of BP in the deliberate one. Indeed, if there were a BP, even of smaller amplitude, in this condition, the argument would immediately fall. There are, however, several points that may challenge this view.

Before addressing the specific points made by the reviewer, we wish to respectfully disagree with the above premise about the interpretation of the data. We do not think that our argument would rise and fall based solely on the dichotomous existence/no-existence of the RP. Rather, we think that the mere fact that the RP is heavily reduced in deliberate decisions challenges previous attempts to generalize from arbitrary decisions to deliberate ones. Thus, while an absence of an RP would perhaps strengthen our claims, a substantial decrease in amplitude is enough for us to draw clear conclusions. At the very least, a considerably reduced RP demonstrates that there are clear differences between the neural processes that drive arbitrary and deliberate decisions. In addition, this means that the RP is more pronounced in arbitrary decisions, which are arguably more driven by random fluctuations, than in deliberate decisions, which are probably based on different neural mechanisms. And so, we think that even without a conclusive null result, the findings of this study would be of importance to the study of voluntary action.

What is more, and as we now discuss at length in the Discussion, we now conducted 6 different kinds of analyses (both using NHST and using Bayesian methods). None of these analyses supported the claim that there exists an RP in deliberate decisions. And all but one of our 6 analyses supported the claim that there is no RP in deliberate decisions, with the sole remaining analysis still suggesting evidence—albeit inconclusive—for the lack of RP. Therefore, we think that— taken together—our results provide clear evidence for an absence of RP in deliberate decisions.

1) Data were high-pass filtered at 0.1 Hz (which corresponds to a time-constant of 1.6 s). This value is way too high for a proper estimate of the BP. As a matter of comparison, the very first measures of BP were even performed under DC conditions, that is without any filtering… At the very minimum, a filter of.01 Hz, or ideally even lower, like.001Hz should be used. Indeed, the BP is a very slow component, such a high filter value has very likely largely deceased the amplitude of the BP. One may argue that filtering has impacted the two conditions in the same way, and hence that any potential distortion cannot explain the results. This is certainly true concerning the difference in amplitude (see however below). But, as indicated above, the rationale put forward by the authors only holds if there is NO BP on the deliberate condition, not if the BP is simply reduced. If, with a more adapted filter, there would have been a BP, the whole argument is invalidated. Second, the response time are much longer in the deliberate than in the arbitrary case. So the time constant of the slow potential might be much longer in this case, and hence much more affected by the inappropriate filter (the control proposed in only partially addressing this issue since i) the comparison is between participants, and ii) at least for the easy deliberate condition, the results are ambiguous).

We thank the reviewer for noticing this point. We first reiterate that we do not think that the rationale only holds if no RP is found in deliberate decisions (see our reply above). More specifically to the question of filtering, our choice of high-pass filter corresponds to some (e.g., MacKinnon et al., 2013; Lew et al., 2012) but indeed not all (e.g., Haggard and Eimer, 1999) studies in the literature. And so, RP was repeatedly found and reported with 0.1 Hz filtering. Yet following this comment, and to make sure our results are indeed not dependent on the filter we used, we reanalyzed the data. A filter of 0.01 is lower than some seminal papers in the field (e.g., Haggard and Eimer, 1999) used a high-pass filter at 0.016). With that filter, we obtained less trials than when using 0.1, as we originally did; but these were enough or analysis. Given the lower number of trials, and that the main question here pertains to arbitrary vs. deliberate (with decision difficulty serving mostly to validate the manipulation), we collapsed the trials across decision difficulty, and only tested RP amplitudes in arbitrary vs. deliberate decisions against each other and against zero. In line with our original results, a difference was found in the RP amplitude between the conditions (t(13)=2.29, p=0.0394), with RP in the arbitrary condition differing from zero (t(13)=-5.71., p<0.0001), as opposed to the deliberate condition, where it did not (t(13)=-0.76, p=0.462). We added this information to the manuscript.

2) Figure 6 presents the stimulus-locked data, supposed to invalidate a stimulus-locked contamination. While I agree that the response-locked BP is not purely a stimulus induced effect, several features of the stimulus-locked averaged deserve comments. First, as can be seen in Figure 6A, a negative shift (CNV-like) is already present BEFORE stimulus presentation in the arbitrary condition, and it seems even modulated by (precued) task difficulty. So, even before participants enter any decision making process related to the relevant choice, one can see differences in the slow negative potentials between the conditions. At minimum, this indicate that the difference observed response-locked might not be specific to the decision period. Furthermore, following the large visual evoked potentials after the stimulus, one can see around 500 ms a first positive component, followed by a negative-going one (starting around 650 ms). This negative-going peaks close to the average RT in the arbitrary condition, and hence likely contribute to the BP. The same activity is present in the deliberate condition (and is actually very similar), but the response being given much later, this early negative-going activity does not contribute to the BP. By itself, this activity, which is not strictly speaking a BP, does create a difference between the two conditions. More generally, the general shape of the late evoked potentials (after 600 ms) are remarkably similar across conditions, which dramatically contrasts with the large difference observed response-locked. This may suggest that the difference observed is, at least in part, simply due the different averaging event is an otherwise very similar signals.

The reviewer’s comment is composed of several points. We respond to them in order.

1) We thank the reviewer for this comment and note that they agree with us that the results cannot be explained solely by stimulus-locked effects. The reviewer’s comment focuses on an apparent difference in the stimulus-locked waveforms: the waveform for arbitrary-easy decisions seems to diverge from all the others from about 500 ms before stimulus onset until stimulus onset. However, and importantly, task difficulty was randomly assigned to each trial within each block, as we explained above. And it was the stimulus that informed the subjects of the decision difficulty in that trial. Therefore, our subjects could not know the decision difficulty in an upcoming trial before stimulus onset. And we see the effect only for arbitrary easy and not for arbitrary hard trials. So, the effect could not be one of precuing in a blocked design. What is more, the response locked waveforms that are the focus of this paper show no such pattern. There, no difference was found between difficulty within each decision type. And the main difference is between the two arbitrary conditions and the two deliberate one, irrespective of difficulty. We now explain that in the manuscript.

2) A few points are worthy of mention here. First, our model predicts similar activity between arbitrary and deliberate decisions when those are *stimulus locked* (Figure 8B). Second, the negative peak to which the reviewer refers (Figure 6A) is around 200-300 ms after the mean RT for arbitrary decisions in the stimulus-locked condition. So, if this peak is what we are seeing in the decision locked condition, we would expect the RP to also peak around 200-300 ms after movement onset. However, the RP peaks 100-200 ms before movement onset (Figure 6B). So, the reviewer’s interpretation of our results would need to account for this 300-500 ms discrepancy in the peak of the RP. What is more, we do not see any of the other components that the reviewer mentions (e.g., the first positive component around 500 ms) in the RP (Figure 6B), providing more evidence against the role of these stimulus-locked components in driving the decision-locked RP. We therefore think that our interpretation of the results is more plausible.

3) The authors focused on Cz to extract the BP. However, there is large literature indicating that the BP is actually made of several sources, some medial in the (pre)SMA, some more lateral in the (primary) motor cortices. Note that, for the authors' rationale to be valid, none of the sources should present a (early) BP. The topographies presented on Figure 3B indicate a clear negative activity above Cz and neighbors in the arbitrary condition, and no activity in the same region in the deliberate condition. But the topography suggests a clear negative activity located more lateral over the left hemisphere. Does this activity correspond to a BP? Since only Cz is shown, one cannot evaluate this possibility.

Again, while we thank the reviewer for this comment, we think their interpretation of our claims is overstated. We do not argue that deliberate decisions are not preceded by neural activity—such a claim would be dualistic in essence (because to assume that for an action to be free, it should not be preceded by neural activity, entails that free actions should not be originated from the brain. If so, from where should they originate? This, in fact, was basis for criticism on Libet’s view; see Wood (BBS, 1985) and Mudrik and Maoz, 2014. Rather, we are simply claiming that the classical RP – which is recorded over Cz – does not generalize to deliberate decisions. And the topography in deliberate decisions does not reveal any sign for such an RP, as it is completely lateralized. We are not aware of any lateralized RP, but if there are such findings that correspond to ours, we will be very thankful if the reviewer could direct us to them, so we could refer to them in the manuscript.

4) In real spontaneous responses, the BP starts much earlier than the LRP. But in the present data set, as can be inferred from Figure 3A, the conditions diverge around -800 ms (roughly estimated). This is basically the same latency for LRP onset. Note that the latency of the LRP (even with all the cautions that may come with this measure…) seems exactly the same for all 4 conditions. So, at the time of BP divergence, the LRP onset indicates not difference in response selection / decision making. Why is the (CZ-)BP more important for the conclusions the authors want to defend? LRP seem to say something very different.

We ran a Maris and Oostenveld cluster-based nonparametric analysis to rigorously test when the RP and LRP diverge from 0, beyond visual inspection alone. The results of this analysis are now reported in Figures 3A and 7. As is apparent, arbitrary decisions diverge from 0 earlier than 1s before movement onset for both decision difficulties. Deliberate decisions never diverge from 0. And all LRP waveforms diverge from 0 only around 500 ms before movement onset. So, according to this standard method, the RP starts much earlier than the LRP.

Regarding the reviewer’s question about why we focused on the RP (from Cz) rather than the LRP, there are several reasons. First, the RP (from Cz) is the component that is most commonly associated with the Libet paradigm. Second, the LRP, being more lateral, is more directly over the left and right motor cortices while Cz is above the SMA. As such, the LRP is typically taken to reflect more motorrelated brain activity and the LRP more general preparation. Our task was specifically constructed such that the motor output of arbitrary and deliberate decisions would be the same. Hence, seeing the same LRP in all conditions encourages us, as it seems to provide evidence that the neural activity associated with the movement in all decision types is very similar.

5) A further issue concerns the modeling part. First, it is very unclear how the fit was performed. In subsection “Drift Diffusion Model (DDM)”, it is indicated that "We fit our DDM to our average empirical reaction-times, which were 2.13, 2.52, 0.98 and 1.00s for the different conditions ". This sentence suggests that only the grand mean were used, not the whole distribution of RTs. Besides the fact that this is in complete deviation with the overall logic of fitting DDM which aims at capturing RT distribution shape and errors at the same time, a fit performed on averages only is largely under-constrained. As a matter of fact, although there is no information about quality of fit, comparison of Figures 8A and B indicate that the empirical and predicted distribution largely differ, especially in terms of spread (i.e. variance). The obtained parameters are also weird and likely invalid: Table 1 indicates that, in the deliberate condition, the estimated drift is actually higher for the hard condition than for the easy one. This does not make any sense, and is disagreement with the empirical RT that are, as expected, longer for the hard than for the easy condition. Furthermore, one reads "[…] The model was further fit to the empirical consistency ratio […]" (emphasis mine…). "Further" suggests that the fit to the mean RT and the consistency ration were not performed simultaneously? Is that the case? Please clarify.Besides these fitting problems, I have conceptual issues. Threshold crossing in the arbitrary condition is supposed to be triggered by noise only. In standard accumulation-to-a-bound models, noise is classically assumed to be gaussian noise with mean 0 and a standard deviation s. For "noise" to hit the threshold one needs to assume a very high s. Or to assume that the mean is not 0… But in this case, this is not noise, this is a drift. Actually, Table 1 indicates: the "drift" in the arbitrary condition is much higher than in the deliberate condition. But what is the meaning of this "drift" if there is no decision, and the response is triggered by random fluctuations of noise? In contrast, what does it mean that a deliberate decision was reached with an information accumulation equals to 0? How can a threshold be reached if there is no information accumulation? If it's simply noise, why is different from the arbitrary condition? Furthermore, if random noise can hit the bound early in the accumulation process, if noise is 0-centered, the probability to hit the threshold decreases with time. So late responses are unlikely to be triggered by noise. So, this whole fitting part is largely based on inconsistent assumptions.

First, as we indicated elsewhere, we now rewrote the section devised to the description of the model. And we think it is much easier to read and understand. Nevertheless, we broke down the reviewer’s comment into sections and respond to them one by one below.

1) We fit our model using the same method that Schurger et al., 2012, fit their model, as we base our model on theirs. Though we added constraints based on the consistency scores (Figure 2) to drive the differences in parameters between the congruent and incongruent DDMs.

2) We thank the reviewer for their careful reading of our manuscript and for spotting errors in Table 1. We now corrected these errors in this table in this version of our manuscript. The correct magnitudes in Table 1 have lower drifts for deliberate hard than for deliberate easy.

3) All that was meant in this sentence is that we fit the model to both the RTs and consistencies. We also explained that we fit the RT and consistency together in the Materials and methods, and in the Results. We thank the reviewer for alerting us that this sentence might be confusing. And we changed “further” to “simultaneously” to clarify this.

4) As the reviewer indicates, the noise in our model has a mean of 0, as is standard, and we employ a drift. In this we follow the Schurger et al., 2012, model. It is certainly possible to model our data by changing both the drift and the threshold. But changing two model parameters instead of one is less parsimonious and adds unneeded degrees of freedom that must then be accounted for. We therefore opted to change only the drift parameter among the conditions. The values of the decision alternatives in our model are designated by the drift—separately for each decision type. Hence, for deliberate decisions, the congruent cause had a higher value and thus also a higher drift rate than the incongruent cause and its associated DDM. For arbitrary decisions, the values of the decision alternatives mattered little and this was reflected in the small differences, if at all, among the drift rates.

5) A drift rate of 0 indicates that the threshold would be reached only if the noise parameter would carry out a random walk to the threshold (up to the leak pushing it back toward the baseline). So, the threshold can be and is reached. Note, however, that the 0 drift in Table 1 exists only for the incongruent decision alternative in easy deliberate decisions. For those decisions, the inconsistent choice should almost never win the race to threshold against the consistent choice. And, indeed, this alternative is selected only about 1% of the time empirically as well as in the model. We therefore do not agree that the reviewer that the fitting part is largely based on inconsistent assumptions.

My general feeling is that the authors provide further arguments against the idea that BP can be used as a marker of volition, but this is not new. In contrast, they provide few (if any) new arguments in the discussion of whether there is free-will or not.

As discussed in the Introduction and Discussion, the Libet results have long been claimed to provide evidence against the existence of free will. However, for that claim to hold water, these results must be generalizable from the purely arbitrary setting of raising a hand for no reason and purpose to the deliberate decisions that are the typical focus of the free will debate. Whether the Libet results generalize from arbitrary to deliberate decisions is therefore key to their applicability to the free will debate, especially when claims are made regarding moral responsibility. To the best of our knowledge, our results are the first to directly compare arbitrary and deliberate decisions. As such, our results are an important neuroscientific contribution to the debate on free will.

Minor comments:– Bayesian analysis. The core idea of Bayesian ANOVA is that the values are indicative, but no hard threshold should be used. Hence, considering that a BF of.332 (that is a probability ratio > 3) is an argument for no effect is wrong. Actually, in this uncertainty zone, one cannot conclude anything, and this value is NOT an argument for H0.

we thank the reviewer for making this point, though we again respectfully disagree. What one could claim based on the BF, is that H_0_ is 3 times more likely than H_1_. Is this not an argument for H_0_? In fact, this is commonly taken in the literature as moderate evidence for the null result, which is exactly how this is described in the manuscript. Importantly, we emphasize there that the evidence are inconclusive to moderate, given the two analyses we conducted.

– EEG results. Authors report a main effect of decision type. But is there an interaction with difficulty? If not, the following t test are not completely valid (although this does not change the conclusions…)

unless we are mistaken, one is not allowed to conduct t-tests following an insignificant interaction when these t-tests are aimed at exploring the source of the interaction. Yet the t-tests we conducted are not aimed at that (we have no claim about the differences between decision types at the different levels of decision difficulty variable, or vice versa). Rather, these are t-tests against zero, and so they are not tested in the ANOVA (or the interaction) to begin with.

– Figure 5B: the data plotted in this figure puzzles me. The y-axis plots the difference (deliberate – arbitrary). 12 points out of 17 (?) are actually negative, indicating that deliberate was more negative than arbitrary. How can this be?

We thank the reviewer for spotting this error. It has now been fixed. The correct plot is included in Figure 5B.

– Subsection “EEG Results: Lateralized Readiness Potential (LRP)”, the argument that LRP amplitude difference is due to the reference used is incorrect: LRP measure is reference-free. Indeed, the formula is (C3 -C4)r + (C4-C3)l. But actually all electrodes values should be written as (C3-ref) and (C4-ref). It becomes then obvious that the reference annihilates and does not intervenes in the computation.

We thank the reviewer for pointing this out. We removed this sentence that refers to a minor point in our manuscript.

– Modeling. The authors sometimes refer to a DDM, sometimes to a LCA, sometimes to a race. Those are very different architectures (although they can be related, see Bogacz et al., 2006). Please clarify what is really used.

We now clarify this in our revised description of the model.

– Experimental procedure. It is never indicated how the response were given! From the description (response keys) and EEG analysis (computation of LRP), one can infer that the response were given with the left and right hand, right? But with which finger?

We thank the reviewer for the thorough read of our manuscript. This information was mistakenly omitted from the original manuscript. Though we explained that subjects were asked to place their fingers on the Q and P keys, we did not explicitly say that it was the right and left index fingers. We added the missing description to the manuscript.

– Subsection “Experimental Procedure” and other: besides providing the t or F values, please provide the behavioral data. For example, please provide error rates (even if they do not statistically differ). Furthermore, considering that a difference with a p value equal to.09 as not significant, is a bit strong.

We now added the behavioral data, as the reviewer requested. However, we do not think there is any problem with considering a value of 0.09 as insignificant. If anything, more claims have been made in recent years against considering p values lower than 0.05 but higher than 0.01 or even 0.005 as significant (e.g., Benjamin et al., Nature Human Behavior, 2018), as means to reduce false discoveries. And so, we stand behind our referral to the 0.09 value as reflecting insignificance.

[Editors' note: further revisions were suggested prior to acceptance, as described below.]

All three reviewers agree that you have made substantial efforts to revise the manuscript in light of the initial round of comments and this clearly took time and effort on your part. However, I'm afraid we are all in agreement that some further revisions would be strongly recommended prior to publication. The remaining issues boil down to two substantive points:First, we are still not convinced that the present results call for any alteration in our conception of free will. The key contribution of the paper seems to be to show that the RP may be specialised for arbitrary action selection and this seems worth reporting. However, beyond suggesting that the RP might not be a universal marker of the 'urge to act', the authors have not yet made a sufficiently clear case that their findings call for any change in how we think about the Libet findings or free will in general. Our recommendation at this point would be that the authors reframe the paper to focus more specifically on the RP and how the current findings bear on functional accounts of this signal.

Thank you for raising this issue. The discovery that RP onset precedes the reported onset of the intention to move was understood by Libet and colleagues—and has been widely understood since—to show that human decisions are not free. The discovery was thought to show either (a) that our decisions to act are unconscious, (b) that our decisions are conscious but are not the causes of our bodily movements, or (c) that our decisions are fully determined by brain activity that precedes them. (The references are too many to list as the original Libet study, (Libet, Gleason et al., 1983), has been cited thousands of times. A small sample of references to the above might include (Libet, 1985, Haynes, 2011, Hallett, 2016, Haggard, 2019)). For various philosophical reasons, deriving from accounts of the necessary and sufficient conditions that an act meets when it is free, (a), (b) and (c) have all, at various times, and by various thinkers, been thought to threaten the freedom of our actions (Again, Mele, 2009, Roskies, 2010, Sinnott-Armstrong and Nadel, 2011, Maoz and Yaffe, 2015 are but a small sample).

Our results here, however, show that any claims about this matter that can be made on the basis of the RP might be unwarranted for deliberate decisions. Given that deliberate decisions are the only ones for which we hold people responsible, and given that one of the reasons freedom has always been thought of interest is because it is necessary for responsibility, our study here has direct and immediate implications for the study of freedom.

Second, the reviewers also have significant outstanding concerns regarding the modelling. At minimum we would suggest clearly flagging the limitations of the adopted approach in the Discussion.

Thank you for this comment. We reconstructed the model and reran all simulations following the comments of the reviewers below (see Modeling subsection of the Results; see also Model and Simulations subsection of the Materials and methods). We also devote more space in the Discussion to the modeling. Importantly, the results of the new model are essentially the same as the previous model. We therefore think that the model, its results, and the conclusions we draw from the modeling are now clearer in the manuscript. Please see specific responses to reviewer comments regarding the model below.

Reviewer #1:

I think the authors have, by and large, done a good job of addressing the methodological concerns raised by the reviewers. Importantly the authors assert that they are not claiming that the RP is actually absent but rather significantly diminished during deliberative decisions. I have a couple of outstanding concerns.

We thank the reviewer for his comments that we did a good job addressing the methodological concerns raised by the reviewers. But we would like to clarify that we claim that *at the very least* the RP is much diminished, and we think that the most plausible conclusion from our analyses is that the RP is absent. We now discuss this and what a strongly diminished RP would entail in more detail in the Discussion.

I am still confused about the overarching premise or rationale for this study. It seems to me that the findings of this study boil down to showing that the RP is smaller for deliberative vs arbitrary decisions but I am not yet convinced that this has any bearing on our understanding of free-will or even on the significance of Libet's original reports. Libet showed that a neural signature of action preparation preceded conscious awareness of the decision to arbitrarily act. Perhaps that same signal (RP) is absent during deliberative decisions (and this is worth reporting) but that tells us nothing regarding the role or timing of conscious processes in this context. Of course it goes without saying that arbitrary and deliberative decisions involve distinct cognitive elements and, consequently, will activate some distinct brain areas. We know that several other signals that reflect action preparation (e.g. LRP, β-band desynchronisation) and evidence accumulation (e.g. P300, LIP spiking activity) precede explicit decision reports by substantial amounts of time.

As we make clear in the manuscript, our aim was not to question Libet’s original report. In fact, we replicate the central finding that arbitrary decisions are preceded by the RP. Our concern is with the further question of whether conclusions that one might reach about the lack of freedom of decisions preceded by the RP can be reached, also, about deliberate decisions. Our results suggest that they cannot, and we explain why this has clear bearing on the question of free will. We believe that this point is now very clear in the manuscript (much of the Discussion is devoted to this).

The authors seem to suggest that Schurger's model presents a challenge to Libet's interpretation: "A further reason to expect such differences stems from a recent computational model, which challenged the claim that the RP represents a genuine marker of unconscious decisions. Rather, the model suggested that the RP might reflect the artificial accumulation, up to a threshold, of stochastic fluctuations in neural activity"I am still struggling to understand what difference it makes, in terms of our understanding of free will, if RP reflects 'action preparation' or 'noisy evidence accumulation' – wouldn't both processes would reflect the emergence of an 'urge to act'? Moreover, the authors seem to imply that Schurger's process must be conscious but I see no reason to make such an assumption. I think that particular sentence really captures my discomfort with the current framing of the findings.

We now describe the Schurger model and its relation to the Libet results in much more detail. We think that our explanation will help the readers understand why the Schurger model poses a challenge to the Libet interpretation. Briefly, the Libet experiment suggests that the RP starts before the onset of the conscious intention to move. It assumes that the RP is a ballistic process and thus its beginning marks the onset of a decision (at least an unconscious one). This was then taken to mean that the decisions are unconscious or that the conscious decisions are inefficacious. However, if the RP is not a mark of an unconscious decision but rather an artifact, its start preceding the conscious decision is of little importance (one place, among many, where this is discussed is Haggard, 2019). Also, we do not imply or think that the formation of the RP (as formulated by Schurger's model) needs to be a conscious process. We thank the reviewer for raising this point and removed the sentence that might have evoked this false impression.

Another way of putting it is that I am not clear on how exactly people were making generalisations to deliberative decisions based on the RP specifically. I get the impression the authors are trying to make the case that the RP has been thought of as the sole signature of pre-conscious decision making/action preparation in the brain but is that really a widely held view, again given the literature on decision making?

The RP is not the *sole* signature of unconscious decision-making. But it is certainly held to be a very prominent signature of unconscious decision making (Libet, Gleason et al., 1983, Libet, 1985, Haggard, 2008, Roskies, 2010, Hallett, 2016, Haggard, 2019 are just some of the very many possible references). And so, while it might be accompanied by other mechanisms, the underlying assumption has been that the RP also characterizes deliberate decisions.

Among other things, this can be learned from the fact that findings related to the RP have been used as basis for arguments about moral responsibility, which clearly pertain to deliberate decisions alone. This is now also clarified in the manuscript. In addition, as the RP has been viewed as such a prominent signature of pre-conscious decision-making, problems with generalizing it from arbitrary to deliberate decisions put the onus on those who wish to use other features to demonstrate that those generalize. We now discuss this too in the manuscript.

I think that perhaps the authors should reframe the paper to focus more narrowly on the apparent domain specificity of the RP. On the same point I think the title of the paper may be too broad given the almost exclusive focus on the RP. I am very much open to being convinced/correct on all of the above points but I feel that currently the paper is rather confused and confusing as regards the relevance of the findings to free will.

We thank the reviewer for alerting us that the manuscript might need to further explain its relevance to the free-will debate and many of the corrections and additions to the paper were made to clarify this. However, we respectfully disagree with the reviewer regarding the need to reframe the paper to focus more on specifically the RP. The paper indeed probes the RP (though notably also the lateralized readiness-potential, LRP), but it does so as part of a large body of literature which used the RP to study volition and the relations between subjects’ conscious decision and the underlying neural activity. Virtually every scientist who has replicated Libet’s result has made the further claim that the findings shed light on the freedom of human decisions. Critically, in many cases this claim related to all types of decisions, and not just arbitrary ones, as we now explain in the manuscript. We therefore do not think that we are overstepping in pointing out the fallacy of this inference.

My only other comment is that I do not agree with the authors insistence on only allowing one DDM parameter to vary across conditions on the grounds of parsimony. Although this is indeed a common approach in behavioural modelling, there are formal model comparison procedures available that would objectively identify the model that provides the best balance between parsimony and goodness of fit. Personally I would probably be ok with the authors acknowledging this in their Discussion as I do not view the modelling as a critical part of the story.

We thank the reviewer for this comment. Following comments from the reviewers, including this one, we reconstructed and re-simulated the model. In particular, we now fit the entire RT distribution of the model (1,200 samples) to the empirical RT distribution (1,200 samples) and, at the same time, we fit the modeling and empirical consistency rates. So, we now fit 4 parameters per condition (16 overall) to these 1,201 points and not just the drift (see Table 1).

Reviewer #2:

In this revision, the authors addressed some of the concerns in my last review. However, although the MS now has more detailed description of the model and simulation results (which was missing in the previous version), there are some additional issues.1) The model used 11 parameters, 8 drift rates, a scaling parameter (1.45), threshold and decay to fit 8 data points (4 mean RT and 4 choice probability). The result is a largely unconstrained model that does not describe behaviour accurately. In the response letter, the authors showed model simulation overlayed with empirical data, and there is a large discrepancy in the fit of arbitrary condition. If the authors are determined to present current modelling results, as a proof-of-concept that the model can provide qualitative RP patterns, the limitation (that it does produce satisfactory quantitative fit to RT distributions as in other applications of DDM) needs to be acknowledged in Discussion.

As we explain in our last response to Reviewer 1, following the important comments made by the reviewers, including this one, we reconstructed and reran the model. We now fit the entire RT distribution and empirical consistency rates. Hence, we fit 4 parameters per condition (16 overall) to the 1,201 points of the reaction time and consistency. (The scaling parameter, 1.45, does not take its value from an optimization procedure. Instead, it is just calculated from the empirical data. But, even if it did, it would be 17 parameters for 1,201 points.) Our modeling procedure is therefore no longer under-constrained. Also, as is apparent in the new Figure 9A—now overlaid—the fit of the model’s RT and consistency to the empirical RT and consistency is rather good, with an average error of just 0.036, or 3.6%, across all the conditions. So, the model describes the behavior rather well.

2) Subsection “Model and Simulations”. "Using a parameter sweep.…". Please provide details on the fitting procedure. Was any optimization algorithm used here? If so, what was the cost function? Was the model fitting performed on averaged data or individual data?

We are glad to give more details. We ran an exhaustive grid search on the averaged data across participants. The optimization algorithm, the cost function, and the procedure in general are now explained in much more detail.

3) In the response letter to my point 1, "we think that the longer RT in deliberate trials stemmed from longer deliberation time, reflected in a higher threshold." This is confusing as the same threshold (0.15) across conditions was used in the paper?

We thank the reviewer for pointing out the mistake in our answer. We had meant to write that the longer RTs in deliberate trials stemmed from lower drift rates. Regardless, we now fit 4 parameters and not just the drift rate (Table 1). So, we think that this discussion is no longer relevant to our modeling.

4) Figure 8A. There are two independent accumulators in the "noise component". From my understanding of the model description, in arbitrary decisions, SMA activities were reflected in the traces of the winning accumulator only. If that is the case, why is the activity of the losing accumulator in the noise component not taken into account, as sculp EEG activity would not be sensitive or selective to one accumulator vs. the other. The same issue holds for the deliberate decisions, if both accumulators in the noise component do not dictate decision (as threshold was set to infinity), which accumulator (or both?) was representative of SMA activity? Please clarify.

We thank the reviewer for this comment and agree with it. Hence, in the new modeling algorithm we averaged the activity of the winning and losing accumulators as the reviewer suggested. This did not much change our results.

Reviewer #3:

In the first round of review, several questions were raised. The authors did a very good job in answering most of them. It is now clear that a RP is observed in the arbitrary condition, but not in the deliberate one. There are still, however, two points that remain unclear to me.

We thank Reviewer 3 for acknowledging the improvement of the manuscript, and for agreeing with us that the data clearly show that the RP is observed in the arbitrary condition but not in the deliberate condition (cf. our reply to Reviewer 1). We address the remaining points below.

1) What are the theoretical consequences of these results?As said above, the authors convincingly show that RP is absent in the deliberate condition. Although this is not entirely new (some previous studies have reported absence of the BP before voluntary movement depending on the context), this report certainly adds to our understanding of the RP, its origins and functional interpretation. However, as can be read in the Abstract, the goals of the authors is much more than that. But I am not sure about the real theoretical impact of the results. Let me try to explain my trouble.Libet's original report was that the RP starts before the conscious decision to move. It was thus argued that our intention to move is rather a consequence, not a cause, of the preparatory brain activity. This was taken as an argument that "free-will" is an illusion.Here, the authors report that there is not RP in the deliberate condition. Hence, the nature of the decision differs between the two conditions. So, what do exactly the authors conclude from that?– the Libet's argument vanishes for deliberate condition, hence there is no evidence against free will in this context. But do they think it still holds for arbitrary ?– since RP might simply accumulated random noise, it is not an indication of voluntary movement decision, and hence Libet's argument is wrong even for arbitrary movement?– if deliberate decision are made on another region X, it might still be that activity in region X starts before conscious detection, but this remains to be explicitly studied.– something else?I must confess that I cannot really get the real conclusion the authors want to defend, and they should try to be more explicit on what these results do really imply, and what they do not, on this free-will debate.

Our primary conclusion is that strong claims about the lack of freedom of decisions preceded by the RP are unsound when asserted about deliberate decisions. This follows immediately from our results and has direct implications for the free will debate. This is now further clarified in various places in the Discussion of our paper. However, we also hold, and now make clear in the paper, that it may also be unsafe to conclude that, where the RP precedes action, the action is unfree. The reason is that our model is compatible with and supportive of Schurger’s findings, which provide strong reason to doubt the validity of this inference.

2) Modeling.Although many points have been clarified in this new version, some still remain a bit unclear.To account for the choice situation, the authors modified Schurger et al. original model (who contained a single accumulator), and implemented two accumulators racing for the response. First, one may wonder why they did not choose the standard competitive version of the leaky accumulator (Usher and McClelland, 2001).

There were two reasons that we opted for the race-to-threshold model over the DDM with dual, upper and lower bounds. The first was that we think such a race-to-threshold model is more biological on the neuronal level at least. Work by Shadlen and others has demonstrated decisions (in random-dot motion for example) that are reached when neuronal firing rate reaches a certain threshold (de Lafuente, Jazayeri et al., 2015). But for a dual bound model, postsynaptic activity would need to differentially depend on the neuron either reaching a high-enough firing rate or a low-enough one. This seems more difficult to realize biologically than a race to a threshold between two neurons or regions based on their firing rate.

Second, if we were to use a model with a dual bound, we would end up with activity that either ramps up toward the upper bound or ramps down toward the lower one. Both of those situations would have had to create the RP, similar to the Schurger case. So, we would have had to assume some neural mechanism that would take those increasing and decreasing RPs as inputs and output a single RP, using for example some kind of absolute value function. This again seems to us less likely to be realized biologically than an implementation of the race-to-threshold model. We now discuss this point in the manuscript.

Second, it is not completely clear how the averaged data on Figure 9C was actually computed. I guess that, for each "trial" the winning accumulator was chosen (left or right) and all the traces of the winning accumulators were averaged. However, if two accumulators are racing within "SMA", the real simulated activity of SMA should be the sum of the two accumulators, not only the winning one. I'm not sure how this would modify the results, but for coherence, this is the way "SMA" activity should be evaluated in the model.

We thank the reviewer for this comment and agree with it. A similar one was made by Reviewer 2 (Comment 4). As we note in that response, in the new modeling algorithm we averaged (or took half the sum of) the activity of the winning and losing accumulators as the reviewer suggested. This did not much change our results.

Third, I still don't understand how the fit was performed. It is said "[…] we fit our DDMs to our average empirical reaction-times […]" (emphasis mine). It the fit was indeed performed only on the averages, this is non standard, and highly under-constrained. Such model are normally fitted to the whole response times distributions. Furthermore, there is no quantitative assessment of the fit quality. Comparison of Figure 9 panels A and B, suggests that the fit was not very good, especially in the arbitrary condition. Fourth, Figure 9 only shows the activity of "SMA". However, there is another actor, which is never shown: region "X"… In the deliberate condition, the decision in made base on the activity of this "region", but its dynamics is likely very different from "SMA" one. It would be of interest to see the accumulated activity of this region "X" in both the deliberate and arbitrary conditions. A last question concerns what "SMA" is doing. In the arbitrary condition, its accumulating random, spontaneous noise. But why is it not doing the same in the deliberate condition (in addition to accumulation in region "X")? Do the authors assume a form inhibition of region "X" on "SMA" to prevent it from accumulating? This part is bit too "magic" and an explicit, mechanistic, explanation would be useful, instead of just claiming that accumulation is done differently as a function of the context/choice (which is vague). Somehow relate to this last point, there seems to be a bit of (simulated) accumulated activity in the deliberate conditions in "SMA". Where does it come from?

We thank the reviewer for making this point. As we explain in our previous replies to Reviewers 1 and 2, who made similar points, we reconstructed the model following the good comments we received (including this one). In particular, we now fit the entire RT distribution of the model (1,200 samples) to the empirical RT distribution (1,200 samples) and, at the same time, we fit the modeling and empirical consistency rates. So, we fit 4 parameters per condition (16 overall) to these 1,201 points. Hence, our model is no longer under-constrained. This is explained in detail in the Modeling sections of the Materials and methods and Results and discussed in the Discussion. Note that the results of the model stay essentially the same, though the fit to the RT distributions is better.

Detailed responses to the comments by the reviewer now appear in the parts of the manuscript above. Briefly, one reason that we do not focus on Region X is that we assume that the activity there is similar to that of the SMA only that it is an unknown region to which we do not have access (at least using EEG). Hence, the RPs we show in Figure 9B are what our model predicts we would pick up in electrode C_z_ over the SMA. The activity of Region X is shown in Figure 8B (in green), where it is demonstrated how, in deliberate decisions, it imposes an early stop at different heights on the SMA component. However, another, perhaps more important, reason that we do not dwell too much on Region X is that the DDM model we chose for it was based on convenience and simplicity. The central result for us, the trend instead of RP in the SMA for deliberate decisions, only requires that the onset of deliberate decisions would remain statistically independent from threshold crossings in the DDMS of the SMA.

Besides, I have some more specific pointsIntroduction section: "[…] Thus, one could speculate that different findings might be obtained when inspecting the RP in arbitrary compared to deliberate decisions. […]" is still very unspecific

We thank the reviewer for bringing this to our attention. We have accordingly modified that sentence and that paragraph. We also more generally added further explanations about our motivation and hypothesis.

Introduction section: […] Demonstrating no, or considerably diminished, RP in deliberate decisions would challenge the interpretation of the RP as a general index of internal decision-making.[…] Ok, but the fact that it is not a "general index" does not, de facto, solve Libet's problem: even if reduced, if RP start before conscious decision, the argument is still valid.

We thank the reviewer for this comment. We now discuss this in some detail in the Discussion. Briefly, we think (and the reviewer agrees) that the most plausible interpretation of our results is that the RP is absent in deliberate decisions. However, even if one takes the RP to appears as only diminished in deliberate but not arbitrary decisions, that goes against its interpretation as reflecting simple motor preparation. This is because the motor output is the same in both decision types. What is more, our model predicts a slow trend in deliberate decisions that might resemble a heavily diminished RP.

[…] More critically, it would question the generalizability of studies focused on arbitrary decisions to everyday, ecological, deliberate decisions […] This is indeed critical for the functional interpretation of the RP, but this sounds partly orthogonal to the free-will debate.

We disagree with the reviewer on this point. Much of the free will debate focuses on deliberate decisions, especially when the debate pertains to moral responsibility. We now explain this further in the manuscript. In fact, some philosophers define free will as the capacity that allows one to be morally responsible (e.g., Mele, 2006, 2009). And ascribing moral responsibility makes sense only for deliberate decisions.

Subsection “EEG Results: Readiness Potential (RP)” paragraph two: why are the student tests corrected for multiple comparisons, since only 4 are performed? Does it mean that the authors performed t-tests for all time-points? In this case, a multiple comparison is, indeed, necessary. But only one t-test is reported! Please clarify.

We did not carry out the t tests on all time points. However, to our understanding, one should correct for multiple comparisons to make sure that the probability of a type-1 error will not be inflated. When performing one t-test, we have 5% chances to obtain a false positive error. When performing four t-tests, this jumps to ~20% (e.g., Miller, RG. Simultaneous Statistical Inference 2nd Ed. Springer Verlag New York, 1981). Thus, we corrected for the performed t-tests which were conducted on the averaged activity. We now clarify this point in the manuscript. Naturally, by correcting for multiple comparisons, we made it harder on ourselves to achieve a statistically significant result. So, our corrected results certainly hold if we remove the correction. As for the non-significant results in the deliberate conditions, these results too remain insignificant without the correction. So, correcting did not affect the results we found.

In the same section: BF = .332 is not a serious evidence for no effect. The authors should not take 3 (or.33 depending on how we compute it) as a threshold above which an effect would become "significant".

As the reviewer notes by putting *significant* in quotation marks, Bayesian statistics is not about significance. It is rather about the likelihood of one model (H_1_) over another (H_0_). It is indeed a mistake to consider 0.33 as a ‘significance threshold’, akin to the 0.05 convention in NHST. We consequently did not use the term *significant* when referring to this BF. Instead, we wrote that there was evidence for the RP being absent in deliberate decisions (that is, evidence for H_0_). This is in agreement with the accepted convention in Bayesian statistics. According to this convention BF < 0.1 implies strong evidence for the lack of an effect (i.e., the data are at least 10 times more likely to be observed given H_0_ than given H_1_). A 0.1 < BF < 0.33 provides moderate evidence for the lack of an effect. 0.33 < BF < 3 suggests insensitivity of the data (anecdotal evidence for the lack or presence of an effect, for 0.33 < BF < 1 or 1 < BF < 3, respectively). 3 < BF < 10 denotes moderate evidence for the presence of an effect (i.e., H_1_). 10 < BF < 100 implies strong evidence. And BF > 100 suggests extreme evidence for the presence of an effect (Lee and Wagenmakers, 2013. *Bayesian Data Analysis for Cognitive Science: A Practical Course*. New York: Cambridge University Press.). Nevertheless, following this comment, and to make our description more accurate, we added the word ‘moderate’ to the sentence.

Subsection “EEG Results: Readiness Potential (RP)” paragraph four: while a BF of.09 is indeed an argument for no effect, a BF of.31 is not really.

We refer the reviewer to Reply 7 above. A BF of 0.31 means that H_0_ is more than 3 times more likely than H_1_, which is considered moderate yet conclusive evidence in favor of H_0_ (as opposed to inconclusive evidence).

Subsection “Differences in reaction times (RT) between conditions, including stimulus-locked potentials and baselines, do not drive the effect”: the authors discuss at length the potential impact of (or absence of) the chosen baseline. Besides all the rather indirect arguments based on different baselines (none is immune of criticisms), one analysis suffices to invalidate this argument: the slopes of the linear regressions are, by construction, independent of baseline (only the intercept is). So the fact the slopes more negative for arbitrary than for deliberate condition is a strong and not disputable fact, much stronger than all the baseline changes.

We sincerely thank the reviewer for highlighting this point and agree with his view. We now added this point to the manuscript.

Figure 6: I'm personally convinced that the RP observed on the arbitrary condition is not (only) a contamination by Stimulus-locked activities. However, the arguments based on Figure 6 are pretty weak. Indeed, in the time windows from about 600 ms to 1200 ms, a negative ramp is observed for all 4 conditions. The response is given in this time windows for the arbitrary condition, but for the deliberate one. So, this stimulus-locked negative ramp likely contributes to the RP.

We are glad to learn that the reviewer has been convinced by our results and analyses that the RP observed on the arbitrary condition is not (only) a contamination by stimulus-locked activities. Nevertheless, to the reviewer’s point, the observed negative ramp at the descriptive level seems to start at 750 ms before movement onset for the hard, arbitrary condition, and at 850 ms for easy arbitrary conditions. And response onset was at about 1s, leaving a time difference of 250-150 ms between the onset of this proclaimed ramp. The RP, on the other hand, started ~1.2 s before movement onset, so it is less likely that the two represent the same component. What is more, the amplitude of the RP was 2 µV, while that of this ramp was 0.8-1.2 µV, at the most. We therefore think that this pattern—which is different in both amplitude and time—cannot explain the RP.

Minor Comments:Subsection “Experimental Procedure”, fourth paragraph: "right and left index finger" -> "left and right index finger" to be more consistent with the rest of the text.

Thank you. This has been corrected.

Subsection “Experimental Procedure”, final paragraph: "[…] We wanted to make sure subjects were carefully reading and remembering the causes also during the arbitrary trials to better equate memory load, attention, and other cognitive aspects between deliberate and arbitrary decisions […]" Although adding a task is a good idea, it may sound a bit naive to say that the task were equated. For example, in the arbitrary decision, there is a short term memory component that is not present in the deliberate.

Thank you. We now clarify further that we do not think that the tasks can be completely equated.

Subsection “ERP analysis”: "offline to the average of all channels,": including mastoids and nose? I guess not… At least, this should not be the case! Please clarify.

We now clarify that the averaging does not include external electrodes. We thank the reviewer for raising this point.

Subsection “ERP analysis”: " which subjects pressed the wrong button " What is a wrong button? Inconsistent response? and in arbitrary condition?

This does not refer to inconsistent responses. Rather, it refers to pressing a button that is not one of the designated response buttons (that is, not the <Q> or  buttons). We explain this better in the current version of the text. Thank you.

Subsection “ERP analysis”: "Channels that consistently had artifacts were replaced using interpolation (4.2 channels per subject, on average ": Although this is the range acceptable by some standards, I personally find this value high. Furthermore, could we have the range of channels interpolated?

We thank the reviewer for catching this mistake. That number did not take into account subjects for whom no interpolation was made (i.e., with 0 channels interpolated). We corrected that average—to 1.95—and added the range (0-6) as requested.

Subsection “Statistical Analysis”: The authors took 1 point over 10 to re-sample the signal. However, re-sampling requires appropriate anti-aliasing filtering to avoid signal distortion. Data were acquired at 512 Hz; Biosemi anti-aliasing filter, if I'm not mistaken, should be around 100Hz. Since no other low-pass filtering was applied, the data contains signal up to 100Hz. Hence, (re)sampling at 50Hz a signal whose max frequency is around 100Hz is extremely problematic. At minimum a 25Hz low pass filter should have been applied… It is very hard to anticipate what would be the impact of such aliasing (especially since the activity of interest is low frequency), but this should be corrected to avoid having incorrect practices published.

We thank the reviewer for making this good point and agree with it. We reran this analysis with a 25 Hz low-pass filter and the results remained the same. This is now reported in the manuscript.

Subsection “Model and Simulations”: "used Δt = 0.001, similar to our EEG sampling rate": if sampling rate is 512Hz, dt should be 0.002.

We removed the “similar to our EEG sampling rate” from the sentence and thank the reviewer for catching this mistake. To be clear, rerunning the modeling with Δt = 0.002 does not change any of the essential modeling results.

[Editors' note: another round of revisions were suggested prior to acceptance.]

The manuscript has been substantially improved. As part of this peer review trial, as reviewing editor I am required to indicate whether all of the reviewer comments have been fully addressed or if minor or major issues remain. There are some minor comments arising from this latest round of reviews that I thought I would give you the opportunity to address prior to finalising the 'Accept' decision so that I can then potentially indicate that all reviewer concerns were addressed. I have outlined these below. If you prefer to expedite the publication and not address these comments you can let me know.1) Table 1. As expected, the difference of drift rates between congruent and incongruent options was larger in the deliberate than the arbitrary conditions. Could the authors comment on the large difference in the noise scaling factor c, which was 10-fold between the two types of decisions? The second result I found difficult to conceptualize was the decay rate k, which doubled in the easy-deliberate than in the hard-deliberate condition. Given that task difficulty was randomized across trials, doesn't this imply that the model (and the participants) adjusted the decay rate according to task difficulty prior to trial onset?

We thank the reviewers for this comment. Note that the noise scaling-factors in deliberate decisions were 35-45% of those in arbitrary (and not 10-fold). We think this makes sense, and—as per the reviewer’s suggestion—we now discuss this, the leak (or rate decay), and the reasons for these values in the manuscript. Briefly, following the reviewer’s comment, we reran the model simulations for the deliberate-hard condition and found that there are two regions in parameter space where the error reaches a local minimum. The first is the one we reported before, (*I_congruent_*, *I_incongruent_*, *k*, *c*) = (0.15, 0.07, 0.21, 0.09). The second, with a somewhat smaller error value is (*I_congruent_*, *I_incongruent_*, *k*, *c*) = (0.18, 0.09, 0.53, 0.11). We therefore now report the values corresponding to the smaller error in the table and elsewhere, but we mention the values associated with the other local minimum too. We agree with the reviewer that the current parameter values make more sense and thank them again for pointing this out to us. We further reran the simulations depicted in Figure 9 based on the new values of the parameters in deliberate hard decisions. Interestingly, the slight difference in trends between deliberate easy and hard in Figure 9B matches that in the empirical data (Figure 3A, B). We now note this briefly in the text.

2) Figure 9A. It is more meaningful to plot the empirical and simulated RT distributions, rather than their fitted γ functions.

As we note in our manuscript, our purpose was to extend the Schurger model to our experimental conditions (Schurger et al., 2012). We therefore followed the same procedure they did, including plotting the γ-function fits instead of the empirical distributions. And we think these should be shown in the paper. However, the reviewer’s point is well taken. So, we decided to plot both. Hence, we added a large inset to Figure 9, where we plot the original data, without fitting it to a γ function.

3) In several instances the authors use the term 'decision onset' when referring (I think) to the completion of the decision. This is potentially confusing because for many readers 'decision onset' may refer to the beginning of deliberation/evidence accumulation which means something entirely different. So I would suggest the authors check their terminology and use 'decision completion' or 'commitment' in such instances.

We see how this could be confusing and are happy to change “decision onset” to “decision completion”. This was now changed in all places in the manuscript.

Minor comments:1) Subsection “Behavioral Results”. DDN.

Fixed. Thank you.

2) Figure 9A. Why was the y-axis labelled as voltage, for RT distributions?

This was a mistake. We thank the reviewer for noticing this and removed the erroneous yaxis label from Figure 9A.

3) Subsection “Model and Simulations” third paragraph and Table 1. I am confused about the scaling parameter 1.45. Does Table 1 show the drift rates only in Region X, and are the drift rates in SMA 1.45 times less than those values? The text and table indicated that the scaling applied only to the deliberate condition, if so, what were the drift rates in SMA in the arbitrary condition? Or do the drift rates in arbitrary decisions in Table 1 refer to the values in SMA?

We thank the reviewer for this comment and now clarify this point further in the manuscript. In particular, as we now hopefully better explain in the caption to Table 1, the values in the table are for Region X for deliberate decisions and for the SMA in arbitrary decisions. The values for the drift-rate parameter in the SMA during deliberate decisions are indeed 1.45 times smaller than those in the table, as the reviewer notes.